# An intestinal sphingolipid confers intergenerational neuroprotection

Wenyue Wang[1], Tessa Sherry ®[1], Xinran Cheng[2], Qi Fan[1], Rebecca Cornell ®[1], Jie Liu[2], Zhicheng Xiao[2] & Roger Pocock ®[1 ✉]

In animals, maternal diet and environment can influence the health of offspring. Whether and how maternal dietary choice impacts the nervous system across multiple generations is not well understood. Here we show that feeding *Caenorhabditis elegans* with ursolic acid, a natural plant product, improves axon transport and reduces adult-onset axon fragility intergenerationally. Ursolic acid provides neuroprotection by enhancing maternal provisioning of sphingosine-1-phosphate, a bioactive sphingolipid. Intestine-to-oocyte sphingosine-1-phosphate transfer is required for intergenerational neuroprotection and is dependent on the RME-2 lipoprotein yolk receptor. Sphingosine-1-phosphate acts intergenerationally by upregulating the transcription of the acid ceramidase-1 (*asah-1*) gene in the intestine. Spatial regulation of sphingolipid metabolism is critical, as inappropriate *asah-1* expression in neurons causes developmental axon outgrowth defects. Our results show that sphingolipid homeostasis impacts the development and intergenerational health of the nervous system. The ability of specific lipid metabolites to act as messengers between generations may have broad implications for dietary choice during reproduction.

Animal exposure to environmental and dietary changes can modify the physiology and development of offspring[1,2]. Certain parentally acquired traits may also be inherited over multiple generations[3,4]. Epigenetic mechanisms of inheritance exploiting small RNAs, DNA methylation and chromatin modifications have previously been defined[2–7]. However, the role of maternal provisioning to offspring as a means of multigenerational inheritance has not been well explored. Maternal provisioning potentially enables the inheritance of information beyond nucleic acids—affording the transmission of lipids, proteins and metabolites[8]. Such information flow could directly report ancestral experience to alter the physiology and development of offspring across generations, and potentially shape the evolutionary trajectory.

Whether altered maternal metabolism and provisioning impacts the neuronal health of offspring across generations is an open question and challenging to resolve. The *Caenorhabditis elegans* model is used to dissect epigenetic mechanisms due to its short generation time and straightforward genetics[3,4,7]. *C. elegans* is also suitable for analysing the temporal effects of dietary and environmental changes, as such conditions can be precisely controlled. Furthermore, defined *C. elegans* neurodevelopmental and disease models allow potential multigenerational effects to be examined[9,10]. We therefore used *C. elegans* to identify molecules that may regulate neuronal health across generations.

Axons are long cytoplasmic projections that transmit information between neurons. Neuronal health maintenance requires the transport of cargo (organelles, RNA, proteins and lipids) along the axonal cytoskeleton[11,12]. Microtubules are major cytoskeletal components comprised of cylindrical structures assembled from α- and β-tubulin heterodimers that mediate intracellular transport[11,13]. Defective microtubule structure disrupts the supply of essential materials and is associated with multiple neurodegenerative disorders[14].

[1]Development and Stem Cells Program, Monash Biomedicine Discovery Institute and Department of Anatomy and Developmental Biology, Monash University, Melbourne, Victoria, Australia. [2]Neuroscience Program, Monash Biomedicine Discovery Institute and Department of Anatomy and Developmental Biology, Monash University, Melbourne, Victoria, Australia. ✉e-mail: roger.pocock@monash.edu

Therefore, identification of molecular mechanisms supporting axonal health under conditions of suboptimal microtubule-associated intracellular transport is important. Posterior lateral mechanosensory (PLM) neurons in *C. elegans* extend axons along the length of the animal to coordinate touch responses[15]. A previous study showed that loss of the MEC-17 (a homologue of αTAT1) α-tubulin acetyltransferase causes progressive adult-onset PLM axon degeneration[9]. MEC-17 loss causes microtubule instability and aberrant axonal transport, resulting in disrupted distribution of mitochondria and synaptic proteins[9]. Degeneration of PLM axons is caused by axon fragility, as the phenotype is suppressed by paralyzing *mec-17*-mutant animals[9]. Furthermore, PLM axon fragility in *mec-17* mutants is exacerbated in animals with increased body length (for example, *lon-2* mutants), probably due to the added demand on axonal transport that is required to maintain a longer axon[9]. We used the well-defined MEC-17 and LON-2-deficient axon fragility model to identify mechanisms that maintain axon integrity across generations.

In this paper we show that supplementation with the natural product ursolic acid (UA), specifically during oocyte production, promotes axon transport and reduces axon fragility intergenerationally in *C. elegans*. Ursolic acid protects axons by upregulating expression of the sphingolipid biosynthetic enzyme acid ceramidase-1 (*asah-1*) gene in the intestine. Intestine-to-oocyte transport of sphingosine-1-phosphate (S1P), a downstream bioactive sphingolipid metabolite, in maternal yolk is required for intergenerational neuroprotection through upregulation of *asah-1* expression in subsequent generations. Furthermore, we show that transcriptional regulation of *asah-1* is dependent on the intestinal transcription factors PQM-1 (GATA zinc finger) and CEH-60 (three-amino-acid loop extension (TALE) class). Together, our study reveals that a short-term dietary supplement during the maternal reproductive period can be neuroprotective over multiple generations—providing a paradigm for intergenerational inheritance of health-promoting traits. We propose that intergenerational regulation of metabolism and metabolic gene expression is a universal principle underlying intergenerational inheritance across evolution.

## UA reduces axon fragility intergenerationally

To identify axon health-promoting molecules, we examined the morphology of PLM neurons in wild-type and *mec-17(ok2109); lon-2(e678)* animals (Fig. 1). As shown previously, the *mec-17(ok2109); lon-2(e678)* animals exhibited approximately 50% penetrant PLM axon breaks in day 3 adults (Fig. 1b,d)[9]. In a screen of natural products, we identified UA as a suppressor of axon fragility in *mec-17(ok2109); lon-2(e678)* animals (Fig. 1 and Extended Data Fig. 1). Ursolic acid is a lipophilic pentacyclic triterpenoid acid found in plants that has broad biological functions, acting as an anti-inflammatory, antioxidant and neuroprotective molecule[16,17] (Fig. 1c). To assess the potency of UA-induced neuroprotection, we fed *mec-17(ok2109); lon-2(e678)* mothers with different concentrations of UA and examined the fragility of the PLM axons (Fig. 1a–d and Extended Data Fig. 2a). We found that the *mec-17(ok2109); lon-2(e678)* $F_1$ progeny of animals incubated with 50 μM UA from larval stage 4 (L4) had reduced axon degeneration (Fig. 1d and Extended Data Fig. 2a). To determine whether the UA-induced suppression of axon fragility was due to reduced body length or motility, we used WormLab tracking. We found no change in the motility or body length of *mec-17(ok2109); lon-2(e678)* animals exposed to UA compared with the controls, which suggested a molecular rather than physical effect (Extended Data Fig. 2b–e).

We investigated whether there was a critical developmental period for UA to reduce PLM axon fragility. For this, we exposed *mec-17(ok2109); lon-2(e678)* animals to UA at the following stages of *C. elegans* development: $P_0$ L3 to L4 (during sperm generation and before oocyte production), $P_0$ L4 to adult (oocytes and sperm present), $F_1$ embryo to L1 (embryogenesis) and $F_1$ L1 to adult (larval stages and adult; Fig. 1e). We found that axon breaks in the $F_1$ adult progeny were only reduced when $P_0$ hermaphrodites containing oocytes and

sperm ($P_0$ L4 to adult) were exposed to UA (Fig. 1e,f). Furthermore, a reduction in PLM axon breaks was not observed in $P_0$ animals or their $F_1$ progeny when the $P_0$ embryos were exposed to UA, suggesting that UA does not penetrate the eggshell (Extended Data Fig. 2f). These data suggest that the UA health-promoting effect is deposited in gametes to maintain axonal health through to adulthood. We thus examined whether UA can protect the nervous system in subsequent generations. We incubated $P_0$ L4 hermaphrodites with dimethylsulfoxide (DMSO; control) or UA for 16 h and then transferred the now adults to untreated plates to lay eggs for 3 h (Fig. 1g). These $F_1$ eggs therefore underwent oocyte maturation during UA exposure. When the $F_1$ animals reached the L4 stage, a cohort were transferred to lay eggs for analysis of the next generation and the remainder matured until day 3 of adulthood to examine PLM axon health (Fig. 1g,h). We repeated this process for the subsequent generations. We found that incubation with UA during $P_0$ oocyte maturation reduced PLM axon breaks in the $F_1$ and $F_2$ generations but not the $F_3$ generation (Fig. 1g,h), revealing an intergenerational neuroprotective effect of UA.

## Inherited neuroprotection requires intestine–oocyte transport

How does UA protect the nervous system intergenerationally? As the functional period is during oocyte production and maturation, we hypothesized that UA may affect maternal yolk provisioning to oocytes. Yolk synthesized in the intestine of *C. elegans* hermaphrodites contains lipids and lipoproteins that provide oocytes, and thus embryos, with nutrients for development[8]. Oocyte yolk import occurs through endocytosis and requires receptor-mediated endocytosis-2 (RME-2), a low-density lipoprotein receptor[18]. We found that RME-2 is required for UA to reduce PLM neuron fragility, providing support for a role for intestine–oocyte transport (Fig. 2a). Animals lacking RME-2 are also defective in the transport of RNAs that are major transmitters of epigenetic inheritance[19]. The HRDE-1 argonaute and ZNFX-1 helicase are critical for small-RNA inheritance; however, *hrde-1* and *znfx-1* are dispensable for UA to reduce PLM breaks intergenerationally (Fig. 2b,c)[20–22]. These data suggest that UA stimulates alternative factors in the maternal yolk and that this information optimizes the oocyte/embryonic environment to promote axon health.

## UA induces intestinal acid ceramidase expression

To identify the factor(s) regulated by UA, we examined the transcriptomes of synchronized L4 larvae that had been exposed to UA (or DMSO) for 12 h (Fig. 2d, Extended Data Fig. 3 and Supplementary Table 1). We identified 49 dysregulated genes (false discovery rate of 0.02) and surveyed this dataset for genes expressed in the intestine (the source of yolk) that potentially control lipid metabolism. We detected increased *asah-1* transcript levels in the animals exposed to UA, which we confirmed by quantitative PCR (qPCR) analysis (Fig. 2d). The *asah-1* gene encodes an acid ceramidase that hydrolyses ceramide into fatty acid and sphingosine[23]. To assess the requirement of *asah-1* for UA to reduce PLM axon fragility, we performed RNA-mediated interference (RNAi) to knockdown *asah-1* in *mec-17(ok2109); lon-2(e678)* animals incubated with UA (Fig. 2e). Treatment with UA did not reduce the PLM axon fragility of *asah-1*-knockdown animals (Fig. 2e). This result was independently confirmed in animals with a genetic *asah-1* deletion (*asah-1(tm495)* deletion allele) and showed that *asah-1* loss caused an increase in PLM axon breaks in untreated animals (Fig. 2f). To identify the tissue in which *asah-1* is expressed and potentially regulated by UA, we monitored the spatiotemporal expression of *asah-1* using a 2,000-bp sequence upstream of *asah-1* to drive nuclear-localized green fluorescent protein (GFP; Fig. 2g and Extended Data Fig. 4). We detected GFP exclusively in the intestinal cells of *Pasah-1::gfp* animals from late embryos through to adult (Fig. 2g and Extended Data Fig. 4). At all stages, we observed an anterior bias in intestinal GFP expression

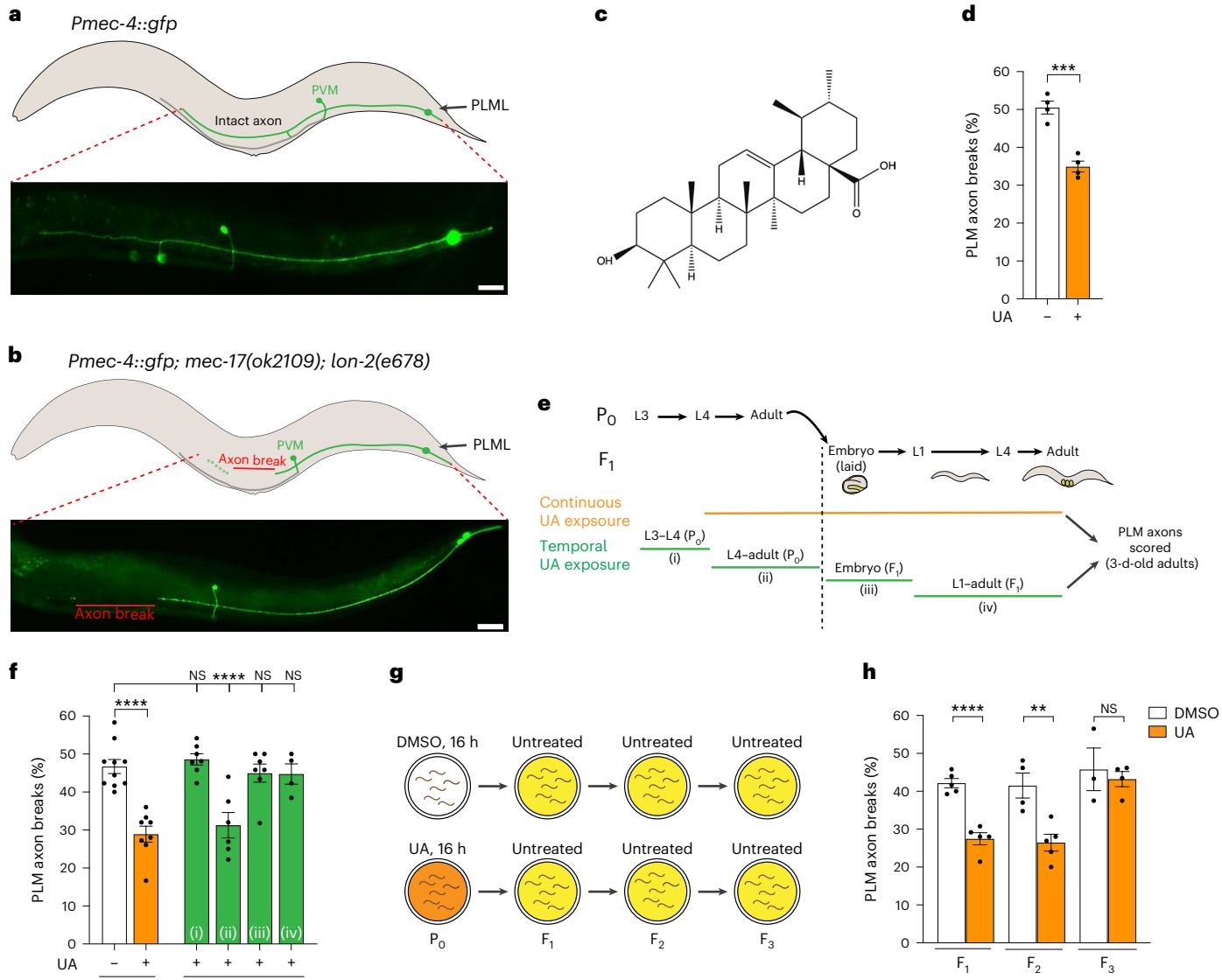

**Fig. 1 | UA promotes neuronal health across generations. a,b**, Schematics (top) and fluorescence micrographs (bottom) of posterior lateral mechanosensory left (PLML) anatomy in wild-type (**a**) and *mec-17(ok2109); lon-2(e678)* (**b**) animals expressing the *Pmec-4::gfp* transgene (*zdIs5*). Left lateral view, anterior to the left. Scale bars, 25 μm. **b**, A typical PLM axon break (red line) observed in adult (3-d-old) *mec-17(ok2109); lon-2(e678)* animals is indicated. **c**, Chemical structure of UA. **d**, Adult *mec-17(ok2109); lon-2(e678)* $F_1$ animals exposed to UA (50 μM) from $P_0$ L4 to day 3 of $F_1$ had reduced PLM axon breaks. **e**, Timeline of UA exposure. The stages of *C. elegans* development relevant to UA exposure in the $P_0$ and $F_1$ generations are shown (top). Vertical dashed line, demarcation between the $P_0$ and $F_1$ generations. **f**, Continuous UA exposure ($P_0$ L4 larva to $F_1$ adult) reduces PLM axon breaks in *mec-17(ok2109); lon-2(e678)* animals. Specifically, UA exposure of $P_0$ animals from the L4 larval stage to adult (ii), but not earlier

(i) or later ((iii) and (iv)) stages results in a reduction in PLM axon breaks in *mec-17(ok2109); lon-2(e678)* animals. **g**, Experimental scheme for the intergenerational inheritance experiment. $P_0$ animals (L4 larvae) were treated with DMSO (control) or UA for 16 h. Animals of each generation were allowed to lay eggs on untreated plates for 3 h and the adults (3-d-old) were assessed for axon breaks. **h**, The progeny of *mec-17(ok2109); lon-2(e678)* $P_0$ mothers exposed to UA for 16 h (L4 to young adult) have reduced PLM axon breaks for two generations ($F_1$ and $F_2$). **d,f,h**, $n = 101$ and 103 (**d**); **f**, $n = 249, 199, 173, 151, 166$ and 107 (**f**); and $n = 126, 128, 103, 128, 77$ and 97 (**h**) hermaphrodite animals per condition (left to right). $P$ values were determined using a one-way analysis of variance (ANOVA; **f**) or unpaired Student's *t*-test (**d,h**). ****$P \le 0.0001$; ***$P \le 0.001$; **$P \le 0.01$; and NS, not significant. Error bars indicate the s.e.m. Source data are provided.

(Fig. 2g and Extended Data Fig. 4). Increased fluorescence was observed in the intestinal nuclei of *Pasah-1::gfp* L4 animals exposed to UA for 12 h (Fig. 2g). Thus, UA induces *asah-1* transcription in the intestine.

## Intestinal ASAH-1 expression protects axons

As *asah-1* was upregulated in animals exposed to UA, we investigated whether *asah-1* overexpression in *mec-17(ok2109); lon-2(e678)* animals could mimic UA-induced neuroprotection. Single-cell sequencing

corroborates *asah-1* intestinal expression; however, low level expression can be detected in other cells/tissues, including the PLM neurons[24,25]. We therefore overexpressed *asah-1* in *mec-17(ok2109); lon-2(e678)* animals using the following heterologous promoters: intestine (*ges-1*), hypodermis (*dpy-7*), muscle (*myo-3*) and mechanosensory neurons (*mec-4*; Fig. 3a and Extended Data Fig. 5)[26–29]. Overexpression of *asah-1* in the intestine, but not hypodermis or muscle, reduced PLM axon breaks in the *mec-17(ok2109); lon-2(e678)* animals (Fig. 3a). We further found that presence of the *Pges-1::asah-1* transgene in progeny is not

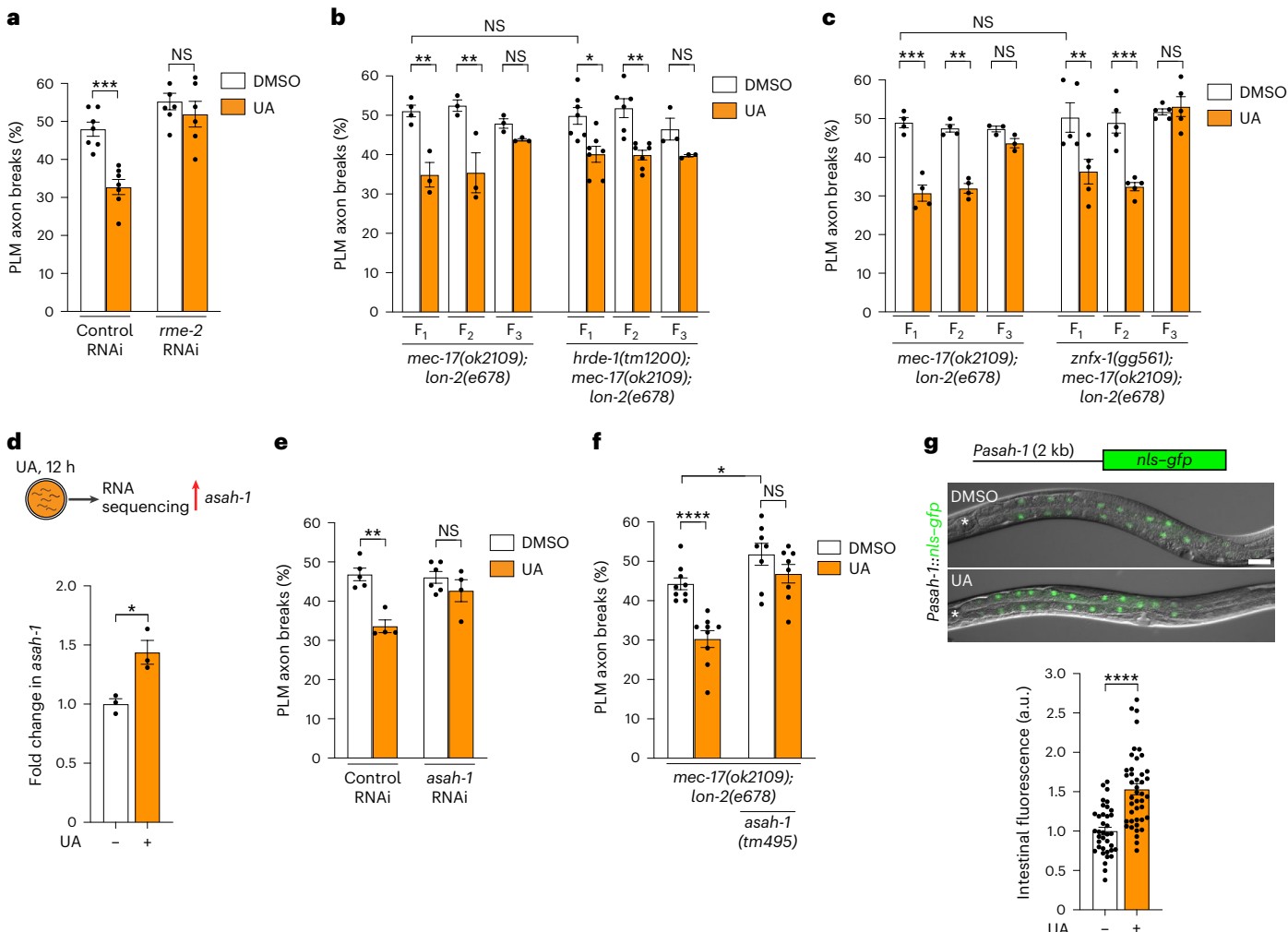

**Fig. 2 | UA induces acid ceramidase (*asah-1*) expression to promote neuronal health. a**, RNAi-mediated knockdown of *rme-2* suppresses UA-induced reduction of PLM axon breaks in *mec-17(ok2109); lon-2(e678)* animals. **b,c**, The progeny of $P_0$ mothers (*mec-17(ok2109); lon-2(e678)*, *hrde-1(tm1200); mec-17(ok2109); lon-2(e678)* (**b**) and *znfx-1(gg561); mec-17(ok2109); lon-2(e678)* (**c**) animals) exposed to UA for 16 h (L4 to young adult) have reduced PLM axon breaks for two generations ($F_1$ and $F_2$). Experimental timing as in Fig. 1g. **d**, Schematic of transcriptome analysis following UA exposure (top). Total RNA was isolated from L4 larvae exposed to UA for 12 h (compared with the DMSO control). *asah-1* was upregulated following UA exposure (Supplementary Table 1). The upregulation of *asah-1* messenger RNA in wild-type L4 larvae after 12 h of UA exposure was independently confirmed using qPCR (bottom; $n = 3$ independent experiments). The housekeeping gene *cdc-42* was used as the control. **e,f**, Loss of *asah-1* by RNAi (**e**) or gene deletion (**f**) suppresses the ability of UA to reduce PLM axon breaks in

*mec-17(ok2109); lon-2(e678)* animals. **g**, Representative fluorescence micrographs (top) and calculated levels (bottom) of the *Pasah-1::nls::gfp* transcriptional reporter. Expression was detected in the intestine from late embryogenesis through to adult (see Extended Data Fig. 4). *Pasah-1::nls::gfp* expression increased in animals exposed to UA for 12 h compared with the controls (DMSO); a.u. arbitrary units. Lateral views, anterior to the left, of L4 larvae are shown. Pharynx marked by a white asterisk. Scale bar, 25 μm. **a–c,e–g**, $n = 186, 186, 162$ and $154$ (**a**); $n = 90, 69, 61, 65, 71, 71, 160, 156, 145, 158, 73$ and $68$ (**b**); $n = 98, 101, 99, 100, 76, 71, 115, 115, 115, 114, 116$ and $121$ (**c**); $n = 126, 101, 148$ and $94$ (**e**); $n = 223, 220, 202$ and $194$ (**f**); and $n = 37$ and $43$ (**g**) hermaphrodite animals per condition (left to right). **a–g**, $P$ values were determined using an ANOVA (**a–c,e**) or unpaired Student's $t$-test (**d,f,g**). ****$P \leq 0.0001$; ***$P \leq 0.001$; **$P \leq 0.01$; *$P \leq 0.05$; and NS, not significant. Error bars indicate the s.e.m. Source data are provided.

required for PLM neuroprotection (Fig. 3b), providing support for the idea that ASAH-1 activity in the hermaphrodite intestine protects the PLM neurons in the next generation.

Overexpression of *asah-1* in the mechanosensory neurons caused PLM axon outgrowth defects in *mec-17(ok2109); lon-2(e678)* animals (Extended Data Fig. 5a,c), precluding analysis of PLM fragility. Overexpression of *asah-1* in the neurons of wild-type animals—either in the mechanosensory neurons (*mec-4* promoter) or pan-neuronally (*rab-3* promoter)—also caused extensive axon outgrowth defects in the anterior lateral mechanosensory (ALM) and PLM neurons, which develop embryonically, but not the post-embryonic posterior ventral microtubule neurons (Fig. 3c,d and Extended Data Fig. 5a–c). This potential embryonic effect was supported by the detection of PVQ

axon guidance defects when *asah-1* was neuronally overexpressed (Extended Data Fig. 5f,g). However, neuronal *asah-1* overexpression did not cause overt motility defects, suggesting intact global nervous system architecture. These data reveal that intestinal *asah-1* reduces PLM axon fragility in animals with defective microtubule stability, and inappropriate neuronal *asah-1* expression—which is likely to be associated with disrupted sphingolipid homeostasis—causes cell-autonomous neurodevelopmental defects.

Sphingolipids are amphipathic bioactive molecules with multiple cellular functions, including cell adhesion and migration, cell death and cell proliferation[30]. Sphingolipid homeostasis is maintained through the de novo or salvage pathways (Fig. 4a)[31]. The serine palmitoyltransferase (SPT) protein complex is the rate-limiting enzyme in

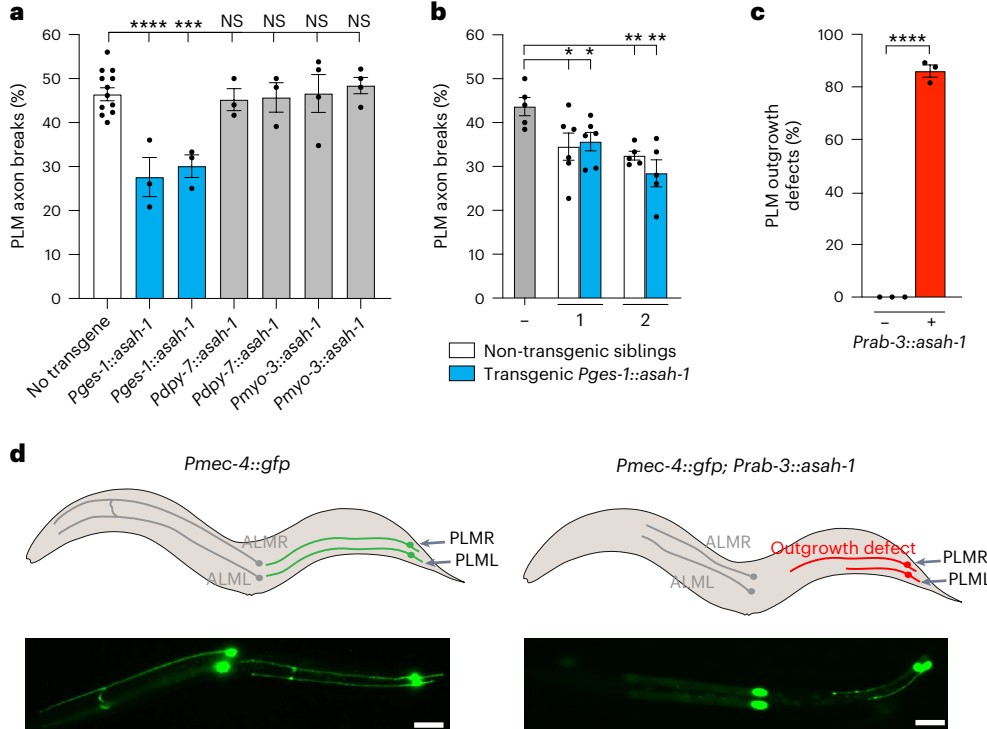

**Fig. 3 | Intestinal and neuronal *asah-1* expression has opposing effects on axonal development and health. a**, Expression of *asah-1* cDNA driven by the heterologous intestinal promoter (*ges-1*), but not the hypodermal (*dpy-7*) or muscle (*myo-3*) promoters, resulted in reduced PLM axon breaks in *mec-17(ok2109); lon-2(e678)* animals. **b**, PLM axon breaks were reduced in *mec-17(ok2109); lon-2(e678)* animals derived from *Pges-1::asah-1* transgenic animals independently of inheritance of the transgene. The *mec-17(ok2109); lon-2(e678)* animals were either injected with *Pges-1::asah-1* (transgenic lines nos. 1 and 2) or uninjected (−). **c,d**, Overexpression of *asah-1* in the nervous system (*rab-3* promoter) caused developmental axon outgrowth defects in the mechanosensory neurons of wild-type animals expressing the *Pmec-4::gfp*

transgene (*zdIs5*). **c**, Proportion of PLM axon outgrowth defects in *Pmec-4::gfp*- and *Pmec-4::gfp; Prab-3::asah-1*-expressing L1 larvae. **d**, Schematic (top) and fluorescence micrographs (bottom) of ALM (left/right) and PLM (left/right) axons in wild-type (left) and *Prab-3::asah-1*-expressing animals (right). The typical axon outgrowth defect observed is marked in red. Left lateral view, anterior to the left. Scale bars, 25 μm. **a**–**c**, $n$ = 305, 72, 80, 75, 72, 98 and 97 (**a**); $n$ = 126, 147, 149, 123 and 124 (**b**); and $n$ = 62 and 95 (**c**) hermaphrodite animals per condition (left to right). $P$ values were determined using an ANOVA (**a**) or unpaired Student's $t$-test (**b**,**c**). Error bars indicate the s.e.m. ****$P ≤ 0.0001$; ***$P ≤ 0.001$; **$P ≤ 0.01$; *$P ≤ 0.05$; and NS, not significant. Source data are provided.

the de novo pathway that generates ceramide, the ASAH-1 substrate (Fig. 4a)[32]. SPTL-1 knockout causes *C. elegans* embryonic lethality/larval arrest, probably due to the loss of multiple sphingolipids. Therefore, to assess SPTL-1 regulation of PLM axon fragility, we overexpressed *sptl-1* complementary DNA in the intestine. Overexpression of *sptl-1* reduced PLM axon breaks in *mec-17(ok2109); lon-2(e678)* animals, confirming that ceramide or its derivatives are important for axon health (Fig. 4b). Indeed, when *mec-17(ok2109); lon-2(e678)* animals were incubated with two distinct ceramides containing different fatty-acid-chain lengths (Cer20 (d18:1/20:0) and Cer22 (d18:1/22:0)), there was a reduction in PLM axon breaks (Fig. 4c). In the salvage pathway, lysosomal membrane ceramide is hydrolysed to sphingosine by acid ceramidases, and sphingosine phosphorylation by sphingosine kinases (SphK) generates S1P (Fig. 4a)[31]. We found that intestinal *sphk-1* expression reduced PLM axon breaks in *mec-17(ok2109); lon-2(e678)* animals, as also shown by overexpression of *sptl-1* or *asah-1* (Fig. 4b,d,e). Furthermore, *sphk-1* loss increased PLM axon fragility and prevented UA-induced reduction of PLM axon breaks in *mec-17(ok2109); lon-2(e678)* animals (Extended Data Fig. 6a,b). Overexpression of *asah-1* in the intestine reduced PLM axon breaks (Fig. 4d). To determine whether *sphk-1*, and thus potentially S1P generation, is required for ASAH-1 to perform this neuroprotective function, we knocked down *sphk-1* in animals overexpressing intestinal *asah-1* and evaluated the PLM axon fragility of *mec-17(ok2109); lon-2(e678)* animals (Fig. 4f). We found that *sphk-1* RNAi suppressed the beneficial effect of *asah-1* overexpression on PLM

axon fragility (Fig. 4f). These data suggest that UA neuroprotection is S1P-dependent.

## S1P prevents axon fragility intergenerationally

We examined the role of S1P in PLM axonal health directly by incubating *mec-17(ok2109); lon-2(e678)* animals with different S1P concentrations for one generation ($P_0$ L4 to 3-d-old $F_1$ adult). A concentration of 20 μM S1P reduced PLM axon fragility (Fig. 5a,b). To determine the functional period of S1P, we exposed *mec-17(ok2109); lon-2(e678)* animals to S1P for two time periods: $P_0$ L4 larvae to adult and L1 larvae to adult. S1P only reduced axon fragility in the $F_1$ progeny when provided to hermaphrodites containing oocytes ($P_0$ L4 to adult)—the same functional period as UA (compare Figs. 5c and 1f). Furthermore, two generations of the progeny of $P_0$ animals incubated with S1P from L4 larvae to adult (16 h) had to reduced axonal fragility, revealing an intergenerational effect (Fig. 5d,e). These data suggest that intestinal S1P is transported within the yolk to oocytes to promote PLM axonal health in subsequent generations. To examine whether S1P can undergo intestine–oocyte transport, we fed wild-type L4 larvae with S1P–fluorescein, a fluorescently labelled S1P analogue (Fig. 5f). Fluorescence was observed in the intestinal tract within 1 h of feeding, suggesting that S1P–fluorescein is not immediately metabolized (Extended Data Fig. 6c). After 16 h of S1P–fluorescein feeding, we detected fluorescence in proximal oocytes, suggesting yolk-dependent intestine–oocyte transport (Fig. 5f). To determine

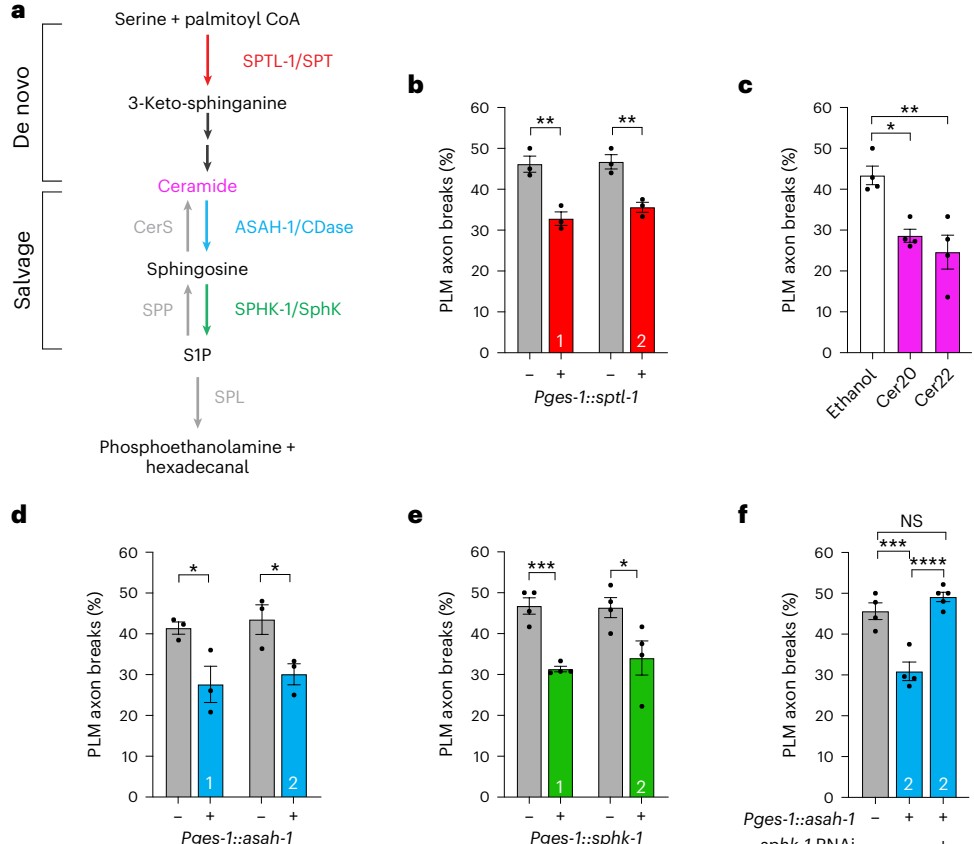

**Fig. 4 | Intestinal *asah-1* neuroprotection requires SphK expression.**
**a**, *C. elegans* orthologues of de novo- and salvage sphingolipid-pathway
components. Sphingolipid metabolic enzymes (nematode and mammalian
orthologue) and sphingolipid intermediates are shown. CerS, ceramide
synthase; CDase, ceramidase; SPP, S1P phosphatase; and SPL, S1P lyase.
**b**,**d**,**e**, The function of SPTL-1, ASAH-1 and SPHK-1 (coloured lettering in **a**) in
PLM axon fragility was examined. Overexpression of *sptl-1* (**b**), *asah-1* (**d**) or
*sphk-1* (**e**) cDNA in the intestine (*ges-1* promoter) reduces PLM axon breaks
in *mec-17(ok2109); lon-2(e678)* animals. Two independent transgenic lines
(nos. 1 and 2, as indicated in the coloured bars) were analysed. **c**, Exposure

of P_0 mothers to ceramide for 16 h (L4 to young adult) results in a reduction
of PLM axon breaks in *mec-17(ok2109); lon-2(e678)* animals. **f**, RNAi-induced
knockdown of *sphk-1* suppresses the ability of intestinal *asah-1* expression to
reduce PLM axon breaks in *mec-17(ok2109); lon-2(e678)* animals. Transgenic
line no. 2 from **d** was used. **b**–**f**, $n = 69, 73, 75$ and 73 (**b**); $n = 92, 81$ and 88 (**c**);
$n = 76, 72, 79$ and 80 (**d**); $n = 94, 102, 99$ and 98 (**e**); and $n = 99, 97$ and 116 (**f**)
hermaphrodite animals per condition (left to right). $P$ values were determined
using an ANOVA (**c**,**f**) or unpaired Student's $t$-test (**b**,**d**,**e**). ****$P \leq 0.0001$;
***$P \leq 0.001$; **$P \leq 0.01$; *$P \leq 0.05$; and NS, not significant. Error bars indicate
the s.e.m. Source data are provided.

the importance of intestine–oocyte S1P transport to prevent axon
fragility, we knocked down *rme-2* in *mec-17(ok2109); lon-2(e678)* animals incubated with S1P. We found that S1P neuroprotection requires
*rme-2* (Fig. 5g), revealing that S1P intergenerational neuroprotection
requires intestine–oocyte transport.

## UA and S1P enhance PLM axon transport
A previous study showed that MEC-17 controls the efficient transport of protein cargos throughout the PLM axon to maintain neuronal
health[9]. We therefore investigated whether reduced PLM axon fragility
afforded by UA and S1P to animals lacking MEC-17 is due to improved
axon transport. We examined the localization of fluorescent reporters for UNC-104 (a kinesin-3 motor protein) and RAB-3 (a synaptic
vesicle-associated small guanosine triphosphatase), which exhibited
inappropriate posterior axonal pooling in *mec-17(ok2109)* animals
(Fig. 6a,b). We found that *mec-17(ok2109)* animals exposed to UA or
S1P had reduced axon transport defects (Fig. 6a,b), suggesting that UA
and S1P promote the transport of protein cargo along the PLM axon.
In a parallel approach to examine the potential effect of UA and S1P on
microtubule health, we treated animals with colchicine—a microtubule
destabilizing agent[33]. Previous work showed that colchicine increases
PLM axon breaks in animals lacking MEC-17 (ref. 9). We found that

UA and S1P reduce PLM axon breaks in *mec-17(ok2109); lon-2(e678)*
animals exposed to colchicine (Fig. 6c). Thus, in the presence of two
microtubule destabilizing conditions (MEC-17 loss and colchicine)
UA and S1P reduce PLM axon fragility. Critically, we found that UA and
S1P protect wild-type PLM axons exposed to high levels of colchicine,
revealing that the neuroprotective effect is not directly associated
with MEC-17 loss (Fig. 6d).

## UA and S1P provide general neuroprotection
To assess the potential broad applicability of UA and S1P neuroprotection, we examined multiple distinct genetic and neuronal models
(Fig. 6e–g). A previous study showed that increased microtubule stability can reduce PLM axon breaks caused by mutations to the LIN-14
transcription factor[34]. We exposed *lin-14(n355n679) lon-2(e678)* animals
to UA or S1P and found that PLM breaks were reduced, suggesting that
UA and S1P can generally promote PLM axonal health (Fig. 6e). Next, we
examined two other neuron classes—the D-type GABAergic motor neurons and the PVQ interneurons (Fig. 6f,g). LIN-14 loss causes axon breaks
of GABAergic D-type motor neuron commissures[34]. Interestingly, we
found that S1P, but not UA, suppresses D-type motor axon breaks in
*lin-14(n355n679)* animals (Fig. 6f), suggesting distinct mechanisms
of action or route of entry for S1P in this context. Finally, we explored

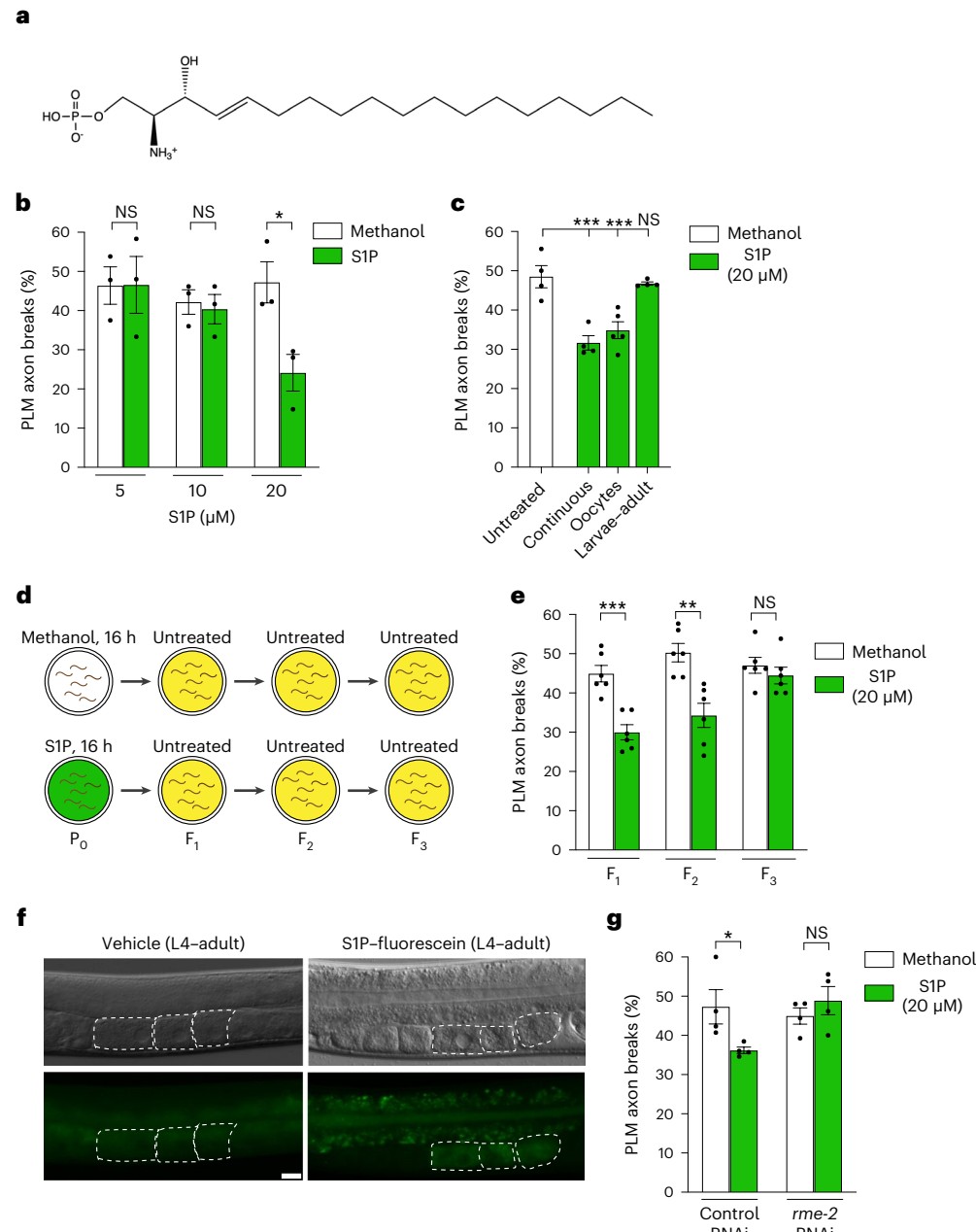

**Fig. 5 | S1P protects PLM neurons intergenerationally. a**, Chemical structure of S1P. **b**, Continuous exposure ($P_0$ L4 larva to $F_1$ adult) to 20 μM S1P reduces PLM axon breaks in *mec-17(ok2109); lon-2(e678)* animals. **c**, PLM axon breaks were reduced in *mec-17(ok2109); lon-2(e678)* animals when $P_0$ mothers (L4 to young adult), but not $F_1$ larvae, were exposed to S1P for 16 h. **d**, Experimental scheme for the intergenerational inheritance experiment. $P_0$ animals (L4 larvae) were treated with methanol (control) or S1P for 16 h. Animals of each generation were allowed to lay eggs on untreated plates for 3 h and 3-d-old adults were assessed for axon breaks. **e**, The progeny of $P_0$ mothers (*mec-17(ok2109); lon-2(e678)* animals) exposed to S1P for 16 h (L4 to young adult) had reduced PLM axon breaks for two generations ($F_1$ and $F_2$). **f**, S1P–fluorescein was detected in the oocytes of wild-type

adult hermaphrodites after 16 h of feeding. A vehicle-treated control (left) and an animal following S1P–fluorescein treatment (right) are shown. Nomarski micrographs (top) and fluorescence images (bottom) of the same animals are provided. The dashed white lines outline oocytes. Lateral view, anterior to the left. Scale bar, 25 μm. **g**, RNAi-mediated knockdown of *rme-2* suppresses S1P-induced reduction of PLM axon breaks in *mec-17(ok2109); lon-2(e678)* animals. **b,c,e,g**, $n$ = 73, 73, 78, 74, 76 and 79 (**b**); $n$ = 105, 104, 135 and 107 (**c**); $n$ = 158, 163, 153, 157, 157 and 155 (**e**); and $n$ = 104, 105, 107 and 104 (**g**) hermaphrodite animals per condition (left to right). $P$ values were determined using an ANOVA (**c,g**) or unpaired Student's $t$-test (**b,e**). ***$P \le 0.001$; **$P \le 0.01$; *$P \le 0.05$; and NS, not significant. Error bars indicate the s.e.m. Source data are provided.

the potential function of UA and S1P in a neurodevelopmental model. We previously identified a point mutation in the CED-10 Rac GTPase that causes PVQ axon defects[35]. We found that both UA and S1P reduced PVQ axon defects in *ced-10(rp100)* animals (Fig. 6g). Together, these data reveal that UA and S1P can promote axonal health in distinct genetic and neuronal contexts.

## PQM-1 and CEH-60 control ASAH-1 expression

How does UA induce *asah-1* intestinal expression to provide intergenerational neuroprotection? As UA induces *asah-1* transcription (Fig. 2d,g), we reasoned that transcription factors related to intestinal stress or metabolic control may regulate *asah-1*. We examined the *asah-1* promoter for conserved transcription factor-binding motifs

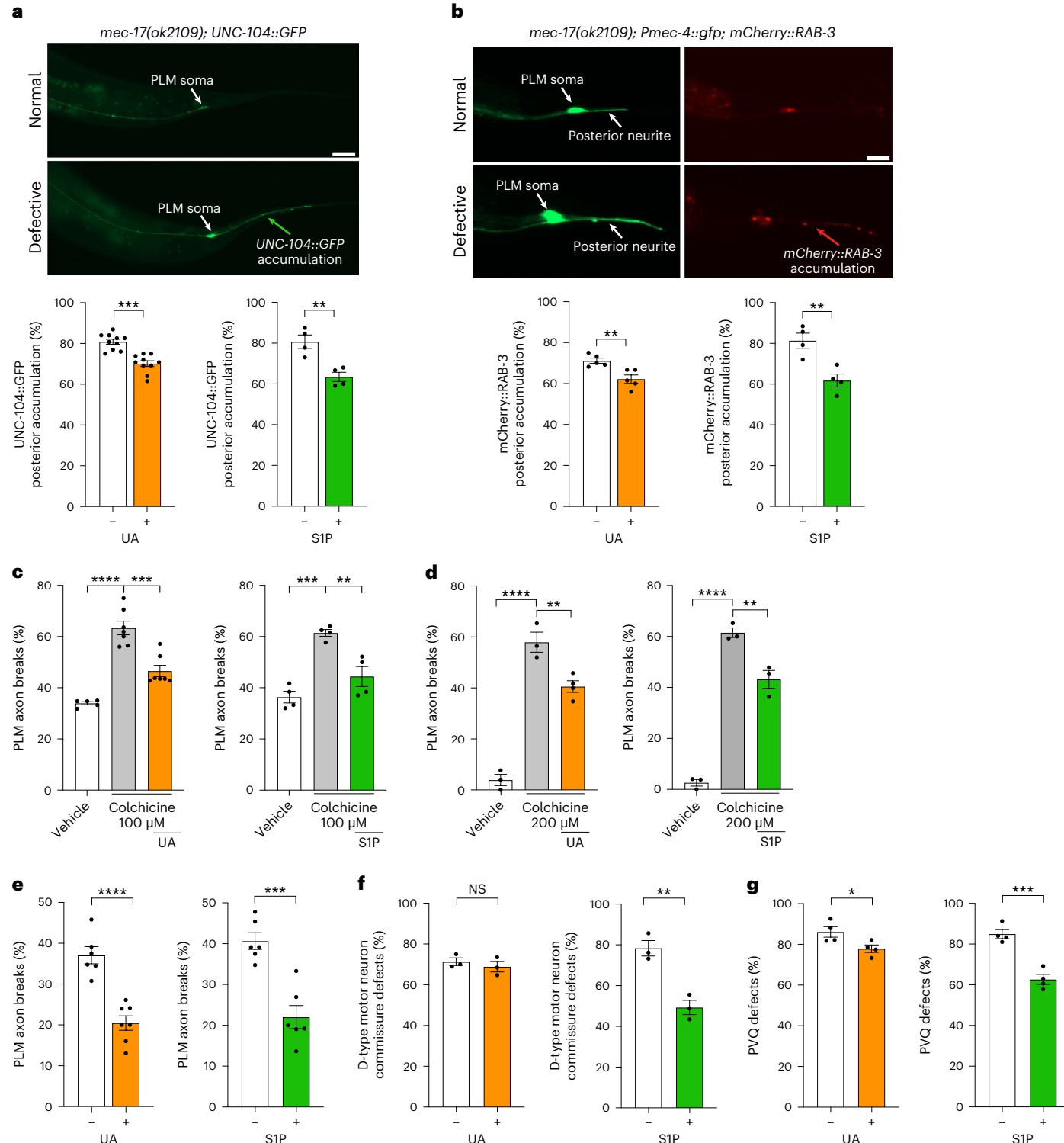

**Fig. 6 | UA and S1P promote PLM axon transport and microtubule stability.**
**a,b,** Continuous exposure to UA or S1P reduces inappropriate posterior accumulation of UNC-104::GFP (kinesin; **a**) and mCherry::RAB-3 (pre-synaptic guanosine triphosphatase; **b**) in the PLM axons of *mec-17(ok2109)* animals. Scale bars, 25 μm. **c,d,** Continuous exposure to UA or S1P reduces colchicine-induced PLM axon breaks in *mec-17(ok2109); lon-2(e678)* (**c**) and wild-type (**d**) animals. Wild-type and *mec-17(ok2109); lon-2(e678)* animals were exposed to 200 μM and 100 μM colchicine, respectively. Two-day-old adult animals were scored. **e,** Continuous UA or S1P exposure reduces PLM axon breaks in *lin-14(n355n679) lon-2(e678)* animals. **f,** Continuous exposure to S1P, but not UA, reduces D-type

motor neuron commissure defects in *lin-14(n355n679)* animals. **g,** Continuous exposure to UA or S1P reduces PVQ axon defects in *ced-10(rp100)* animals (**g**). **a–g,** Continuous exposure: P₀ to L4 larva to F₁ to adult; $n$ = 245, 247, 103 and 99 (**a**); $n$ = 107, 119, 96 and 94 (**b**); $n$ = 112, 152, 146, 99, 101 and 102 (**c**); $n$ = 76, 74, 96, 76, 75 and 67 (**d**); $n$ = 149, 175, 138 and 144 (**e**); $n$ = 109, 106, 97 and 107 (**f**); and $n$ = 102, 117, 156 and 150 (**g**) hermaphrodite animals per condition (left to right). $P$ values were determined using an ANOVA (**c,d**) or unpaired Student's $t$-test (**a,b,e–g**). ****$P \le 0.0001$; ***$P \le 0.001$; **$P \le 0.01$; *$P \le 0.05$; and NS, not significant. Error bars indicate the s.e.m. Source data are provided.

and surveyed publicly available chromatin immunoprecipitation–sequencing (ChIP–seq) data for transcription factors exhibiting peaks upstream of the *asah-1* locus (Fig. 7a and Extended Data Fig. 7a). We identified ChIP–seq peaks for two transcription factors in the *asah-1* promoter: PQM-1 (a GATA zinc-finger transcription factor) and CEH-60 (an orthologue of mammalian PBX; a TALE class transcription factor; Fig. 7a and Extended Data Fig. 7a). These ChIP–seq peaks coincide with putative binding sites we identified in silico (Fig. 7a and Extended Data Fig. 7a). PQM-1 and CEH-60 function in the intestine to balance transcriptional networks governing stress responses and nutrient supply to progeny[36,37]. We therefore investigated their potential role in controlling PLM axon fragility and *asah-1* expression.

To directly assess whether PQM-1 and CEH-60 transcriptionally regulate endogenous *asah-1*, we generated a fluorescent reporter by inserting a *f2A-gfp-h2B* cassette immediately downstream of the *asah-1* coding sequence (Fig. 7a,b). As ribosomal skipping occurs at the *f2A* sequence, independently translated GFP–H2B protein is visualized in nuclei (Fig. 7b and Extended Data Fig. 7b)[38]. We detected GFP–H2B expression in anterior intestinal nuclei (except the most anterior nuclei pair), with weak/undetectable expression in the posterior (Fig. 7b and Extended Data Fig. 7b). This anteriorly biased expression mirrors the *asah-1* transcriptional reporter (compare Fig. 7b with Extended Data Fig. 4), suggesting transcriptional regulation of spatial expression. GFP–H2B expression by the *asah-1* endogenous reporter was induced by exposure to UA (Fig. 7c and Extended Data Fig. 8a), providing support for our earlier analysis (Fig. 2g). We next crossed *pqm-1* and *ceh-60* loss-of-function mutants into *asah-1::f2A-gfp-h2B* animals and measured GFP–H2B fluorescence in the intestinal nuclei. Loss of PQM-1 elevated GFP–H2B expression and loss of either PQM-1 or CEH-60 prevented GFP–H2B induction by UA (Fig. 7d). As PQM-1 and CEH-60 are also required for yolk delivery to oocytes due to their control of intestinal vitellogenin expression[36,39], we examined their role in UA-induced neuroprotection. We found that loss of PQM-1 or CEH-60 inhibited UA neuroprotection in *mec-17(ok2109); lon-2(e678)* animals (Fig. 7e–g). Therefore, PQM-1 and CEH-60 are required for induction of *asah-1* by UA and for neuroprotection in response to UA exposure.

As PQM-1 and CEH-60 control metabolic homeostasis and yolk production[36,37], *asah-1* dysregulation in these transcription factor mutants may be a secondary consequence of disrupted lipid homeostasis. To directly examine the importance of PQM-1 and CEH-60 binding for *asah-1* expression, we mutated the putative binding motifs that coincide with the PQM-1 and CEH-60 ChIP peaks in the *asah-1* promoter (Fig. 7a and Extended Data Fig. 7a). We found that GFP–H2B intestinal expression was induced by mutations to the PQM-1 binding motif; mutations to the CEH-60 site had a weaker effect (Fig. 7h and Extended Data Fig. 8b). These data support a direct role for these transcription factors in *asah-1* transcriptional repression and suggest that PQM-1 and CEH-60 occupancy at the *asah-1* promoter may control sphingolipid homeostasis. We also noticed that PQM-1 motif mutations caused a posterior shift in GFP–H2B expression, revealing spatial regulation of intestinal expression (Fig. 7h). Given that PQM-1 motif mutations increased *asah-1* expression, we investigated whether this is neuroprotective. When the *asah-1::f2A-gfp-h2B* (PQM-1 motif mutant) strain was crossed into *mec-17(ok2109); lon-2(e678)* animals, we observed robust reduction in PLM axon fragility (Fig. 7i). This revealed that *asah-1* derepression has a functionally relevant outcome for axonal health.

## S1P levels regulate ASAH-1 expression intergenerationally

Ceramide hydrolysis is the only catabolic source of sphingosine and therefore ceramidase (for example, ASAH-1) activity is a rate-limiting step in governing intracellular sphingosine and S1P levels[40]. We hypothesized that S1P intergenerational inheritance is triggered by an imbalance of sphingolipid homeostasis, such as increased S1P.

Thus, detection and recycling of S1P to sphingosine and then ceramide may feedback to induce *asah-1* expression in progeny. To explore this possibility, we exposed *asah-1::f2A-gfp-h2B* $P_0$ animals to S1P for 16 h (L4 to young adult) and measured the intestinal expression of GFP–H2B in the subsequent generations. Remarkably, the levels of GFP–H2B were increased in the $F_1$ and $F_2$ progeny following $P_0$ S1P exposure (Fig. 7j), establishing a regulatory mechanism for S1P-induced intergenerational inheritance.

## Discussion

Here we show that short-term maternal dietary supplements can prevent axon fragility intergenerationally in *C. elegans* (Extended Data Fig. 8c). Supplementation with UA induces intestinal expression of *asah-1*, which encodes a rate-limiting enzyme in the sphingolipid salvage pathway. Our genetic evidence reveals that intestinal overexpression of either the de novo or salvage sphingolipid pathway prevents axon fragility and that SphK expression is required to promote axon health. The metabolite generated by SphK is S1P, and we found S1P supplementation also prevents axon fragility intergenerationally. Neuroprotection is dependent on S1P transport in intestinally derived yolk to oocytes[18]. Axon stability requires the correct transport of protein cargo along microtubules[41]. We found that UA and S1P improve axon transport and protect against axon fragility caused by a microtubule-destabilizing drug. Thus, S1P probably promotes microtubule stability, as has previously been shown for ceramide in a cell culture model[42]. We discovered that the transcription factors PQM-1 and CEH-60, known regulators of intestinal lipid homeostasis[36,37], are required for *asah-1* regulation and UA neuroprotection. Remarkably, mutation of a PQM-1-repressive element in the *asah-1* promoter is sufficient to reduce axon fragility, corroborating the critical role for *asah-1* in the regulation of neuronal health.

### Sphingolipid locale impacts neuron development and health

We showed that neuronal development and health require correct spatial regulation of sphingolipid biosynthetic enzymes. Elevated intestinal *asah-1* expression reduced PLM axon fragility in an SphK-dependant manner. In contrast, overexpression of *asah-1* in the nervous system caused developmental axon outgrowth defects in the ALM and PLM neurons. However, the absence of overt motility defects in animals overexpressing *asah-1* in neurons suggests that axon architecture is largely intact and disrupted sphingolipid homeostasis has context-specific effects. *asah-1* is expressed at low levels in embryonic ALM and PLM neurons, as are other members of the sphingolipid salvage pathway such as *sphk-1* (ref. 24). In contrast, expression of *sptl-1*, which encodes the rate-limiting enzyme of the sphingolipid de novo pathway, is undetected in these neurons. Perhaps *asah-1* overexpression in embryonic ALM and PLM neurons causes depletion of ceramide and/or other ceramide derivates that are not replenished by the de novo pathway. Such disrupted ceramide homeostasis in neurons could affect multiple signalling pathways controlling axon development. Future genetic and biochemical studies are required to dissect the roles of ASAH-1 and the other *C. elegans* ceramidase enzymes (ASAH-2 and W02F12.2) in the nervous system and more broadly.

We showed that maternal S1P transport to oocytes reduces the fragility of axons with impaired cytoskeletal function. We propose that elevated S1P in the oocyte/embryo influences the axonal environment to limit axon fragility. The important roles of sphingolipid homeostasis in plasma-membrane fluidity, lipid-raft integrity and molecular trafficking may be central to the protective role S1P plays in this context[43,44]. Disrupted sphingolipid metabolism has been implicated in neurodegenerative disorders including Alzheimer's disease, Parkinson's disease, Huntington's disease and multiple sclerosis[45]. The distinct autonomous and non-autonomous effects of sphingolipid homeostasis we revealed in *C. elegans* may also be relevant in these disease contexts.

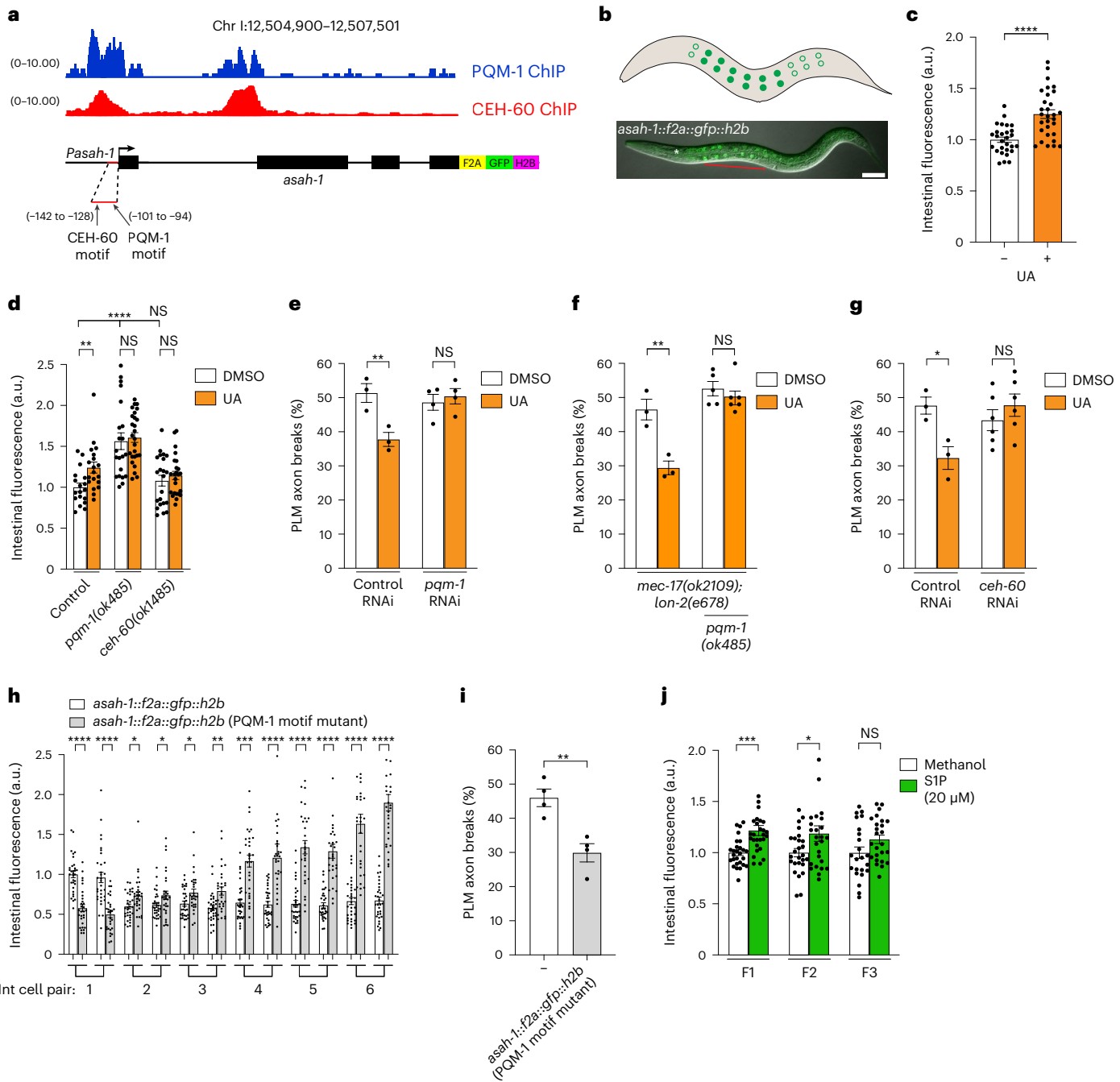

**Fig. 7 | ASAH-1-mediated neuroprotection requires PBX and MEIS-regulated transcription. a**, PQM-1 and CEH-60 transcription factor ChIP–seq peaks at the *asah-1* gene locus (top) aligned with a schematic of the *asah-1* endogenous reporter (bottom). The *asah-1* upstream region contains binding motifs for CEH-60 and PQM-1 (red line). The location of each binding motif (arrows) is indicated as the distance from the ATG (+1). **b**, Expression of *asah-1::f2a::gfp::h2b* in the intestine of an L4 larva (see Extended Data Fig. 7b for other developmental stages). Schematic (top) and overlay of Nomarski and fluorescence image (bottom). Filled circles indicate that expression was detected and open circles that weak/no expression was detected. Lateral view, anterior to the left. Pharynx is marked by a white asterisk; red line indicates bright *asah-1::f2a::gfp::h2b* expression. Scale bar, 25 μm. **c**, The expression of *asah-1::f2a::gfp::h2b* increases following exposure to UA (see Extended Data Fig. 8a for individual intestinal cell measurements). **d**, Loss of *pqm-1* or *ceh-60* prevents UA-induced induction of *asah-1::f2a::gfp::h2b* expression. **e–g**, Loss of *pqm-1* (via RNAi (**e**)) or gene deletion

(**f**)) or *ceh-60* (via RNAi (**g**)) suppresses UA-induced PLM neuroprotection in *mec-17(ok2109); lon-2(e678)* animals. **h**, Mutation of the putative PQM-1 binding motif in the *asah-1* promoter induces *asah-1::f2a::gfp::h2b* expression. Int cell pair 1–6, second–seventh pair, respectively, of intestinal cells from the pharynx. **i**, Mutation of the putative PQM-1 binding motif in the *asah-1* promoter reduces PLM axon breaks in *mec-17(ok2109); lon-2(e678)* animals. **j**, The progeny of P₀ mothers exposed to S1P for 16 h (L4 to young adult) increases *asah-1::f2a::gfp::h2b* expression for two generations (F₁ and F₂). **c–j**, $n = 28$ and 30 (**c**); $n = 18, 19, 21, 26, 22$ and 24 (**d**); $n = 74, 77, 99$ and 103 (**e**); $n = 73, 75, 122$ and 147 (**f**); $n = 65, 71, 150$ and 151 (**g**); $n = 32, 30, 32, 30, 32, 30, 32, 30, 32, 30, 32$ and 30 (**h**); $n = 100$ and 104 (**i**); and $n = 27, 28, 26, 26, 23$ and 25 (**j**) hermaphrodite animals per condition (left to right). *P* values were determined using an ANOVA (**d–g,j**) or unpaired Student's *t*-test (**c,h,i**). ****$P \le 0.0001$; ***$P \le 0.001$; **$P \le 0.01$; *$P \le 0.05$; and NS, not significant. Error bars indicate the s.e.m.; a.u., arbitrary units. Source data are provided.

**Sphingolipid regulation of intergenerational inheritance**

The neuroprotective effect of intestinal *asah-1* is dependent on the expression of SphK-1, an enzyme that phosphorylates sphingosine to generate S1P. In support of this, maternally supplied S1P is transferred from the intestine to oocytes to protect axons. We showed that maternal S1P is probably transported within low-density yolk lipoproteins to promote neuroprotection. Essential roles for S1P and SphK in protecting vertebrate oocytes from radiation-induced apoptosis and the development of embryonic cardiovascular tissue have also been revealed[46–49]. In vertebrates, plasma lipoproteins also carry S1P, highlighting a conserved mode of carriage[50].

How does S1P act intergenerationally? Intestinal *asah-1* expression is elevated for two generations following exposure of reproductive stage hermaphrodites to S1P. This suggests that mechanisms in intestinal cells detect changes in sphingolipid levels, including S1P, that alter sphingolipid-pathway gene expression (for example, *asah-1*) in response. Examples of such sphingolipid autoregulatory networks have previously been identified in cellular and zebrafish models where sphingolipid metabolic enzymes are induced by accumulation of their sphingolipid substrate[51,52]. Increased *asah-1* in the intestines of $F_1$ animals could generate high S1P that would be transferred to the $F_2$ generation to maintain the intergenerational effect. We suggest that a threshold of intestinal S1P must be maintained to initiate *asah-1* upregulation and this threshold may be lost over time—hence, a reason for the intergenerational rather than transgenerational effect. At the level of *asah-1* gene regulation, subtle changes in the subcellular localization or promoter occupancy of *asah-1* regulators, including the transcription factors PQM-1 and CEH-60, could be responsible. PQM-1 nuclear localization is indeed responsive to certain stressors and perturbed insulin growth factor signalling[37].

Periods of prenatal and postnatal interactions enable mammalian mothers to influence offspring[53]. Alterations in the reproductive behaviour of offspring permits the transmission of these effects to succeeding generations[53]. We show that the transmission of a lipid metabolite in the yolk can intergenerationally protect specific neurons in *C. elegans*. We suggest that our findings point to a broader underlying role for metabolism and metabolic gene expression in controlling intergenerational effects across species.

## Online content

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

## Methods

### *C. elegans* strains and culture

*C. elegans* hermaphrodites were maintained according to standard protocols at 20 °C on nematode growth medium (NGM) plates with *Escherichia coli* (OP50) bacteria as a food source. The wild-type strain used was Bristol, N2. Mutant strains were backcrossed to N2 a minimum of three times and the animals were well-fed for at least two generations before performing experiments. A list of the strains and transgenes used in this study is provided in Supplementary Table 2.

### Natural compound screen

All compounds were solubilized in DMSO and diluted to 50 μM by mixing with NGM agar. For the control plates, animals were grown with 1% DMSO. None of the compounds had an overt effect on OP50 growth. $P_0$ animals were grown from L4 stage on compound plates and their $F_1$ progeny were scored as 3-d-old adults. All plates were prepared fresh before each experiment and used within 2 d. To reduce metabolic breakdown of the compound by bacteria, NGM plates (containing compound or DMSO) were seeded with 150 μl heat-killed OP50 (65 °C for 30 min) and dried overnight at 20 °C. A 5-μl volume of live OP50 bacteria (1/10 dilution with Luria broth (LB)) was added to the dead bacteria to enable worm growth.

### *C. elegans* expression constructs and transgenic strain generation

Reporter gene constructs were generated by PCR amplifying DNA elements and cloning these into Fire vectors. The constructs were injected into young adult hermaphrodites using standard methods.

***Pasah-1::nls::gfp* reporter construct.** A 2,044 bp sequence upstream of the *asah-1* start codon was amplified from *C. elegans* genomic DNA using forward (5′-GAAATGAAATAAGCTTAAAGAGA GAATAATAATCGAGTGAG-3′) and reverse (5′-GCAGGCATGCAAGC TTCTTTCTTGACTAGCTCTGAATAGTG-3′) oligonucleotides incorporating an HindIII restriction site. The HindIII-digested promoter fragment was ligated into HindIII-digested pPD95.67 vector, resulting in *Pasah-1::NLS::gfp*. *Pasah-1::NLS::gfp* was injected at 25 ng μl⁻¹, with 5 ng μl⁻¹ *Pmyo-2::mCherry* vector and 120 ng μl⁻¹ bacterial DNA. The resultant *Pasah-1::NLS::gfp* extrachromosomal array line was integrated using ultraviolet light irradiation. The integrated *Pasah-1::NLS::gfp* (*rp165*) was backcrossed to N2 six times before analysis.

***Pges-1::asah-1* overexpression construct.** Complementary DNA to *asah-1* (1,183 bp) was amplified from an N2 cDNA library using forward (5′-AGGACCCTTGGCTAGCATGCTCCGAGAATTGTCGG-3′) and reverse (5′-GATATCAATACCATGGCTACCATGGATAGCATTCTCCCGG-3′) oligonucleotides, and then cloned into a *Pges-1::egl-8* vector (digested with NheI and NcoI to remove the *egl-8* sequence) using an In-Fusion HD cloning kit (Takara Bio).

***Pdpy-7::asah-1* overexpression construct.** Complementary DNA to *asah-1* (1,183 bp) was amplified from an N2 cDNA library using forward (5′-CGACTCTAGAGGATCCATGCTCCGAGAATTGTCGG-3′) and reverse (5′-CGCTCAGTTGGAATTCCTACCATGGATAGCATTCTCCCGG -3′) oligonucleotides, and then cloned into a *Pdpy-7::nls-dsRed2* vector (digested with BamHI and EcoRI to remove the *nls-dsRed2* sequence) using an In-Fusion HD cloning kit (Takara Bio).

***Pmyo-3::asah-1* overexpression construct.** Complementary DNA to *asah-1* (1,183 bp) was amplified from an N2 cDNA library using forward (5′-AGGACCCTTGGCTAGCATGCTCCGAGAATTGTCGG-3′) and reverse (5′-GATATCAATACCATGGCTACCATGGATAGCATTCTCCCGG-3′) oligonucleotides, and then cloned into the pPD95.86 vector using an In-Fusion HD cloning kit (Takara Bio) after linearizing the vector with NheI and NcoI.

***Prab-3::asah-1* overexpression construct.** Complementary DNA to *asah-1* (1,183 bp) was amplified from an N2 cDNA library using forward (5′-TGGCTAGCGTCGACGGTACCATGCTCCGAGAATTGTCGG-3′) and reverse (5′-GATATCAATACCATGGCTACCATGGATAGCATTCTCCC GG-3′) oligonucleotides, and then cloned into a *Prab-3::nfya-1* vector (digested with KpnI and NcoI to remove the *nfya-1* sequence) using an In-Fusion HD cloning kit (Takara Bio).

***Pmec-4::asah-1* overexpression construct.** Complementary DNA to *asah-1* (1,183 bp) was amplified from an N2 cDNA library using forward (5′-AGGACCCTTGGCTAGATGCTCCGAGAATTGTCGG-3′) and reverse (5′-TACCGTCGACGCTAGCTACCATGGATAGCATTCTCCCGG-3′) oligonucleotides, and then cloned into the *pSM::Pmec-4::unc-54 3′UTR* vector using an In-Fusion HD cloning kit (Takara Bio) after linearizing the vector with NheI.

***Pges-1::sptl-1* overexpression construct.** Complementary DNA to *sptl-1* (1,378 bp) was amplified from an N2 cDNA library using forward (5′-AGGACCCTTGGCTAGCATGGGATTTCTACCAGATTCGTGG-3′) and reverse (5′-GATATCAATACCATGTTAGAATTTATGAGCAACAACTCGG-3′) oligonucleotides, and then cloned into a *Pges-1::egl-8* vector (digested with NheI and NcoI to remove the *egl-8* sequence) using an In-Fusion HD cloning kit (Takara Bio).

***Pges-1::sphk-1* overexpression construct.** Complementary DNA to *sphk-1* (1,423 bp) was amplified from an N2 cDNA library using forward (5′-AGGACCCTTGGCTAGCATGTTCATAGTAGTGGTAAC-3′) and reverse (5′-GATATCAATACCATGGCTAGGCAGTTGATGAGAAAA-3′) oligonucleotides, and then cloned into a *Pges-1::egl-8* vector (digested with NheI and NcoI to remove the *egl-8* sequence) using an In-Fusion HD cloning kit (Takara Bio).

### UA treatment

A 10 mg ml⁻¹ stock solution of UA (Sigma-Aldrich, 89797) dissolved in DMSO was prepared. UA (or DMSO for the control plates) was added to a 3.5-cm petri dish before the addition of NGM agar. All plates were freshly prepared before each experiment and used within 2 d. To reduce metabolic breakdown of UA by bacteria, 150 μl of heat-killed OP50 (65 °C for 30 min) was seeded on NGM plates (containing UA or DMSO) and dried overnight at 20 °C. A 5-μl volume of live OP50 (1/10 dilution with LB broth) was added to the dead bacteria to enable worm growth.

### S1P treatment

A 2 mM stock solution of S1P (Sigma-Aldrich, 73914) dissolved in methanol was prepared. NGM agar was poured into a 3.5 cm petri dish, to which S1P (or methanol for the control plates) was added and mixed. All plates were freshly prepared before each experiment and used within 2 d. To reduce metabolic breakdown of S1P by bacteria, 150 μl of heat-killed OP50 (65 °C for 30 min) was seeded and spread on NGM plates (containing S1P or methanol) and dried overnight at 20 °C. A 5-μl volume of live OP50 (1/10 dilution with LB broth) was added to the dead bacteria to enable worm growth.

### Ceramide treatment

Stock solutions of 0.5 mg ml⁻¹ C20 (d18:1/20:0; Sigma-Aldrich, 860520P) and C22 (d18:1/22:0; Sigma-Aldrich, 860501P) dissolved in ethanol were prepared. Before seeding with bacteria, 60 μg ceramide was added to the surface of NGM plates. To reduce metabolic breakdown of ceramides by bacteria, 150 μl of heat-killed OP50 (65 °C for 30 min) was seeded and spread on NGM plates (containing ceramide or ethanol as the control) and dried overnight at 20 °C. A 5-μl volume of live OP50 (1/10 dilution with LB broth) was added to the dead bacteria to enable worm growth.

## S1P–fluorescein application

A stock solution of 1 mM S1P–fluorescein (Echelon, S-200F) dissolved in methanol was prepared. To reduce metabolic breakdown of S1P–fluorescein by bacteria, 150 μl of heat-killed (65 °C for 30 min) OP50 was mixed with S1P–fluorescein or methanol (as the control), seeded on NGM plates and used immediately once it dried.

## Microtubule drug treatment

A stock solution of 20 mM colchicine (Sigma-Aldrich, C9754) dissolved in DMSO was prepared. Colchicine (or DMSO for the control plates) was added to a 3.5 cm petri dish before the addition of NGM agar. All plates were freshly prepared before each experiment and used within 2 d. To reduce metabolic breakdown of colchicine by bacteria, 150 μl of heat-killed OP50 (65 °C for 30 min) was seeded on NGM plates (containing colchicine or DMSO) and dried overnight at 20 °C. A 5-μl volume of live OP50 (1/10 dilution with LB broth) was added to the dead bacteria to enable worm growth.

## Intergenerational experiments

$P_0$ animals were treated from mid-L4 to young adult (16 h) with UA or S1P (DMSO and methanol were used the respective controls) and the worms were then transferred to untreated plates for 3 h to lay semi-synchronized eggs. The mothers were removed after laying and the progeny ($F_1$ generation) were cultured on untreated plates. When the $F_1$ animals reached the 1-d-old adult stage, they were transferred to new untreated plates for 3 h to lay eggs ($F_2$ generation) and this was repeated for the progeny of these animals ($F_3$ generation). For each generation, PLM fragility defects were scored in 3-d-old adults.

## CRISPR–Cas9 genome editing

**Endogenous asah-1::f2a::gfp::h2b reporter.** Cas9 protein, *trans*-activating CRISPR RNA and a crispr RNA (crRNA; TGGTAGATTG-GATTCAACAT) from IDT were used to generate *asah-1(rp176[asah-1::f2a::gfp::h2b])*. A double-stranded DNA hybrid donor sequence of *f2a::gfp::h2b* was amplified by PCR, as described previously[54]. To facilitate the scoring of *asah-1* expression, a nuclear-localized *gfp* was inserted downstream of the *asah-1* coding region, separated by an F2A sequence that splits the endogenous ASAH-1 and GFP–H2B proteins.

**Mutagenesis of the *asah-1* promoter.** For the mutagenesis of *asah-1* promoter motifs, crRNAs and single-stranded oligonucleotide (ssODN) donors were used as follows.

CEH-60 motif: crRNA, 5′-AAAACCGGTGTGATTGATGA-3′; and ssODN, (5′-TTGTAAATTATGCTCCATAAACCAAAAACCGGTGTGATT GATGACATTTTCATGAGGGAAATAAATTAAA-3′). Mutation of the motif introduced a NotI restriction site (GCGGCCGC) to facilitate genotyping.

PQM-1 motif site: crRNA, 5′-AAGTGATAAGGAGTAAAGTG-3′; and ssODN, 5′-ACATTTTCATGAGGGAAATAAATTAAAAAGTGATAAGGAG TAAAGTGTGGCCACGTGTTTTCCGCAAAAA-3′. Mutation of the motif introduced a BamHI restriction site (GGATCC) to facilitate genotyping.

## RNAi experiments

HT115(DE3) *E. coli* bacteria expressing RNAi plasmids for specific genes or empty vector (L4440) were cultured at 37 °C for 16 h in LB broth containing 50 μg ml$^{-1}$ ampicillin. Saturated RNAi cultures were plated on RNAi plates and dried for 24 h. L4 hermaphrodites were plated on RNAi plates and incubated for one generation at 20 °C, unless otherwise stated. Due to the potency of *rme-2* RNAi in preventing progeny survival, we diluted the *rme-2* RNAi bacteria to 10% with 90% L4440 RNAi bacteria. Dilution of *rme-2* RNAi bacteria to 10% was shown previously to reduce embryonic yolk levels[55].

## Microscopy

Worms were anaesthetized using 20 mM sodium azide for neuroanatomical scoring and levamisole (0.1 ng μl$^{-1}$) for fluorescence measurements. The worms were mounted on 5% agarose pads on glass slides. Images were acquired using a Zeiss Axio Imager M2 and Zen software tools. Figures were prepared using ImageJ and Adobe Photoshop and Illustrator.

## Phenotypic analysis

Phenotypic analysis of PLM neurons was performed on L1 larvae and 3-d-old adults where indicated. Axonal breaks were identified only when the distal fragment was visible. PVQ and D-type motor neuron analyses were performed on L1 and L4 larvae, respectively. All phenotypic scoring was performed at least in triplicate on independent days.

## Axon transport analysis

UNC-104::GFP and mCherry::RAB-3 pooling at the distal end of PLM neurites was scored in 3-d-old adults. All phenotypic scoring was performed at least in triplicate on independent days.

## Analysis of *asah-1* expression

Expression of *Pasah-1::gfp* and *asah-1::f2a::gfp::h2b* was imaged and measured in six pairs of intestinal nuclei from the second anterior pair backwards (the expression in the first pair of anterior nuclei was weak and inconsistent). Texas red fluorescence in each nucleus was used as background.

## Sequence alignment

Sequences of the *asah-1* promoter in nematode species were extracted from WormBase (http://www.wormbase.org) and aligned using Clustal Omega[56].

## RNA sequencing

N2 gravid hermaphrodites were bleached with hypochlorite solution and eggs were allowed to hatch overnight in M9 buffer. The resultant synchronized L1 larvae were fed with OP50 and incubated for 34 h until they reached the L3 stage. These worms were washed from the plates with M9 buffer and exposed to UA (50 μM), DMSO (vehicle) or no treatment for 12 h. After treatment, the worms (now L4 larvae) were harvested and washed three times with M9 buffer to remove bacteria before being resuspended in TRIzol and flash-frozen in liquid nitrogen. Total RNA was extracted by phase separation with chloroform, followed by the RNeasy mini kit (Qiagen) protocol. RNase-free DNase was applied to the RNA samples to remove DNA contamination. Library preparation and sequencing on Illumina HiSeq instruments were performed by the Monash Micromon Sequencing Facility and data analysis was performed by the Monash Bioinformatics Platform. The transcriptome of five independent replicates for each group was analysed. The RNA-sequencing data were converted to a per-gene read-count matrix using RNAsik pipeline v1.5.0. Briefly, single-end reads were mapped to the *C. elegans* genome (WBcel235) using STAR[57] in splice-aware mode, duplicates were marked using Picard Markduplicates and reads were assigned to gene models (exonic only) from the WBcel235 annotation using featureCounts v1.5.2 (ref. 58). Genes were only considered 'testable' if they had at least ten read counts in a sample and at least 1.0 counts-per-million in at least three samples. After applying this filter for testable genes, 14,037 protein-coding genes were retained. Genes that were differentially expressed between the treatment and control groups were identified using the EdgeR-quasi workflow[59]. The RNA-sequencing data are available at the Gene Expression Omnibus under the accession number GSE214425.

## cDNA preparation and qPCR

Total RNA was isolated as described for RNA sequencing but with independent samples. An iScript kit (BioRad) was used for cDNA synthesis. PowerUp SYBR Green (Thermo Fisher) was used for the qPCR. The cDNA samples from three biological replicates were run in triplicate. The relative transcript levels were normalized to the housekeeping

gene *cdc-42*. The following oligonucleotide pairs were used: *asah-1* F, 5′-CTACTGTTCCTTGTGTCGGA-3′ and *asah-1* R, 5′-GTTGGAGGCA TTTGTTGCA-3′; and *cdc-42* F, 5′-GACAATTACGCCGTCACAG-3′ and *cdc-42* R, 5′-CGTAATCTTCCTGTCCAGCA-3′.

### ChIP–seq data analysis
Published ChIP–seq datasets for CEH-60 (ref. 36) and PQM-1 (refs. 37,60) were used to identify binding peaks in the *asah-1* promoter using IGV[61].

### Locomotion analysis
Worm locomotion was analysed at room temperature on day 1 and 3 adults using WormLab imaging (MBF Bioscience). NGM plates used for tracking were freshly seeded with 20 µl OP50 10 min before each experiment. Animals treated with UA or DMSO were randomly picked for tracking. Six animals were placed on each plate and their tracks were recorded for 6 min. To allow worms to adapt to the new conditions, the initial minute of tracking was not included for analysis. The average speed was calculated from the total distance moved (forwards and backwards) during the next 5 min.

### Body-length analysis
The body length of worms was analysed at room temperature on day 1 and 3 adults using WormLab imaging (MBF Bioscience). NGM plates used for body-length measurements were freshly seeded with 20 µl OP50 10 min before each experiment. Animals treated with UA or DMSO were randomly picked for measurement. Six animals were placed on each plate for imaging and their body length was automatically meas-ured by the WormLab software.

### Statistics and reproducibility
Parameters such as the *n* value, mean ± s.e.m. and significant *P* values are reported in the figures, figure legends and Source Data. Significance was defined as $P < 0.05$.

Statistical analysis was performed in GraphPad Prism (v9.5) using an ANOVA, followed by Dunnett's or Tukey's multiple comparison tests for three or more conditions. An unpaired Student's *t*-test was performed if the comparison was for two conditions. All experiments were performed with at least three independent biological replicates, with similar results (unless specified otherwise in the figure legends); the investigator was blinded to the genotype. No statistical method was used to pre-determine sample size. No data were excluded from analysis. The experiments were not randomized.

### Reporting summary
Further information on research design is available in the Nature Port-folio Reporting Summary linked to this article.

### Data availability
The RNA-sequencing data obtained in this work have been depos-ited at NCBI Gene Expression Omnibus under the accession number GSE214425. Previously published ChIP–seq datasets for the transcrip-tion factors CEH-60 and PQM-1 that were re-analysed here and used to identify binding peaks in the *asah-1* promoter using IGV are available under the accession codes GSE112981 and GSE25811. All data are avail-able in the main text or the extended data. Source data are provided with this paper.

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

## Acknowledgements
We thank B. Neumann and the members of the Pocock laboratory for their comments on the manuscript. We thank the Monash Micromon Sequencing Facility and Monash Bioinformatics Platform for the RNA sequencing and data analysis. Some strains used in this study were provided by the *Caenorhabditis* Genetics Center, which is funded by NIH Office of Research and Infrastructure Programs (P40 OD10440) and the National BioResource Project (NBRP) Japan. This work was supported by the following funding: National Health and Medical Research Council grant nos. GNT1105374, GNT1137645 and GNT2000766 (R.P.); and Apex Biotech Research Pty Ltd donation 281589068 (Z.X.).

## Author contributions
Conceptualization: W.W., Z.X. and R.P. Methodology: W.W., J.L. and R.P. Investigation: W.W., T.S., X.C., Q.F., R.C. and R.P. Visualization: W.W., X.C., R.C. and R.P. Funding acquisition: Z.X. and R.P. Project administration: W.W. and R.P. Supervision: J.L., Z.X. and R.P. Writing of the original draft: R.P. Writing, review and editing: W.W., T.S., X.C., Q.F., R.C., J.L., Z.X. and R.P.

## Competing interests
The authors declare no competing interests.

## Additional information
**Extended data** is available for this paper at https://doi.org/10.1038/s41556-023-01195-9.

**Correspondence and requests for materials** should be addressed to Roger Pocock.

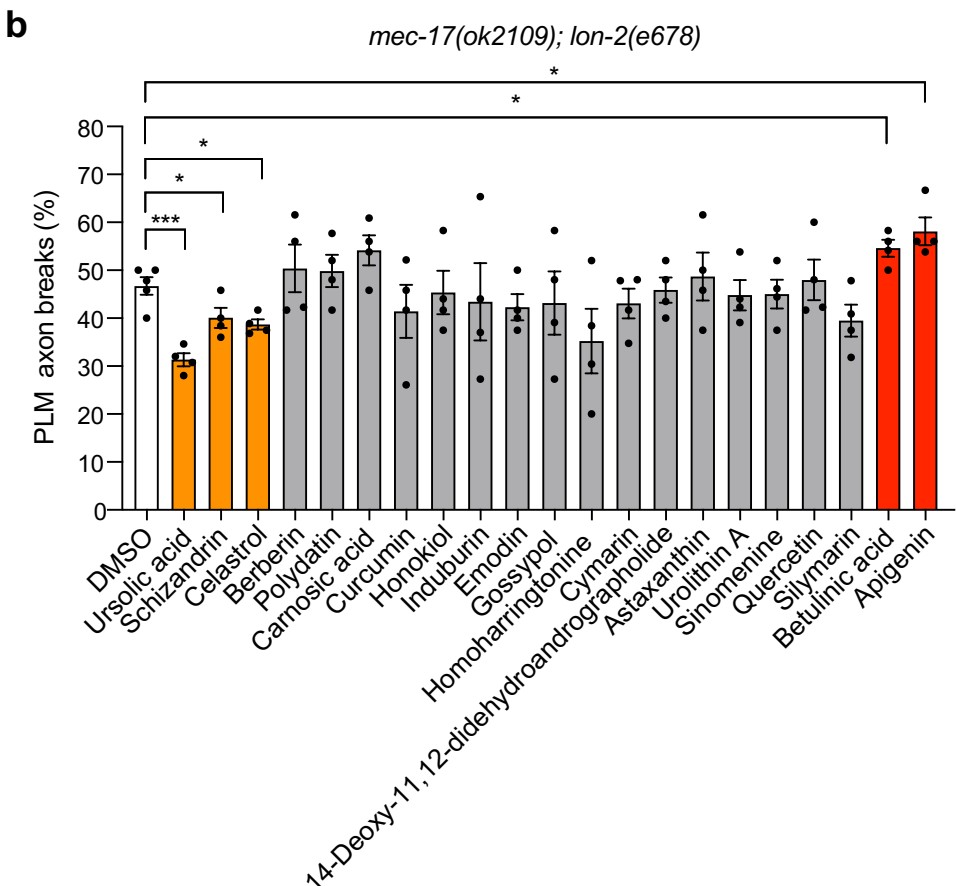

**Extended Data Fig. 1 | Natural-product screen for regulators of axon fragility.**
**a**, Natural-product exposure timeline. Upper schematic shows the stages of
*C. elegans* development relevant to continuous natural-product exposure in the
$P_0$ and $F_1$ generations. **b**, Quantification of PLM axon breaks in *mec-17(ok2109);
lon-2(e678)* animals following continuous exposure ($P_0$ to L4 larva to $F_1$ to adult)
of a natural-product library at 50 μM. Orange bars. reduced PLM breaks; red bars,
increased PLM breaks; grey bars, not significant. Quantification of PLM axon
breaks shown in **b**, $n$ = 122, 102, 100, 85, 101, 100, 98, 94, 97, 100, 97, 94, 99, 95, 98,
98, 100, 102, 100, 91, 99 and 100 hermaphrodite animals per condition (left to
right). *P* values were determined using an unpaired *t*-test. ***$P \le 0.001$; *$P \le 0.05$.
Error bars indicate the s.e.m. Source data are provided.

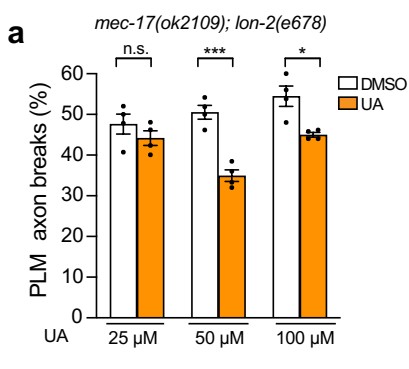

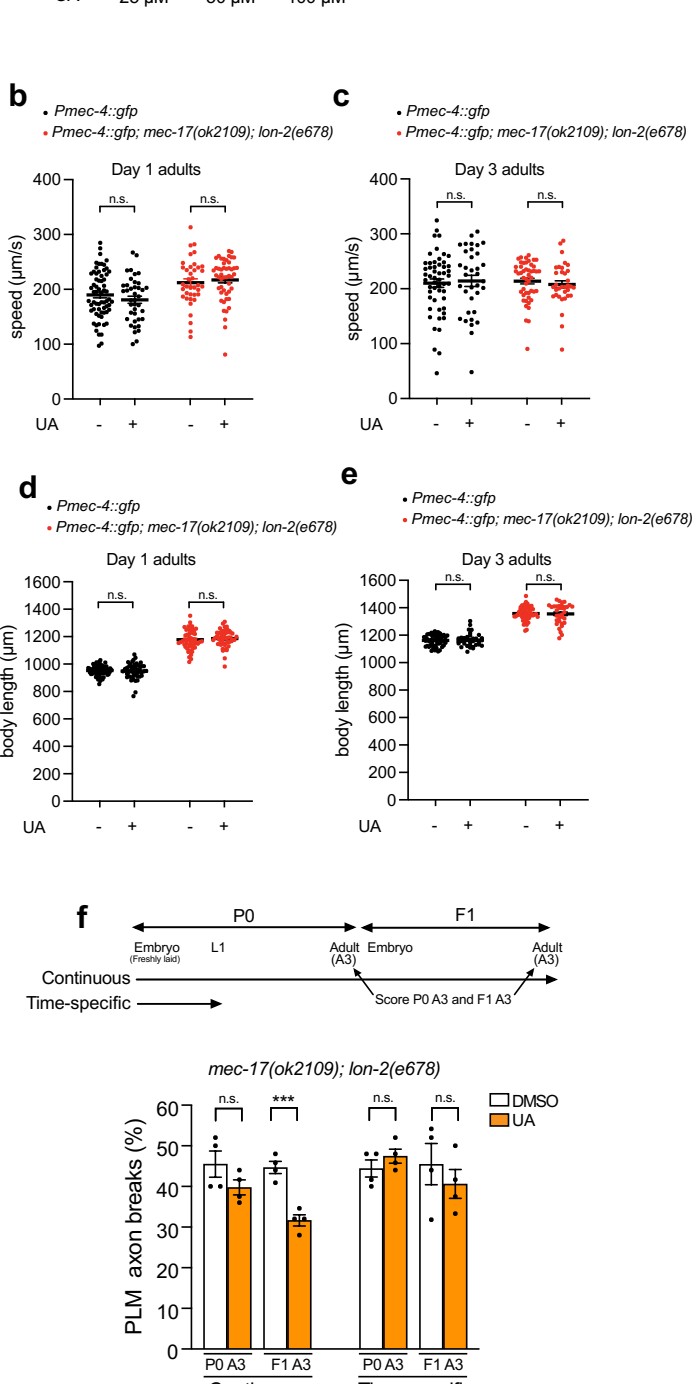

**Extended Data Fig. 2 | See next page for caption.**

**Extended Data Fig. 2 | Effect of UA on axon fragility and worm growth/behaviour. a**, Continuous exposure ($P_0$ to L4 larva to $F_1$ to adult) of UA at 50 or 100 μM, but not 25 μM, rescues PLM axon breaks in *mec-17(ok2109); lon-2(e678)* animals. **b,c**, Quantification of locomotion speed of *zdIs5* and *zdIs5; mec-17(ok2109); lon-2(e678)* $F_1$ animals on day 1 (**b**) and 3 (**c**) of adulthood treated with DMSO (control) or UA ($P_0$ mothers treated for 16 h, L4 to young adult). Data presented as speed in μm s$^{-1}$ (points). **d,e**, Quantification of body length of *zdIs5* and *zdIs5; mec-17(ok2109); lon-2(e678)* $F_1$ animals at day 1 (**d**) and 3 (**e**) of adulthood treated with DMSO (control) or UA ($P_0$ mothers treated for 16 h, L4 to young adult). Data presented as individual lengths in μm (points). **f**, Exposure of UA (50 μM) from $P_0$ embryo to $F_1$ 3-d-old adults ($F_1$ to A3, continuous) reduces PLM axon breaks in *mec-17(ok2109); lon-2(e678)* $F_1$ animals.

In contrast, UA exposure (50 μM) from $P_0$ embryo to $P_0$ 3-d-old adults ($P_0$ to A3, continuous), or only during embryogenesis ($P_0$ to A3 and $F_1$ to A3, time-specific) does not reduce PLM axon breaks in *mec-17(ok2109); lon-2(e678)* $F_1$ animals. Quantification of PLM axon breaks shown in **a**, $n$ = 101, 102, 101, 103, 101 and 100; **f**, $n$ = 99, 98, 96, 101, 99, 99, 96 and 97 hermaphrodite animals per condition (left to right). Quantification of locomotion speed shown in **b**, $n$ = 65 and 40; **c**, $n$ = 49 and 38 hermaphrodite animals per condition (left to right). Quantification of body length shown in **d**, $n$ = 66 and 46; **e**, $n$ = 50 and 50 hermaphrodite animals per condition (left to right). $P$ values were calculated using a one-way analysis of variance (ANOVA; **b**–**e**) or unpaired $t$-test (**a,f**). \*\*\*$P \le 0.001$; \*$P \le 0.05$; and NS, not significant. Error bars indicate the s.e.m. Source data are provided.

**a**

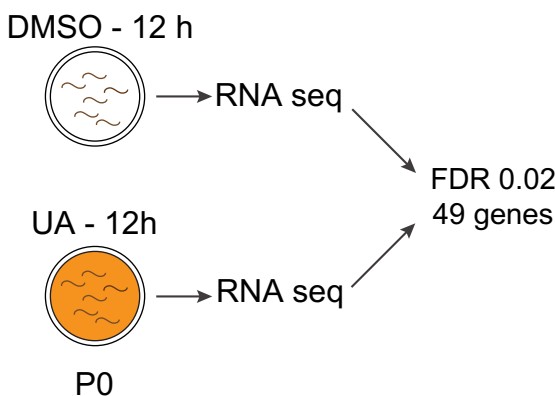

**b**

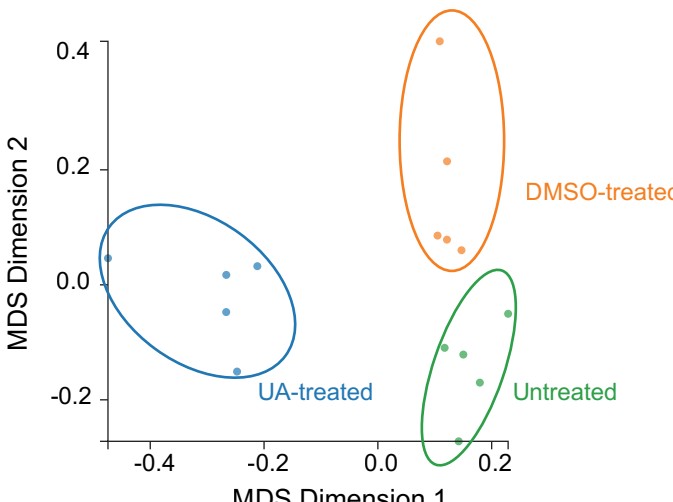

**Extended Data Fig. 3 | UA RNA-sequencing analysis. a**, Schematic of transcriptome analysis following UA exposure. Total RNA isolated from L4 larvae exposed to DMSO or UA for 12 h; 49 genes were regulated by UA exposure (false discovery rate, 0.02). **b**, Multidimensional scale plot (MDS) of untreated, DMSO-treated (control) and UA-treated RNA-sequencing samples. Source numerical data are provided in Supplementary Table 1.

## *Pasah-1::gfp* expression

3-fold embryo 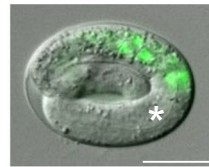

L1 at hatching 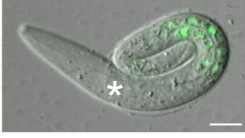

Late L1 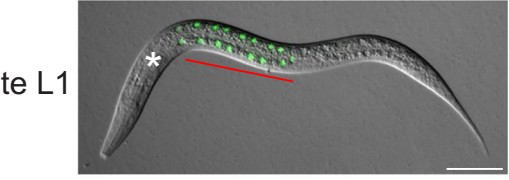

L4 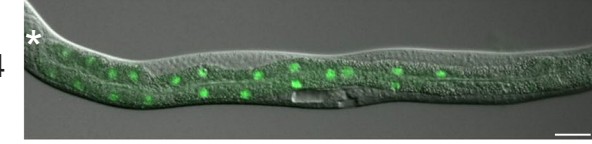

Young adult 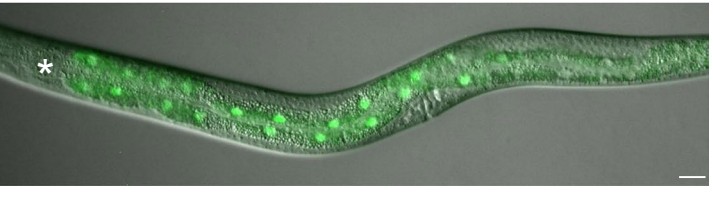

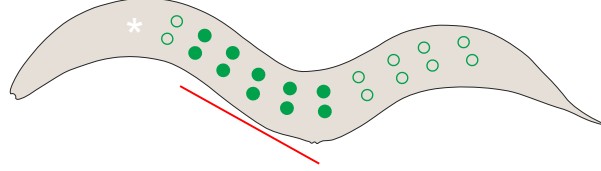

**Extended Data Fig. 4 | *asah-1* transcriptional reporter expression analysis.** The *Pasah-1::NLS::gfp* transcriptional reporter (*rpIs165*) is expressed in the intestine from the threefold stage of embryogenesis throughout larval development and into adulthood. A schematic of a young adult is shown below the fluorescence micrograph. Note that *Pasah-1::NLS::gfp* expression is brighter in the anterior intestine (red line). Anterior to the left in larval image. Pharynx marked by a white asterisk. Scale bars, 20 μm.

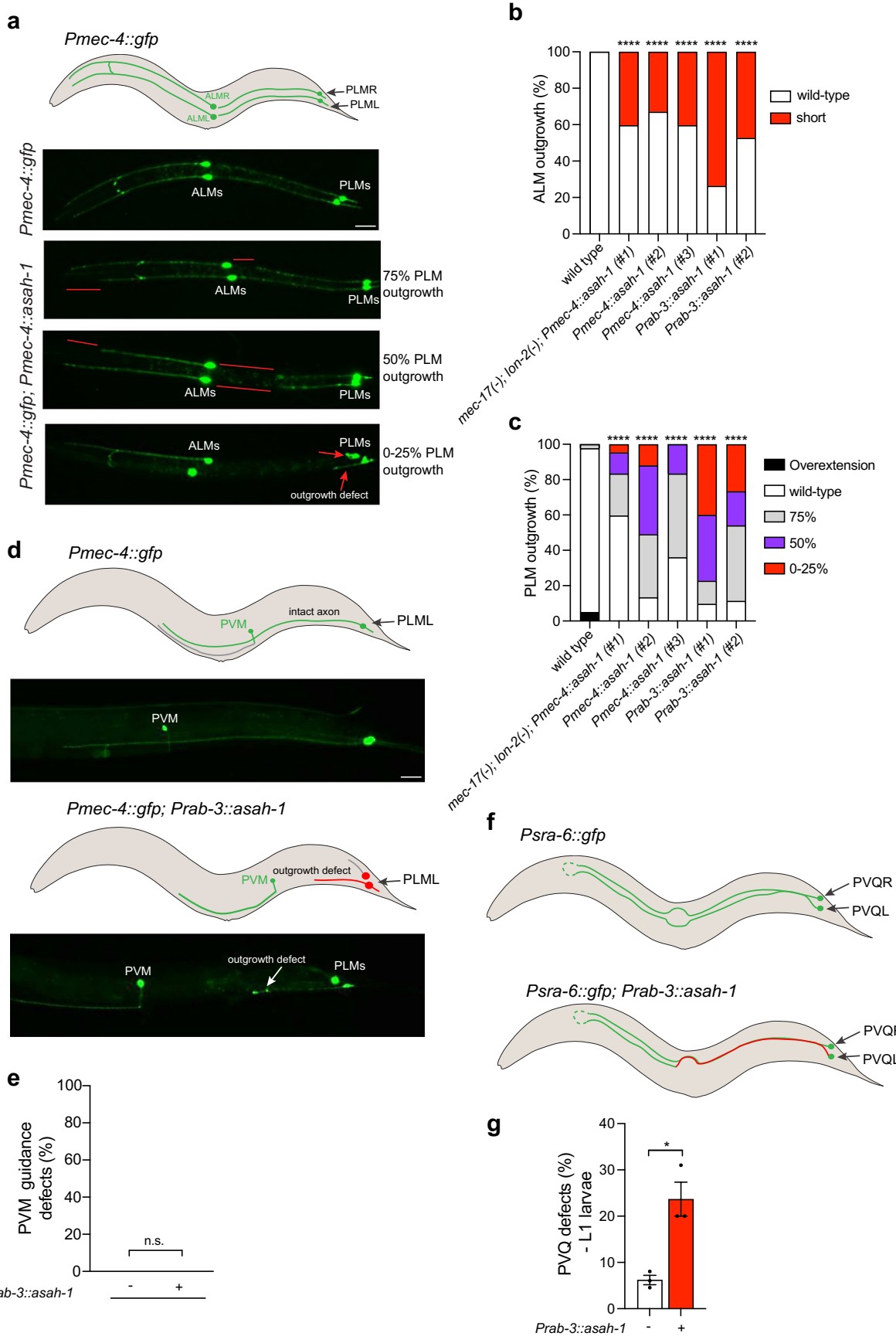

**Extended Data Fig. 5 | See next page for caption.**

**Extended Data Fig. 5 | *asah-1* overexpression analysis. a**, Schematic and fluorescence micrographs of ALM and PLM anatomy in L1 larval wild-type (top micrograph) and *Pmec-4::asah-1*-overexpressing (bottom micrographs) animals expressing the *Pmec-4::gfp* transgene (*zdIs5*). Outgrowth defects are marked with red lines and arrows. Scale bar, 20 μm. **b,c**, Quantification of ALM (**b**) and PLM (**c**) outgrowth defects in wild-type, *mec-17(ok2109); lon-2(e678); Pmec-4::asah-1, Pmec-4::asah-1* and *Prab-3::asah-1* animals. The ALM and PLM outgrowth defects correspond to the micrographs in **a**. **d,e**, Schematics and fluorescence micrographs (**d**) as well as quantification (**e**) of PVM axon guidance in L4 wild-type and *Prab-3::asah-1*-overexpressing animals expressing the *Pmec-4::gfp* transgene (*zdIs5*). Scale bar, 25 μm. **f,g**, Schematics (**f**) and quantification (**g**) of PVQ axon guidance in L1 wild-type and *Prab-3::asah-1*-overexpressing animals expressing the *Psra-6::gfp* transgene (*oyIs14*). Quantification of ALM outgrowth defects shown in **b**, *n* = 60, 42, 39, 46, 48 and 54 hermaphrodite animals per condition (left to right). Quantification of PLM outgrowth defects shown in **c**, *n* = 60, 42, 39, 46, 48 and 54 hermaphrodite animals per condition (left to right). Quantification of PVM guidance defects shown in **e**, *n* = 45 and 75 hermaphrodite animals per condition (left to right). Quantification of PVQ axon defects shown in **g**, *n* = 65 and 73 hermaphrodite animals per condition (left to right). *P* values were determined using a two-way analysis of variance (ANOVA; **b,c**) or unpaired *t*-test (**e,g**). ****$P \leq 0.0001$; *$P \leq 0.05$; and NS, not significant. Error bars indicate the s.e.m. Source data are provided.

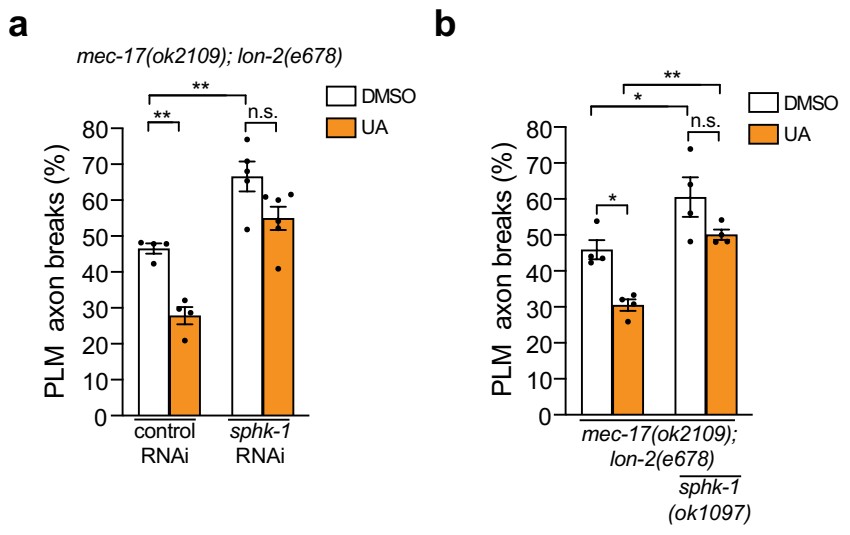

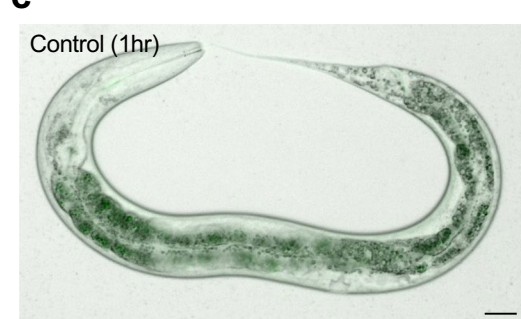

**c**

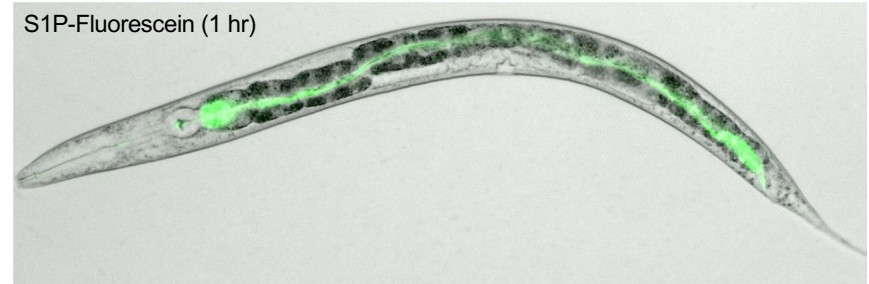

**Extended Data Fig. 6 | Analysis of *sphk-1* function and S1P–fluorescein detection. a,b,** Loss of *sphk-1* via RNAi (**a**) and gene deletion (*sphk-1-(ok1097)* deletion mutant; **b**) enhances PLM axon breaks in *mec-17(ok2109); lon-2(e678)* animals and prevents the protective effect of UA. **c,** Brightfield and fluorescence image overlay of L4 hermaphrodites exposed to control (methanol) or S1P–fluorescein for 1 h. S1P–fluorescein was detected in the intestinal lumen. Scale bar, 20 μm. Quantification of PLM axon breaks shown in **a**, *n* = 99, 97, 128 and 138; **b**, *n* = 100, 105, 100 and 100 hermaphrodite animals per condition (left to right). *P* values were determined using a one-way analysis of variance (ANOVA). **P ≤ 0.01; *P ≤ 0.05; and NS, not significant. Error bars indicate the s.e.m. Source data are provided.

## a

**asah-1 promoter (200bp)**

TTTAATGAATTCTTCTCTGAAAACGCTTGTAAATTATGCTCCATAAACCAAAAACC
GGTGTGATTGATGACATTTTCATGAGGGAAATAAATTAAAAAGTGATAAGGAGTA
AAGTGTGGCCACGTGTTTTCCGCAAAAAGTATTGATAAGTCGCCTCGCGGAGC
ACACGCTTTGCCACTATTCAGAGCTAGTCAAGAAAG

TGTGATTGATGACA = CEH-60 motif
TGATAAG = PQM-1 motif

**Mutations introduced to the asah-1 promoter using CRISPR-Cas9**

TTTAATGAATTCTTCTCTGAAAACGCTTGTAAATTATGCTCCATAAACCAAAAACC
GGT<mark>GCGGCCGC</mark>TGACATTTTCATGAGGGAAATAAATTAAAAAGTG<mark>GGATCC</mark>GA
GTAAAGTGTGGCCACGTGTTTTCCGCAAAAAGTATTCCATGGTCGCCTCGCGG
AGCACACGCTTTGCCACTATTCAGAGCTAGTCAAGAAAG

## b

*asah-1::f2a::gfp::h2b* 3-fold embryo

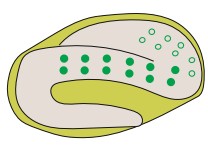
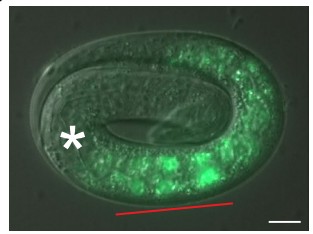

*asah-1::f2a::gfp::h2b* L1 larva

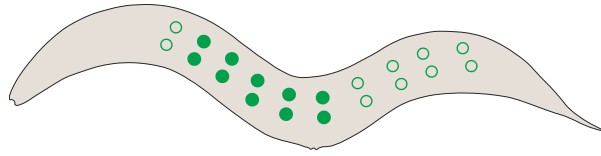
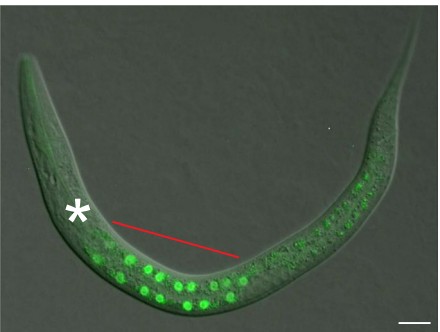

**Extended Data Fig. 7 | Analysis of the *asah-1* endogenous reporter.**
**a**, Nucleotide sequence 200 bp upstream of the *asah-1* start codon. Upper sequence shows the wild-type sequence highlighting putative transcription factor-binding motifs for CEH-60 (red) and PQM-1 (blue, motif). The bottom sequence shows the independent mutations introduced to delete each putative transcription factor-binding motif (yellow highlights) that introduced the following restriction enzyme recognition sequences: CEH-60 motif (NotI) and PQM-1 motif (BamHI). **b**, GFP–H2B expression from the *asah-1::f2a::gfp::h2b* endogenous reporter (*rpIs176*) was detected in the intestine from late embryogenesis (threefold stage) throughout larval development and into adulthood. A schematic of each animal is shown next to the fluorescence micrographs. Note that *Pasah-1::NLS::gfp* expression is consistently brighter in the anterior intestine (red line). Anterior to the left in the worm image. Pharynx marked by a white asterisk. Scale bar, 20 μm.

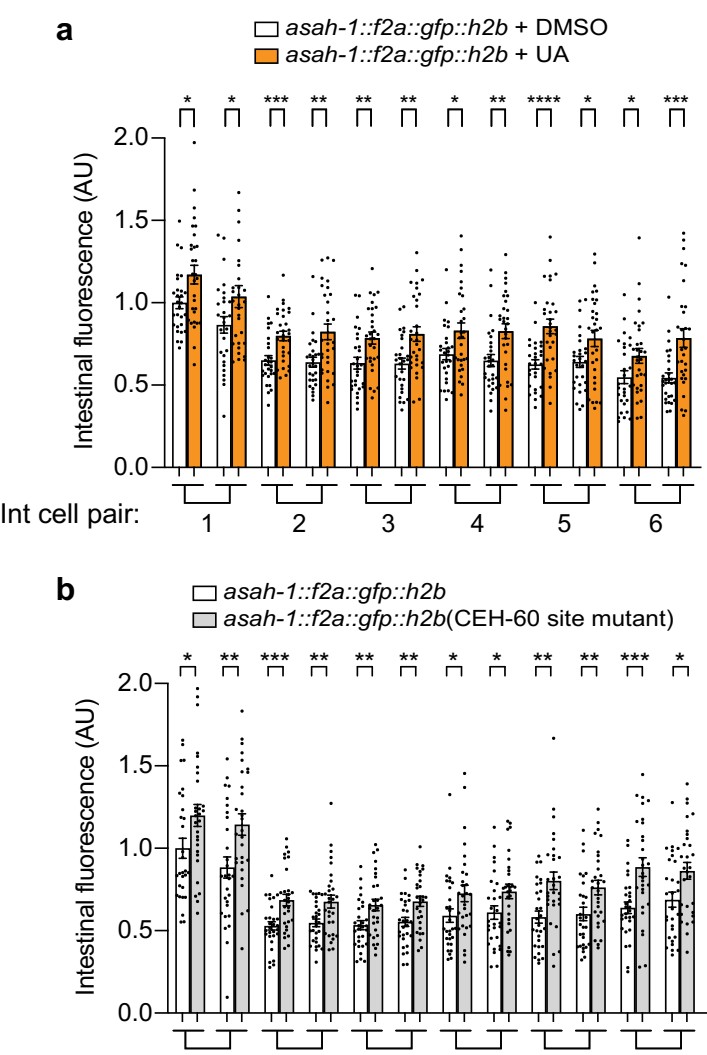

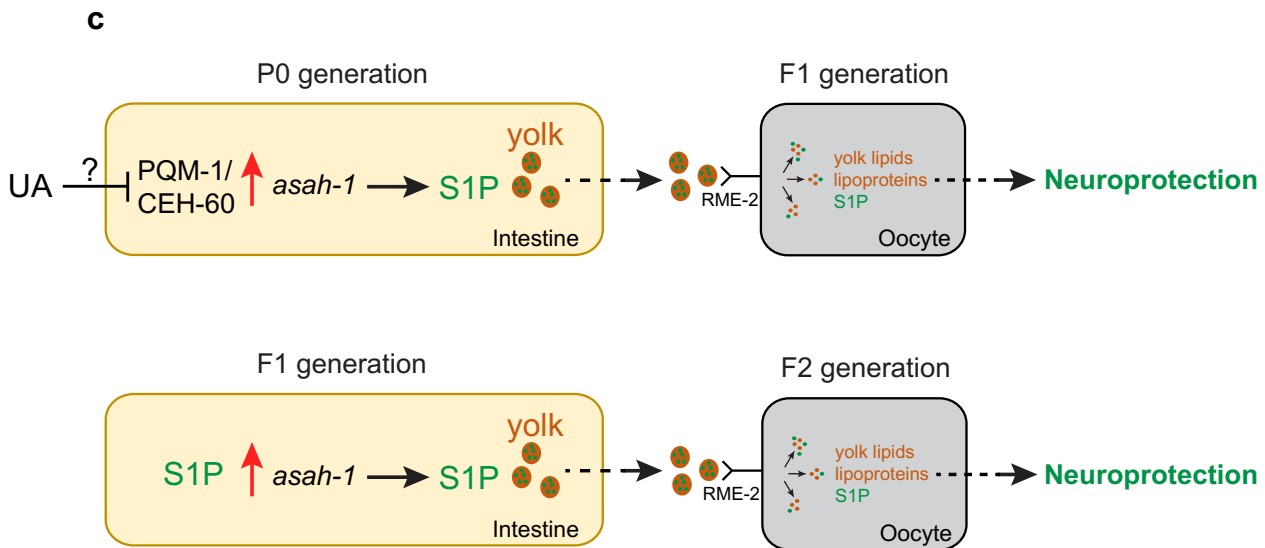

**Extended Data Fig. 8 | See next page for caption.**

**Extended Data Fig. 8 | Regulation of endogenous *asah-1* expression. a**, GFP–H2B expression from the *asah-1::f2a::gfp::h2b* endogenous reporter increases when animals are exposed to UA. Quantification of nuclear fluorescence from each pair of intestinal cells per worm is shown (Int cell pair 1 = second pair of intestinal cells from the pharynx). **b**, Mutation of putative CEH-60 binding motif in the *asah-1* promoter induces GFP–H2B expression from the *asah-1* endogenous reporter. Quantification of nuclear fluorescence from each pair of intestinal cells per worm is shown (Int cell pair 1, second pair of intestinal cells from the pharynx). **c**, Proposed model of UA/S1P intergenerational neuroprotection. UA exposure of $P_0$ animals (L4 to young adult) elevates intestinal *asah-1* expression (probably through inhibition of PQM-1/CEH-60-directed transcriptional

repression) and increases S1P levels. S1P is transferred within intestinally derived yolk to oocytes via the RME-2 receptor and promotes axon health in these animals (top). In $F_1$ progeny, elevated S1P is detected in the intestine, which causes *asah-1* expression to increase and generate additional S1P that is transmitted, via the yolk, to $F_2$ oocytes and promotes axon health in these animals (bottom). UA/S1P neuroprotection is not observed in $F_3$ progeny, probably due to reduced S1P levels that are below the threshold required for neuroprotection. Quantification of *asah-1::f2a::gfp::h2b* fluorescence shown in **a**, $n = 28, 29, 28, 29, 28, 29, 28, 29, 28, 29, 28$ and 29; **b**, $n = 29$ animals per condition (left to right). $P$ values were determined using and unpaired $t$-test. ****$P \leq 0.0001$; ***$P \leq 0.001$; **$P \leq 0.01$; *$P \leq 0.05$. Error bars indicate the s.e.m. Source data are provided.

# Reporting Summary

## Statistics

For all statistical analyses, confirm that the following items are present in the figure legend, table legend, main text, or Methods section.

| n/a | Confirmed | |
|---|---|---|
| ☐ | ☒ | The exact sample size (*n*) for each experimental group/condition, given as a discrete number and unit of measurement |
| ☒ | ☐ | A statement on whether measurements were taken from distinct samples or whether the same sample was measured repeatedly |
| ☐ | ☒ | The statistical test(s) used AND whether they are one- or two-sided<br>*Only common tests should be described solely by name; describe more complex techniques in the Methods section.* |
| ☒ | ☐ | A description of all covariates tested |
| ☐ | ☒ | A description of any assumptions or corrections, such as tests of normality and adjustment for multiple comparisons |
| ☐ | ☒ | A full description of the statistical parameters including central tendency (e.g. means) or other basic estimates (e.g. regression coefficient) AND variation (e.g. standard deviation) or associated estimates of uncertainty (e.g. confidence intervals) |
| ☐ | ☒ | For null hypothesis testing, the test statistic (e.g. *F*, *t*, *r*) with confidence intervals, effect sizes, degrees of freedom and *P* value noted<br>*Give P values as exact values whenever suitable.* |
| ☒ | ☐ | For Bayesian analysis, information on the choice of priors and Markov chain Monte Carlo settings |
| ☒ | ☐ | For hierarchical and complex designs, identification of the appropriate level for tests and full reporting of outcomes |
| ☒ | ☐ | Estimates of effect sizes (e.g. Cohen's *d*, Pearson's *r*), indicating how they were calculated |

*Our web collection on statistics for biologists contains articles on many of the points above.*

## Software and code

Policy information about availability of computer code

| Data collection | Wormbase (www.wormbase.org, release WS285), Zeiss Zen (v2.0) software, Fiji software (v2.0.0), Wormlab (WL-15) software (MBF Bioscience). |
|---|---|
| Data analysis | Data analysis was performed using:<br>GraphPad Prism v9.5<br>Integrated Genomics Viewer v2.12.2<br>featureCounts v1.5.2<br>RNAsik pipeline v1.5.0<br>STAR v2.5.2b<br>Picard Markduplicates v2.18.0<br>EdgeR-quasi v3.26.8 |

For manuscripts utilizing custom algorithms or software that are central to the research but not yet described in published literature, software must be made available to editors and reviewers. We strongly encourage code deposition in a community repository (e.g. GitHub). See the Nature Portfolio guidelines for submitting code & software for further information.

# Data

Policy information about availability of data

All manuscripts must include a data availability statement. This statement should provide the following information, where applicable:
- Accession codes, unique identifiers, or web links for publicly available datasets
- A description of any restrictions on data availability
- For clinical datasets or third party data, please ensure that the statement adheres to our policy

RNA sequencing data obtained in this work has been deposited at NCBI under the GEO accession number GSE214425. Previously published ChIP-seq datasets for CEH-60 and PQM-1 TFs that were re-analyzed here and were used to identify binding peaks in the asah-1 promoter using IGV are available under accession codes GSE112981 and GSE25811. All data are available in the main text or the extended data. Source data have been provided in Source Data.

# Human research participants

Policy information about studies involving human research participants and Sex and Gender in Research.

| | |
|---|---|
| Reporting on sex and gender | N/A |
| Population characteristics | N/A |
| Recruitment | N/A |
| Ethics oversight | N/A |

Note that full information on the approval of the study protocol must also be provided in the manuscript.

# Field-specific reporting

Please select the one below that is the best fit for your research. If you are not sure, read the appropriate sections before making your selection.

☒ Life sciences          ☐ Behavioural & social sciences          ☐ Ecological, evolutionary & environmental sciences

For a reference copy of the document with all sections, see nature.com/documents/nr-reporting-summary-flat.pdf

# Life sciences study design

All studies must disclose on these points even when the disclosure is negative.

| | |
|---|---|
| Sample size | Sample size was based on previous experiments and prior literature using similar experimental paradigms:<br><br>Reference for nervous system analysis:<br>Neumann, B. & Hilliard, M.A. Loss of MEC-17 leads to microtubule instability and axonal degeneration. Cell Rep 6, 93-103 (2014). doi: 10.1016/j.celrep.2013.12.004.<br>Norgaard, S., Deng, S., Cao, W. & Pocock, R. Distinct CED-10/Rac1 domains confer context-specific functions in development. PLoS Genet 14, e1007670 (2018). doi: 10.1371/journal.pgen.1007670<br><br>Reference for expression analysis:<br>Rasoul Godini and Roger Pocock. Characterization of the Doublesex/MAB-3 transcription factor DMD-9 in Caenorhabditis elegans. G3 (Bethesda). 2023 Feb 9;13(2):jkac305. doi: 10.1093/g3journal/jkac305. |
| Data exclusions | No data was excluded. |
| Replication | All data were generated over multiple days and in triplicate, with all replication attempts successful. |
| Randomization | Experiments were not randomized and controlling covariates was not necessary. In all experiments, control vs. experimental samples were analyzed in parallel. For functional analysis, groups were allocated based on genotype, confirmed by PCR or based on fluorescent marker expression for rescue strains. |
| Blinding | Study was blinded unless mutant phenotypes were easily observable. |

# Reporting for specific materials, systems and methods

We require information from authors about some types of materials, experimental systems and methods used in many studies. Here, indicate whether each material, system or method listed is relevant to your study. If you are not sure if a list item applies to your research, read the appropriate section before selecting a response.

## Materials & experimental systems

| n/a | Involved in the study |
|-----|------------------------|
| ☒ | ☐ Antibodies |
| ☒ | ☐ Eukaryotic cell lines |
| ☒ | ☐ Palaeontology and archaeology |
| ☐ | ☒ Animals and other organisms |
| ☒ | ☐ Clinical data |
| ☒ | ☐ Dual use research of concern |

## Methods

| n/a | Involved in the study |
|-----|------------------------|
| ☒ | ☐ ChIP-seq |
| ☒ | ☐ Flow cytometry |
| ☒ | ☐ MRI-based neuroimaging |

## Animals and other research organisms

Policy information about studies involving animals; ARRIVE guidelines recommended for reporting animal research, and Sex and Gender in Research

| | |
|---|---|
| Laboratory animals | Caenorhabditis elegans hermaphrodites were used - see attached strain list for full details. The stage of animals imaged/analysed were L4 larvae, 1-day, 2-day or 3-day adults (depending on the experiment). Specific ages are described in the methods and specific Figure legends. Hermaphrodites and males were used to generate new strains via crossing. |
| Wild animals | The study did not involve wild animals. |
| Reporting on sex | Hermaphrodites were solely used in our analysis. This is except for analysis of embryos and early larval stages where the sex could not be categorically determined, however, males are 0.1% of the population so extremely rare. |
| Field-collected samples | This study did not involve samples collected from the field. |
| Ethics oversight | No ethical guidance was required as this was a C. elegans study. |

Note that full information on the approval of the study protocol must also be provided in the manuscript.

