## [Peer Review File · Nature Cell Biology]

Peer Review Information

Journal: Nature Cell Biology

Manuscript Title: An intestinal sphingolipid confers intergenerational neuroprotection

Corresponding author name(s): Professor Roger Pocock

Editorial Notes:

Reviewer Comments & Decisions:

Decision Letter, initial version:
--

Dear Professor Pocock,

Thank you for submitting your manuscript, "Intergenerational Neuroprotection by an Intestinal Sphingolipid in *Caenorhabditis elegans*", to Nature Cell Biology. It has now been seen by 4 referees, who are experts in S1P signaling (Referee #1); *C. elegans*, metabolism, longevity (Referee #2); *C. elegans*, neurodegeneration (Referee #3); and epigenetics, intergenerational inheritance, *C. elegans* (Referee #4). As you will see from their comments (attached below), they found this work of potential interest but have raised substantial concerns, which in our view would need to be addressed with considerable revisions before we can consider publication in Nature Cell Biology.

As per our standard editorial process, we have discussed the referee reports in detail within the

editorial team, including the chief editor, to identify key referee points that should be addressed with priority, as opposed to requests that are overruled as being beyond the scope of the current study. To guide the scope of the revisions, I have listed these points below. Our typical revision period is six months, and we are committed to providing a fair and constructive peer-review process, so please feel free to contact me if you would like to discuss any of the referee comments further or if you anticipate any issues or delays addressing the reviews.

I should stress that the referees' concerns regarding the strength of the analyses and their relevance to axon health would need to be addressed with experiments and data, and reconsideration of the study for this journal and re-engagement of referees would depend on strength of these revisions. We feel that efforts are needed in revision to address the following over-arching points:

1- three reviewers stressed that the claims of protective effects in the context of *neurodegeneration* are not warranted based on the use of the single model of axon breaks (the published mec-17 and lon-1 mutant model). To maintain these claims and/or adjust them to the protection against axon breaks, we agree with the referees that testing at least one other model of axon break/neurodegeneration and another class of neuron will be very important:

Rev#2 point #2

Rev#3 point #1 (following "There are two main conceptual issues")

Rev#4 point #7

2- the reviewers also feel that the claims of a neuroprotective effects are not explained at the level of neurons and axons. We agree that efforts along the lines they suggest to bolster the claims of the roles of UA and S1P are needed:

Rev#2 points #1-2

Rev#3 major point #2

3- as highlighted by several reviewers, more information is needed about the screen performed to focus on UA. Please share the screen (Rev#3 other important points, #1; Rev#4 #2) and please note that, per journal policy, should the paper be accepted ultimately, you would be asked to deposit the results of the screen prior to acceptance.

4- While the genetic analyses are of high quality, the reviewers all asked for further analyses to support the model and resolve potential inconsistencies:

Rev#1 point F #2

Rev#3 points #5, #6, #8

Rev#4 points #1, #3, #4, #5, #6

5- Rev#1 brings up important considerations when it comes to ceramide/sphingolipid metabolic pathways that should be considered and at a minimum discussed: points F #1, F #4, F #6. Please also follow up on their questions to further track S1P fate via C17-S1P and LC/MS analyses in the eggs (F #8).

6- All other referee concerns pertaining to strengthening existing data, providing controls, methodological details, clarifications and textual changes, should also be addressed.

7- Finally, please pay close attention to our guidelines on statistical and methodological reporting (listed below) as failure to do so may delay the reconsideration of the revised manuscript. In particular

please provide:

We would be happy to consider a revised manuscript that would satisfactorily address these points, unless a similar paper is published elsewhere, or is accepted for publication in Nature Cell Biology in the meantime.

In contrast, although we agree with Referee #3 that analyzing the contribution of gamete type ["7. While the authors show that gametes exposed to UA drive the main effect, it would be interesting to determine which gametes are sufficient to cause it or if both are necessary. This could be tested by exposing male nematodes to UA in the sperm producing stage and then crossing these animals to untreated hermaphrodites to then analyse neuroprotection in their cross-progeny."] would provide valuable insights, we consider this point to be beyond the scope of the present study. Thus, addressing it experimentally will not be necessary for reconsideration of the manuscript at this journal.

- ensure that it conforms to our format instructions and publication policies (see below and <https://www.nature.com/nature/for-authors>).

- provide a point-by-point rebuttal to the full referee reports verbatim, as provided at the end of this letter.

- provide the completed Reporting Summary (found here <https://www.nature.com/documents/nr-reporting-summary.pdf>). This is essential for reconsideration of the manuscript will be available to editors and referees in the event of peer review. For more information see <http://www.nature.com/authors/policies/availability.html> or contact me.

When submitting the revised version of your manuscript, please pay close attention to our [href="https://www.nature.com/nature-portfolio/editorial-policies/image-integrity">Digital Image Integrity Guidelines](https://www.nature.com/nature-portfolio/editorial-policies/image-integrity). and to the following points below:

Nature Cell Biology is committed to improving transparency in authorship. As part of our efforts in this direction, we are now requesting that all authors identified as 'corresponding author' on published papers create and link their Open Researcher and Contributor Identifier (ORCID) with their account on the Manuscript Tracking System (MTS), prior to acceptance. ORCID helps the scientific community achieve unambiguous attribution of all scholarly contributions. You can create and link your ORCID from the home page of the MTS by clicking on 'Modify my Springer Nature account'. For more information please visit www.springernature.com/orcid.

This journal strongly supports public availability of data. Please place the data used in your paper into a public data repository, or alternatively, present the data as Supplementary Information. If data can only be shared on request, please explain why in your Data Availability Statement, and also in the correspondence with your editor. Please note that for some data types, deposition in a public repository is mandatory - more information on our data deposition policies and available repositories appears below.

[Redacted]

We hope that you will find our referees' comments and editorial guidance helpful. Please do not hesitate to contact me if there is anything you would like to discuss. Thank you again for considering NCB for your work.

Best wishes,

Melina

Melina Casadio, PhD
Senior Editor, Nature Cell Biology
ORCID ID: <https://orcid.org/0000-0003-2389-2243>

Reviewers' Comments:

Reviewer #1:
Remarks to the Author:

A. Wang et al provide evidence to support the concept that in *C. elegans*, intestinal expression and function of acid ceramides (Asah1) leads to the production of S1P, transfer to the yolk/ oocytes via the lipoprotein receptor rme-2 which leads to the ability to resist axonal breaks in the worm neurons brought about by microtubule defects. They suggest that maternal induction of Asah-1 in the intestinal cells is critical for this event, which lasts for 2-3 generations. They suggest that intergenerational transfer of maternal sphingolipid signal to the yolk is perhaps explained by epigenetic changes in the embryo which are sensitive to sphingolipid auto regulatory networks.

B. the work is highly novel and potentially very significant.

C. It should be mentioned that I am not well versed in the nuances of *C. elegans* genetics. My expertise is in sphingolipids and cell biology. Nevertheless, my opinion is that the sphingolipid work and cell biological mechanisms that the authors are examining are overall fine. There are some minor issues that the authors should address which I will detail below. However, the overall message of the paper is OK.

D. OK

E. I feel the authors' data are quite solid. the experiments seem to have appropriate controls and appear to support the conclusions.

F. Suggestions for improvement:

1. Ceramidase enzymes have several isoforms - acid, alkaline and neutral. I assume that the authors have not characterized the biochemistry of *C. elegans* ceramidase(s). It would be helpful to mention the possible existence of isoforms by gene sequence homology at the very least. In mammals, intestinal enzyme is alkaline and acid ceramides is either on the cell surface or lysosomes. Mammalian enzyme is also known for its important function in early embryonic development. These issues should be considered and/or discussed.

2. Although the authors rule out the requirements for RNAi/ Ago pathway in intergenerational inheritance, their conclusion of "epigenetic" mechanisms is speculative. This should be further explained. Is it possible that S1P action on novel GPCRs or other downstream effectors is involved in the oocyte and embryonic development? The authors might want to consider that GPR3 which is critical for mammalian oocyte meiosis has been suggested to be a sensor of S1P even though this has not been fully vetted since this GPCR is also a constitutive activator of Gs and adenylate cyclase.

3. The list of genes regulated by UA (transcriptome data mentioned) should be provided with proper statistics on induction/ repression properties in the data supplement.

4. The authors oversimplify their consideration of ceramide/ sphingolipid pathways. They fail to consider sphingomyelinase pathway which can generate ceramide. This should be considered. Do homologues/ orthologues of SPMD enzymes exist in the worm?

5. Page 10, penultimate line - "complexity" is not a proper word in this context - do the authors mean "levels"?

6. In Sptl-1 overexpression experiments, is this enzyme a single subunit enzyme in the worm? in mammals, at least 3 subunits are needed.

7. Figure 4D is not described in the results.

8. How does 20 μM S1P supplied exogenously taken up by the worms and transported to the oocytes/yolk? Is it metabolized? The authors can use C17-S1P (odd chain) and follow by LC/MS in the eggs.

9. The idea of sphingolipid auto regulatory networks should be discussed more fully. Please refer to the following examples in the literature (PMID: 28956531, PMID: 27226645).

10. S1P and Sphk enzymes were shown to be very important in early embryonic events in vertebrates. In addition, S1P protects mammalian oocytes and sperm from stress-induced death. This should be discussed.

Reviewer #2:

Remarks to the Author:

This manuscript reports that a natural product, ursolic acid (UA) attenuates PLM axon degeneration across two generations. The authors have characterized the mechanism underlying this protection. UA induces the expression of *asah-1*, encoding acid ceramidase that hydrolyzes ceramide into fatty acid and sphingosine, in the intestine, and intestinal overexpression of *asah-1* is sufficient to reduce PLM axon degeneration. Through genetic analyses of the metabolic steps upstream and downstream of *ASAH-1*, the authors have narrowed down sphingosine-1-phosphate (S1P) responsible for the transgenerational protective effect. Using fluorescence-labeled S1P and genetic analysis, the authors revealed that S1P transports from the intestine to oocytes and acts through RME-2 receptor in the germline to mediate the protective effect. Furthermore, the authors discovered that two transcription factors respond to UA and negatively regulate *asah-1* expression. Together, the manuscript discovers an interesting protective effect of environmental UA and endogenous S1P on PLM axon degeneration through generations and delineates the molecular mechanism mediating this protective effect.

Two major weaknesses to me are:

1. It remains unclear what happens to neurons in F1 and F2 generations. Is the protective effect due to higher S1P levels in neurons? With a higher concentration of the S1P treatment, is it possible to see a protective effect in P0? The authors show that the ectopic overexpression of *asah-1* in neurons leads to axon outgrowth in P0. How about neuronal overexpression of *sphk-1* or overexpression of both *asah-1* and *sphk-1* at the same time? Will this lead to a protective effect in P0? If the protective effect is not due to the S1P increase in neurons, what could be other possible mechanisms?

2. Whether this protective effect is specific to PLM adult-onset axon degeneration. *MEC-17* loss in the *lon-2* mutant background leads to axon degeneration likely due to defective axonal transport. Does the UA and the S1P treatments and/or the *asah-1* and *sphk-1* overexpression protect against disrupted mitochondrial distribution and/or synaptic protein localization. The current manuscript is solely dependent on axon breaks counting. Cellular characterization will help strengthen the claim on the improvement of neural health.

One minor point:

Line 203, "*asah-1* overexpression disrupts axon outgrowth of the ALM and PLM axons" should be

“asah-1 overexpression causes axon outgrowth of the ALM and PLM axons” based on the context.

Reviewer #3:

Remarks to the Author:

In this study by Wang et al entitled “Intergenerational neuroprotection by an intestinal sphingolipid in *Caenorhabditis elegans*” the authors show an intergenerational neuroprotective effect of acute ursolic acid (UA) intake supplementation in *C. elegans*. The authors demonstrate that 16 h exposure to UA during the gamete producing stage is sufficient to prevent axonal breaks for 2 subsequent generations in mutant animals lacking the MEC-17/TAT1 α -tubulin acetyltransferase sensitized by an increased body length due to a lon-1 mutation. They reveal that such neuroprotection is caused by the upregulation of ASAH-1 expression in the intestine, which leads to maternal transfer of sphingosine-1-phosphate (S1P) to the offspring through the yolk receptor RME-2. Importantly, they show that S1P supplementation is sufficient to induce neuroprotection and that the level of this lipid regulates this effect. In addition to the above, the authors show that two transcription factors PQM-1 and CEH-60 are involved in asah-1 regulation, and that two enzymes, SPTL-1 and SPHK-1, functioning within the S1P pathway are also implicated.

This manuscript covers an important biological question of how parental diet affects the health of the progeny. However, despite the large set of data presented and genetic pathway identified, there are some key elements of concern that need to be addressed to support their conclusions. There are two main conceptual issues:

1. The use of the term ‘neuroprotection’ evokes an idea of protection from many neurodegenerative insults. However, the entire manuscript is based on a single model of axonal degeneration that requires two mutations, *mec-17* and *lon-1*, to be present and that have been described to affect microtubule stability (Neumann & Hilliard, Cell Rep 2014). The paper does not address any role of the UA (and the related identified pathway) on these specific genes. An alternative axonal degenerative model should be tested for the author to conclude a broader neuroprotective function of UA. For example, have the authors tried the effect of UA on axonal degeneration upon injury, or upon treatment with microtubule drugs (colchicine and paclitaxel), or on another independent axonal degeneration mutations? Moreover, did they test if the neuroprotection extends to other classes of neurons, and thus independent on the *mec-17* effect, which appears selective for the *mec* neurons?
2. There is no indication on how the UA treatment affects the axon. Did they test whether UA alters microtubules dynamics in these axons, or axonal transport, or both? While an effect of UA is not questioned, it was not shown, mechanistically, how this relates to the axon (i.e. what this pathway does to the axonal structure to prevent breaks).

Considering these two issues combined, the paper in its currently format falls short in providing a molecular mechanism of neuroprotection; further experiments as suggested above will allow to validate, strengthen, and broaden their conclusions.

Other important points

1. The authors indicated on line 111 that they performed a screen of natural products from which UA was identified. They should provide the list of those natural products that have been screened and

their effect (presumably negative).

2. The authors should expand the UA concentration curve as the effect seems to reduce at 100 μ M. Given this is still a relative low concentration, testing one or two additional higher points will allow them to determine if the effect is peaking at 50 μ M or increases at higher concentrations.

3. The statement on line 118 – 119 “for one generation” is not clear. If I understand correctly, these animals have been exposed for two generations, the P0 and the F1, with the F1 adults being tested.

4. In line 120, the reference to Figure 1D does not indicate when the animals were incubated, and that the population scored is F1.

5. Related to the point above, in Figure 1E it seems important to determine whether the incubation with UA of the P0 at early embryo (before L4) is sufficient to induce the effect in the next generation (i.e. the UA incubation described in “c” should be tested in the P0). This is also important for the statement in lines 131-133 in which only incubation of L4-adults gives a phenotype.

6. In relation to *sptl-1*, did they test the function in different tissues? Did they test whether *sptl-1* and *sphk-1* RNAi also block the UA protection? These experiments should be performed.

7. While the authors show that gametes exposed to UA drive the main effect, it would be interesting to determine which gametes are sufficient to cause it or if both are necessary. This could be tested by exposing male nematodes to UA in the sperm producing stage and then crossing these animals to untreated hermaphrodites to then analyse neuroprotection in their cross-progeny.

8. The authors hypothesize that the intergenerational effect is not due to maternal small RNA transfer by investigating HRDE-1-deficient animals; however, it is not clear whether this molecule is the sole nematode controller of sRNA transport. If that is not the case, then sRNA transfer cannot be ruled out as a mechanism as other molecules with redundant effect might compensate for the lack of HRDE-1.

9. In relation to the *ASAH-1* expression, the authors did not test whether expression in PLM neurons can rescue the defect of the *asah-1* RNAi. Do *asah-1* mutants phenocopy the *asah-1* RNAi-treated animals? This would be an important confirmation and, more importantly, will allow them to conclusively test cell-autonomy in a precise way, without over-expressing it in a wild-type background. At this regard, in the experiments in which they observed an over-expression defect in mechanosensory neurons when *ASAH-1* is driven by the *mec-4* promoter, what concentration and how many transgenic lines were studied? Have they tried the single copy expression to ensure the minimal amount is present?

10. It is very curious that *ASAH-1* overexpression specifically, affects PLM/ALM morphology and development but not movement controlling neurons. Moreover, the main neuroprotective effect observed in this study is also in PLM/ALM. It would be interesting to discuss why this is the case, especially considering that PLM/ALM are microtubule-rich neurons. Maybe *ASAH-1* interferes with microtubule stabilization? *ASAH-1* effect could be tested in conjunction with microtubules (de)stabilizing drugs, as also indicated in the two major points above.

11. To complement the *SPTL-1* analysis, it would be interesting to try the conditional degradation of this molecule within the intestines through the AID system to overcome lethality.

Text suggestions

1. Missing reference on line 218 after "...ASAH-1 substrate..."
2. The statement starting on line 360 about outgrowth/guidance contradicts the previous statement where it is said that "...caused developmental axon outgrowth defects in the ALM and PLM neurons.". Perhaps the authors meant to say that outgrowth/guidance of movement controlling neurons is not affected?
3. Figure 4 legend, please replace 'worm' with nematode in "... (worm and mammalian ortholog) ..."
4. Figure 6 legend, please replace 'worm' with animal in "...of intestinal cells per worm..."
5. Figure 1D and 2D, n number is not indicated in the legend.

Reviewer #4:

Remarks to the Author:

Here, Wang et al., identify ursolic acid as a neuroprotectant model that prevents axon damage in *mec-17; lon-2* *C. elegans* mutants which are particularly prone to axon damage due to mechanical stress that occurs during movement. They show that ursolic acid induces the expression of the acid ceramidase *asah-1*, which in turn promotes sphingosine-1-phosphate production. Finally, the authors show that S1P can be provisioned into oocytes in a way that promotes neuroprotection in F1 and F2 descendants of animals exposed to ursolic acid. This study would be among the first to demonstrate the provisioning of a metabolite into oocytes that can promote heritable changes in animal physiology.

The major strength of this manuscript is the demonstration that maternal metabolites present in oocytes can influence a future generations long-term health, a likely understudied area of biology that potentially plays a more impactful role in biology than currently appreciated. The biggest drawback of this study is that the model system is couched in the framework of neurodegeneration (based on Neumann and Hillard 2014), but it remains unclear if this model (*mec-17; lon-2* mutants) really represents neurodegeneration or instead just axon fragility which indirectly leads to degeneration when neurons are mechanically damaged. (i.e. it seems unlikely that the mechanism of protection in this model system would apply to other protein aggregation models of neurodegeneration). There are also some drawbacks related to a, relatively, modest change in the average number of axon breaks on the population scale and unclear specificity of the mechanism. Overall, the claims and proposed mechanism are very exciting, but for suitability for publication in Nature Cell Biology I think the following concerns, including a few major concerns, need to be addressed:

Major concerns:

(1) The data for *PQM-1* seem to conflict with the proposed model. In Fig. 6D the authors show that *pqm-1* mutants express more *asah-1*. Based on their model, this should result in fewer PLM axon breaks. However, in Fig. 6E RNAi knockdown of *pqm-1* has no effect on axon breaks and actually prevents UA from protecting animals from axon breaks. While RNAi and mutants can behave differently, at first pass these results seem fundamentally contradictory based on the model. The authors should perform the PLM axon assay +/- UA in the *pqm-1(ok485)* mutant model. In addition, an explanation for why RNAi of *pqm-1* does not lead to fewer breaks should be included.

(2) The authors report that they identified UA in a screen of natural products, but I can't find anywhere in the manuscript how many molecules were screened or what the results of that screen

were. This information is critical to interpreting how specific these findings are for UA or if many bioactive molecules/lipids might affect axon fragility. At minimum, it would be important to show data for 20-30 (if not more) additional bioactive molecules to determine what the variation is in both this assay and any screen. In particular, it would be important to test other ceramides or precursors/derivatives besides S1P to see how specific this effect might be.

(3) To conclusively claim that ASAH-1 activity in intestines affects offspring, and rule out any background effects or misexpression it would be ideal to cross heterozygous animals expressing ges-1P::asah-1 with wild-type males and then perform the same assay on offspring that have lost the transgene. If the model is correct, the wild-type offspring from this cross should be protected from axon breaks in a way that cannot be explained by transgene misexpression or the direct exposure of germ cells to UA or S1P as in the supplementation experiments. This would substantially solidify the intergenerational claims.

(4) There exist sphk-1 mutants in *C. elegans* that can be readily obtained from the CGC. These mutants should be assayed here and, if the model is correct, UA exposure to sphk-1 mutants should not protect offspring because sphingosine production via increased asah-1 expression (induced by UA) cannot be converted to S1P. This experiment would substantially add to the specificity of the claims here.

a. Along these lines, if S1P affects asah-1 expression via a feedback loop then asah-1 should exhibit altered expression in sphk-1 mutants? Is this true? Seems like an easy test.

(5) RME-2/yolk deposits many lipids into oocytes. Because the specificity of this effect to S1P alone and not any other lipid is unclear, it would be helpful if the authors could assay other lipid pathways, using mutants, and demonstrate that decreasing the deposition of other lipids into oocytes does not have the same effect of offspring axon breaks. This is along the same lines as comment number 2. It would be important to rule out that supplementation of any lipid couldn't increase resistance to the mechanical stress that causes axon breaks.

(6) In Fig. 6 the authors demonstrate increased asah-1 expression in F1 animals, which is proposed to drive increased S1P production and continued protection from neuron breaks in F2 animals. However it remains unclear to me if an ~10% increase in asah-1 expression would be sufficient to drive substantial increases in S1P levels deposited into F2 oocytes. This is particularly concerning because the authors demonstrate that supplementing S1P at both 5 and 10 micromolar did not affect axon breaks and they needed to get to 20 micromolar supplementation to see an effect – a rather large amount of S1P, so one might expect a substantial change in F1 S1P is required for this effect to be mediated via the mechanism proposed. Ideally, it would be useful for the authors to quantify S1P levels in F1 and F2 embryos to see if the relatively modest change in asah-1 expression can really drive a meaningful and impactful increase in S1P levels. (I am aware that the amount of S1P supplemented into media does not necessarily correlate with the concentrations in live *C. elegans*, however the authors could compare their metabolomics results to the amount of S1P increase that might be observed in animals supplemented with 10 or 20 micromolar S1P).

(7) The authors should remove “neurodegeneration” from the abstract/intro/results and perhaps leave the speculation on how this might apply to neurodegeneration (as people think about it in humans) for the discussion. What is studied here is axon breaks/axon fragility. Some definitions of neurodegeneration or links to classic neurodegeneration might consider this a model of

neurodegeneration, but I suspect that many readers will be confused by this framing and misinterpret how much this actually applies to other models of neurodegenerations (right now it appears that the *mec-17;lon-2* model is only published in a single paper to date, so not exactly a well-established and long standing model). Plus, I do not think it hurts the authors main claim about the intergenerational effect to say it's an intergenerational protection of neurons from axon breaking, which is still important.

Minor concerns that should be addressed and/or would benefit the manuscript:

(8) The authors show the individual replicate values for each replicate in Fig. 2G but do not do so in any other of the many bar graphs in this manuscript. To better interpret the variance and significance of these findings these values should be added. Along these lines, I am aware that the raw values are part of a supplemental table, which includes the percentages of animals in several replicates of each experiment but does not list the number of animals assayed in each experiment – only the total number of animals assayed across all experiments. (i.e. they report an “n” of 249 animals for one experimental condition, but then provide only the percentages for ~7-10 replicates without stating the number of animals assayed in each replicate). In this case it seems like the actual “n” for each experiment is 7-10 experiments and the number of animals assayed in each replicate should be included in the raw data.

(9) The authors report that 49 genes were differentially expressed in response to UA. *Asah-1* was among them and they followed up on this one and it worked. It would be nice to know if they tested any of the other 48 and if they didn't work or would many of these genes do the same thing?

(10) *Sphk-1* antagonizes other ceramide related genes like *hyl-1/lagr-1*. It would be nice to see those mutants tested here to put this effect in context with other ceramide stress response studies in *C. elegans*.

(11) The model in Fig. 7 could be cleaned up substantially to make it more interpretable. Also, for the F1/F2 generations in the model, are the authors proposing that the effect of S1P and/or *asah-1* is in oocytes or developing embryos or in adults? Based on their experiments in the early figures of the manuscript the timing of these changes matters and the figure just labels generations with no regard to different developmental stages and timing of the action of these molecules. The model should be updated to account for these timing effects so readers can interpret it.

(12) The authors should speculate on the potential role of this biology in wild-type animals. It is very unlikely such a mechanism would have evolved to prevent axon breaks in *mec-17; lon-2* mutants, and some more discussion of what the authors think they are actually tapping into here or why it might be important for other animals would be of high interest.

REFERENCES – are limited to a total of 70 for Articles, Resources, Technical Reports; and 40 for Letters. This includes references in the main text and Methods combined. References must be numbered sequentially as they appear in the main text, tables and figure legends and Methods and

must follow the precise style of Nature Cell Biology references. References only cited in the Methods should be numbered consecutively following the last reference cited in the main text. References only associated with Supplementary Information (e.g. in supplementary legends) do not count toward the total reference limit and do not need to be cited in numerical continuity with references in the main text. Only published papers can be cited, and each publication cited should be included in the numbered reference list, which should include the manuscript titles. Footnotes are not permitted.

Methods should be written concisely, but should contain all elements necessary to allow interpretation and replication of the results. As a guideline, Methods sections typically do not exceed 3,000 words. The Methods should be divided into subsections listing reagents and techniques. When citing previous methods, accurate references should be provided and any alterations should be noted. Information must be provided about: antibody dilutions, company names, catalogue numbers and clone numbers for monoclonal antibodies; sequences of RNAi and cDNA probes/primers or company names and catalogue numbers if reagents are commercial; cell line names, sources and information on cell line identity and authentication. Animal studies and experiments involving human subjects must be reported in detail, identifying the committees approving the protocols. For studies involving human subjects/samples, a statement must be included confirming that informed consent was obtained. Statistical analyses and information on the reproducibility of experimental results should be provided in a section titled "Statistics and Reproducibility".

All Nature Cell Biology manuscripts submitted on or after March 21 2016 must include a Data availability statement as a separate section after Methods but before references, under the heading "Data Availability". For Springer Nature policies on data availability see <http://www.nature.com/authors/policies/availability.html>; for more information on this particular policy see <http://www.nature.com/authors/policies/data/data-availability-statements-data-citations.pdf>. The Data availability statement should include:

- Accession codes for primary datasets (generated during the study under consideration and designated as "primary accessions") and secondary datasets (published datasets reanalysed during the study under consideration, designated as "referenced accessions"). For primary accessions data should be made public to coincide with publication of the manuscript. A list of data types for which submission to community-endorsed public repositories is mandated (including sequence, structure, microarray, deep sequencing data) can be found here <http://www.nature.com/authors/policies/availability.html#data>.
- Unique identifiers (accession codes, DOIs or other unique persistent identifier) and hyperlinks for datasets deposited in an approved repository, but for which data deposition is not mandated (see here for details <http://www.nature.com/sdata/data-policies/repositories>).
- At a minimum, please include a statement confirming that all relevant data are available from the authors, and/or are included with the manuscript (e.g. as source data or supplementary information), listing which data are included (e.g. by figure panels and data types) and mentioning any restrictions on availability.

- If a dataset has a Digital Object Identifier (DOI) as its unique identifier, we strongly encourage including this in the Reference list and citing the dataset in the Methods.

We recommend that you upload the step-by-step protocols used in this manuscript to the Protocol Exchange. More details can found at www.nature.com/protocolexchange/about.

All imaging data should be accompanied by scale bars, which should be defined in the legend. Cropped images of gels/blots are acceptable, but need to be accompanied by size markers, and to retain visible background signal within the linear range (i.e. should not be saturated). The boundaries of panels with low background have to be demarked with black lines. Splicing of panels should only be considered if unavoidable, and must be clearly marked on the figure, and noted in the legend with a statement on whether the samples were obtained and processed simultaneously. Quantitative comparisons between samples on different gels/blots are discouraged; if this is unavoidable, it should only be performed for samples derived from the same experiment with gels/blots were processed in parallel, which needs to be stated in the legend.

- We do not recommend using Adobe Photoshop for designing figures, but we can accept Photoshop

generated (.PSD or .TIFF) files only if each element included in the figure (text, labels, pictures, graphs, arrows and scale bars) are on separate layers. All text should be editable in 'type layers' and line-art such as graphs and other simple schematics should be preserved and embedded within 'vector smart objects' - not flattened raster/bitmap graphics.

Unprocessed scans of all key data generated through electrophoretic separation techniques need to be presented in a supplementary figure that should be labelled and numbered as the final supplementary figure, and should be mentioned in every relevant figure legend. This figure does not count towards the total number of figures and is the only figure that can be displayed over multiple pages, but should be provided as a single file, in PDF or TIFF format. Data in this figure can be displayed in a

relatively informal style, but size markers and the figures panels corresponding to the presented data must be indicated.

The total number of Supplementary Figures (not including the “unprocessed scans” Supplementary Figure) should not exceed the number of main display items (figures and/or tables (see our Guide to Authors and March 2012 editorial <http://www.nature.com/ncb/authors/submit/index.html#suppinfo>; <http://www.nature.com/ncb/journal/v14/n3/index.html#ed>). No restrictions apply to Supplementary Tables or Videos, but we advise authors to be selective in including supplemental data.

GUIDELINES FOR EXPERIMENTAL AND STATISTICAL REPORTING

REPORTING REQUIREMENTS – We are trying to improve the quality of methods and statistics reporting in our papers. To that end, we are now asking authors to complete a reporting summary that collects information on experimental design and reagents. The Reporting Summary can be found here <https://www.nature.com/documents/nr-reporting-summary.pdf> If you would like to reference the guidance text as you complete the template, please access these flattened versions at <http://www.nature.com/authors/policies/availability.html>.

We strongly recommend the presentation of source data for graphical and statistical analyses as a separate Supplementary Table, and request that source data for all independent repeats are provided when representative experiments of multiple independent repeats, or averages of two independent experiments are presented. This supplementary table should be in Excel format, with data for different figures provided as different sheets within a single Excel file. It should be labelled and numbered as one of the supplementary tables, titled “Statistics Source Data”, and mentioned in all relevant figure legends.

Author Rebuttal to Initial comments

Reviewers' Comments:

Reviewer #1:

Remarks to the Author:

A. Wang et al provide evidence to support the concept that in *C. elegans*, intestinal expression and function of acid ceramides (Asah1) leads to the production of S1P, transfer to the yolk/ oocytes via the lipoprotein receptor *rme-2* which leads to the ability to resist axonal breaks in the worm neurons brought about by microtubule defects. They suggest that maternal induction of *Asah-1* in the intestinal cells is critical for this event, which lasts for 2-3 generations. They suggest that intergeneration transfer of maternal sphingolipid signal to the yolk is perhaps explained by epigenetic changes in the embryo which are sensitive to sphingolipid auto regulatory networks.

B. the work is highly novel and potentially very significant.

C. It should be mentioned that i am not well versed in the nuances of *C. elegans* genetics. My expertise is in sphingolipids and cell biology. Nevertheless, my opinion is that the sphingolipid work and cell biological mechanisms that the authors are examining are overall fine. There are some minor issues that the authors should address which i will detail below. However, the overall message of the paper is OK.

D. OK

E. I feel the authors' data are quite solid. the experiments seem to have appropriate controls and appear to support the conclusions.

F. Suggestions for improvement:

1. Ceramidase enzymes have several isoforms - acid, alkaline and neutral. I assume that the authors have not characterized the biochemistry of *C. elegans* ceramidase(s). It would be helpful to mention the possible existence of isoforms by gene sequence homology at the very least. In mammals, intestinal enzyme is alkaline and acid ceramides is either on the cell surface or lysosomes. Mammalian enzyme is also known for its important function in early embryonic development. These issues should be considered and/or discussed.

C. elegans encodes three ceramidases: ASAH-1 (acid), ASAH-2 (acid) and W02F12.2 (alkaline) - none of which have been characterized biochemically and the acid/alkaline nature has been designated by homology only. We have added a sentence in the discussion to mention the existence of the other isoforms but as ASAH-1 is the only isoform induced by ursolic acid we only focused on this isoform.

In addition, we found that ASAH-2 intestinal overexpression is not neuroprotective and *asah-2* RNAi does not prevent UA-induced neuroprotection (see Reviewer Figure below). Hence, *asah-1* seems to be playing a specific role in this context.

Legend for this figure:

Left graph - Three independent transgenic lines overexpressing the *asah-2* in the intestine does not reduce PLM axon breaks in *mec-17(-); lon-2(-)* animals. *P* values assessed by unpaired t test. n.s., not significant. Error bars indicate SEM. n = 72-79.

Right graph - RNAi knockdown of *asah-2* does not prevent UA from reducing PLM axon breaks in *mec-17(-); lon-2(-)* animals. *P* values assessed by unpaired t test. ***P* ≤ 0.05. Error bars indicate SEM. n = 101-128.

2. Although the authors rule out the requirements for RNAi/Ago pathway in intergenerational inheritance, their conclusion of "epigenetic" mechanisms is speculative. This should be further explained. Is it possible that S1P action on novel GPCRs or other downstream effectors is involved in the oocyte and embryonic development? The authors might want to consider that GPR3 which is critical for mammalian oocyte meiosis has been suggested to be a sensor of S1P even though this has not been fully vetted since this GPCR is also a constitutive activator of Gs and adenylate cyclase.

We agree and have changed the term 'epigenetic' to 'intergenerational inheritance' where applicable in the manuscript.

It is possible that S1P interacts with an unknown GPCR in *C. elegans*. However, GPR3 has no worm ortholog and identifying which putative GPCR interacts with S1P is beyond the scope of this study.

3. The list of genes regulated by UA (transcriptome data mentioned) should be provided with proper statistics on induction/ repression properties in the data supplement.

This was provided in Table S1 of the original submission and includes false discovery rate (FDR) and P value for differentially expressed genes.

4. The authors oversimplify their consideration of ceramide/sphingolipid pathways. They fail to consider sphingomyelinase pathway which can generate ceramide. This should be considered. Do homologues/ orthologues of SPMD enzymes exist in the worm?

Yes - in our original manuscript we focussed on ASAH-1/SPHK-1 branch of the pathway as we found that ASAH-1 expression is positively regulated by Ursolic Acid. There are three sphingomyelinases (ASM-1, ASM-2 and ASM-3) encoded in *C. elegans* with only ASM-2 with detectable intestinal expression.

We have now examined whether transgenic overexpression of ASM-2 in the intestine (*ges-1* promoter) can affect PLM axon fragility of *mec-17(-); lon-2(-)* animals. However, we found that animals overexpressing ASM-2 in the intestine did not detectably affect PLM axon fragility (see Reviewer figure below - left). In addition, we supplemented *mec-17(-); lon-2(-)* animals with sphingomyelin and did not observe a change in PLM axon fragility (See Reviewer Figure below - right). These data suggest that manipulation of sphingomyelin levels do not protect the PLM neurons and perhaps that the subcellular source of ceramide (e.g. lysosome) is a critical factor in this context. We have not mentioned these data in the revised manuscript as we do not believe it adds to the story. Obviously, the sphingolipid pathway is very complicated with multiple enzymes, acting in distinct subcellular compartments to generate many different lipids. We decided to focus this manuscript on S1P as this is the most downstream factor that protects the PLM neurons.

Legend for this figure:

Left graph - Three independent transgenic lines overexpressing the sphingomyelinase ASM-2 in the intestine does not reduce PLM axon breaks in *mec-17(-); lon-2(-)* animals. *P* values assessed by unpaired t test. n.s., not significant. Error bars indicate SEM. n = 88-95.

Right graph - Supplementing Sphingomyelin (SM) at three concentrations (10, 20 and 40 μ M) does not reduce PLM axon breaks in *mec-17(-); lon-2(-)* animals. *P* values from one-way analysis of variance (ANOVA). n.s., not significant. Error bars indicate SEM. n = 65-73.

5. Page 10, penultimate line - "complexity" is not a proper word in this context - do the authors mean "levels"?

We have changed 'loss of sphingolipid complexity' to 'loss of multiple sphingolipids'.

6. In Sptl-1 overexpression experiments, is this enzyme a single subunit enzyme in the worm? in mammals, at least 3 subunits are needed.

Yes - As in mammals there are 3 SPTL subunits (SPTL-1, SPTL-2 and SPTL-3). We have changed the text from 'Serine palmitoyltransferase (SPT) is the rate-limiting enzyme' to "The serine palmitoyltransferase (SPT) protein complex is the rate-limiting enzyme".

7. Figure 4D is not described in the results.

We described Figure 4D in this sentence "We found that driving intestinal *sphk-1* expression reduces PLM axon degeneration of *mec-17(ok2109); lon-2(e678)* animals, as we previously showed by overexpressing *sptl-1* or *asah-1* (Figure 4B-D)."

8. How does 20 μ M S1P supplied exogenously taken up by the worms and transported to the oocytes/ yolk?

We observed exogenously supplied S1P-fluorescein in the intestine of worms and in oocytes within these same animals (Figure 5F). Further, the neuroprotective effect of S1P is dependent on a yolk receptor (RME-2) that is expressed in oocytes (Figure 5G). This suggests that S1P is transported from the intestine to oocytes in the yolk which was previously identified to contain lipids and lipoproteins.

Is it metabolized? The authors can use C17-S1P (odd chain) and follow by LC/MS in the eggs.

Unfortunately, we were unable to detect S1P fluorescein in embryos suggesting that it is metabolized/is incorporated into membranes of certain cells or is at very low levels. Using C17-S1P and performing LC/MS in worm eggs is a good idea in principle but not feasible. Multiple studies have also shown that S1P is often challenging to detect in worm embryos due to their rapid development. Therefore, we believe that attempting to detect changes in S1P levels in rapidly developing embryos is not possible at present.

9. The idea of sphingolipid auto regulatory networks should be discussed more fully. Please refer to the following examples in the literature (PMID: 28956531, PMID: 27226645).

Thanks - we have now noted these examples in the discussion.

"Examples, of such sphingolipid autoregulatory networks have previously been identified in cellular and zebrafish models. In these studies, sphingolipid metabolic enzymes were induced by accumulation of their sphingolipid substrate."

10. S1P and Sphk enzymes were shown to be very important in early embryonic events in vertebrates. In addition, S1P protects mammalian oocytes and sperm from stress-induced death. This should be discussed.

We have now added some sentences in the discussion regarding the roles of S1P and SPHK in vertebrates.

"Essential roles for S1P/Sphk have also been revealed in protecting vertebrate oocytes from radiation-induced apoptosis and for the development of cardiovascular tissue in embryos."

Reviewer #2:

Remarks to the Author:

This manuscript reports that a natural product, ursolic acid (UA) attenuates PLM axon degeneration across two generations. The authors have characterized the mechanism underlying this protection. UA induces the expression of *asah-1*, encoding acid ceramidase that hydrolyzes ceramide into fatty acid and sphingosine, in the intestine, and intestinal overexpression of *asah-1* is sufficient to reduce PLM axon degeneration. Through genetic analyses of the metabolic steps upstream and downstream of *ASAH-1*, the authors have narrowed down sphingosine-1-phosphate (S1P) responsible for the transgenerational protective effect. Using fluorescence-labeled S1P and genetic analysis, the authors revealed that S1P transports from the intestine to oocytes and acts through RME-2 receptor in the germline to mediate the protective effect. Furthermore, the authors discovered that two transcription factors respond to UA and negatively regulate *asah-1* expression. Together, the manuscript discovers an interesting protective effect of environmental UA and endogenous S1P on PLM axon degeneration through generations and delineates the molecular mechanism mediating this protective effect.

Two major weaknesses to me are:

1. It remains unclear what happens to neurons in F1 and F2 generations. Is the protective effect due to higher SIP levels in neurons? With a higher concentration of the SIP treatment, is it possible to see a protective effect in P0?

We performed the suggested experiment by applying 40uM of S1P (double the 20uM we originally applied) to L1 larvae and examined axon breaks in day 3 adults. We did not observe a significant reduction in axon breaks in the *mec-17*; *lon-2* mutant with higher S1P in these P0 animals (see Reviewer graph below).

In combination with our analysis of the spatiotemporal requirements for *asah-1* and S1P, this strengthens our original data that the critical window for S1P neuroprotection is during oocyte/embryonic development and that S1P is transferred between the intestine and oocyte for the intergenerational effect to occur. This suggests that S1P alters oocyte and/or embryonic membrane/microtubule health (our reply to point 2 below supports this).

mec-17(ok2109); lon-2(e678); zdl5

The authors show that the ectopic overexpression of *asah-1* in neurons leads to axon outgrowth in P0. How about neuronal overexpression of *sphk-1* or overexpression of both *asah-1* and *sphk-1* at the same time? Will this lead to a protective effect in P0?

As suggested, we overexpressed *sphk-1* in the nervous system however this caused outgrowth defects (as we showed with *asah-1* overexpression in neurons) and an increase in PLM axon fragility (see Figure below). We do not think that these data add anything new to the manuscript, so we provide the data to the Reviewer. In the light of these data, we did not attempt to overexpress *asah-1* and *sphk-1* simultaneously as overexpression of them individually already causes developmental defects.

mec-17(ok2109);lon-2(e678)

Legend for this figure:

Three independent transgenic lines overexpressing *sphk-1* in the nervous system increases PLM axon breaks in *mec-17(-); lon-2(-)* animals. *P* values assessed by unpaired t test. **P* ≤ 0.05; ***P* ≤ 0.01. Error bars indicate SEM. n = 112-128.

If the protective effect is not due to the SIP increase in neurons, what could be other possible mechanisms?

Our data in point 2 below show that UA and S1P improve axonal transport. This suggests that elevated S1P in the oocyte/embryo is incorporated into neuronal membranes and provides stability to microtubules. This is incredibly challenging to confirm in the *C. elegans* embryo. However, a previous study (PMID: 33380429) showed using cells in culture that sphingolipids can stabilize microtubules through a direct interaction.

This may be counterintuitive to our data showing that *asah-1* or *sphk-1* overexpression in neurons causes axon defects. However, these overexpression experiments likely disrupt multiple sphingolipid metabolites that could interfere with signalling and cytoskeletal dynamics that are required for axon development.

2. Whether this protective effect is specific to PLM adult-onset axon degeneration. MEC-17 loss in the *lon-2* mutant background leads to axon degeneration likely due to defective axonal transport. Does the UA and the S1P treatments and/or the *asah-1* and *sphk-1* overexpression protect against disrupted mitochondrial distribution and/or synaptic protein localization. The current manuscript is solely dependent on axon break counting. Cellular characterization will help strengthen the claim on the improvement of neural health.

We have now performed extensive experiments to determine the effect of UA and S1P on mitochondrial distribution and axon transport in the PLM neurons of *mec-17* mutant animals. We found that UA and S1P reduce the *mec-17(-)* transport defects of two models: *mCherry::RAB-3* (a synaptic vesicle-associated protein) and *UNC-104::GFP* (Kinesin-3). These data are now in Figure 6. In contrast, UA and S1P did not improve mitochondria distribution in the PLM neurons of *mec-17(-)* animals where mitochondria are inappropriately located in the posterior PLM neurite (see Reviewer data graph below). This suggests that the improvement in transport garnered by UA/S1P is insufficient to improve organelle transport. This may be why axon breaks are not completely prevented when animals are treated with UA/S1P.

vdEx484(Pmec-4::tommm-20::mRFP); mec-17(ok2109);zdl5

Legend for this figure:

UA incubation does not improve mitochondrial distribution in the PLM neurons of *mec-17(-)* animals. *P* values assessed by unpaired t test. n.s. = not significant. Error bars indicate SEM. n = 45-55.

In addition, we show that:

- 1) UA and S1P reduce PLM axon fragility of wild-type and *mec-17* mutant animals incubated with the microtubule destabilizing drug colchicine. This suggests that S1P acts to stabilize microtubules independent of the *mec-17* mutation (as has been shown previously for ceramide - PMID 33380429).
- 2) S1P reduces fragility of the PLM and GABAergic D-type motor neurons in an alternative genetic model (*lin-14* mutant).

3) UA and S1P reduce PVQ axon defects in an alternative genetic model (ced-10 mutant)

These new data are presented now in new Figure 6.

One minor point:

Line 203, "asah-1 overexpression disrupts axon outgrowth of the ALM and PLM axons" should be "asah-1 overexpression causes axon outgrowth of the ALM and PLM axons" based on the context.

We don't agree with this - inappropriate asah-1 overexpression in neurons does cause axon outgrowth defects (see Figure 3C and S5).

Reviewer #3:

Remarks to the Author:

In this study by Wang et al entitled "Intergenerational neuroprotection by an intestinal sphingolipid in *Caenorhabditis elegans*" the authors show an intergenerational neuroprotective effect of acute ursolic acid (UA) intake supplementation in *C. elegans*. The authors demonstrate that 16 h exposure to UA during the gamete producing stage is sufficient to prevent axonal breaks for 2 subsequent generations in mutant animals lacking the MEC-17/TAT1 α -tubulin acetyltransferase sensitized by an increased body length due to a lon-1 mutation. They reveal that such neuroprotection is caused by the upregulation of ASAH-1 expression in the intestine, which leads to maternal transfer of sphingosine-1-phosphate (S1P) to the offspring through the yolk receptor RME-2. Importantly, they show that S1P supplementation is sufficient to induce neuroprotection and that the level of this lipid regulates this effect. In addition to the above, the authors show that two transcription factors PQM-1 and CEH-60 are involved in asah-1 regulation, and that two enzymes, SPTL-1 and SPHK-1, functioning within the S1P pathway are also implicated.

This manuscript covers an important biological question of how parental diet affects the health of the progeny. However, despite the large set of data presented and genetic pathway identified, there are some key elements of concern that need to be addressed to support their conclusions. There are two main conceptual issues:

1. The use of the term 'neuroprotection' evokes an idea of protection from many neurodegenerative insults. However, the entire manuscript is based on a single model of axonal degeneration that requires two mutations, *mec-17* and *lon-2*, to be present and that have been described to affect microtubule stability (Neumann & Hilliard, Cell Rep 2014). The paper does not address any role of the UA (and the related identified pathway) on these specific genes. An alternative axonal degenerative model should be tested for the author to conclude a broader neuroprotective function of UA. For example, have the authors tried the effect of UA on axonal degeneration upon injury, or upon treatment with microtubule drugs (colchicine and paclitaxel), or on another independent axonal degeneration mutations? Moreover, did they test if the neuroprotection extends to other classes of neurons, and thus independent on the *mec-17* effect, which appears selective for the *mec* neurons?

We have performed extensive experiments to address these questions and found that:

- 1) UA and S1P reduce PLM axon fragility of wild-type and *mec-17* mutant animals incubated with the microtubule destabilizing drug colchicine. This suggests that S1P acts to stabilize microtubules (as has been shown previously for ceramide - PMID 33380429).
- 2) S1P reduces fragility of the PLM and GABAergic D-type motor neurons in an alternative genetic model (*lin-14* mutant).
- 3) UA and S1P reduces PVQ defects (alternative neuronal model) in a *ced-10* model (alternative genetic model).

These new data are presented now in new Figure 6.

2. There is no indication on how the UA treatment affects the axon. Did they test whether UA alters microtubules dynamics in these axons, or axonal transport, or both? While an effect of UA is not questioned, it was not shown, mechanistically, how this relates to the axon (i.e. what this pathway does to the axonal structure to prevent breaks).

We have now performed extensive experiments to determine the effect of UA and S1P on mitochondrial distribution and axon transport in the PLM neurons of *mec-17* mutant animals. We found that UA and S1P reduce the *mec-17*(-) transport defects of two models: *mCherry::RAB-3* (a synaptic vesicle-associated protein) and *UNC-104::GFP* (Kinesin-3). In contrast, UA and S1P did not improve mitochondria distribution in the PLM neurons of *mec-17*(-) animals in which mitochondria are inappropriately located in the posterior PLM neurite (see Reviewer data graph below). This suggests that the improvement in transport garnered by UA/S1P is insufficient to improve organelle transport. This may be why axon breaks are not completely prevented when animals are treated with UA/S1P.

These new data are presented now in new Figure 6 and in the Reviewer Figure below.

vdEx484(*P_{mec-4}::tomm-20::mRFP*);
mec-17(ok2109);zdl5

Legend for this figure:

UA incubation does not improve mitochondrial distribution in the PLM neurons of *mec-17(-)* animals. *P* values assessed by unpaired t test. n.s. = not significant. Error bars indicate SEM. n = 45-55.

Considering these two issues combined, the paper in its currently format falls short in providing a molecular mechanism of neuroprotection; further experiments as suggested above will allow to validate, strengthen, and broaden their conclusions.

Together these new data reveal a broader role for UA and S1P in nervous system development and health and show that UA/S1P have a positive effect on axon transport. These data are presented in new Figure 6.

Other important points

1. The authors indicated on line 111 that they performed a screen of natural products from which UA was identified. They should provide the list of those natural products that have been screened and their effect (presumably negative).

We have now included data relating to the entire screen of 21 natural products in Figure S1 with their names listed in the Source data file. UA was the strongest hit from this screen and hence we pursued its mechanistic function in this manuscript.

2. The authors should expand the UA concentration curve as the effect seems to reduce at 100 μ M. Given this is still a relative low concentration, testing one or two additional higher points will allow them to determine if the effect is peaking at 50 μ M or increases at higher concentrations.

We did try this previously but did not include it in the manuscript. We found that when 200 μ M UA did not protect the PLM axons which may be due to adverse effects of the higher DMSO concentration. Therefore, we only present data up to 100 μ M UA in Figure S2.

3. The statement on line 118 – 119 “for one generation” is not clear. If I understand correctly, these animals have been exposed for two generations, the P0 and the F1, with the F1 adults being tested.

We have now modified this ‘We found that incubating animals with 50 μ M UA from larval stage 4 (L4) robustly reduced axon degeneration in *mec-17(ok2109); lon-2(e678)* F1 progeny’

4. In line 120, the reference to Figure 1D does not indicate when the animals were incubated, and that the population scored is F1.

We have now made this clear in Figure 1D by adding the incubation period to the Figure legend. ‘Exposure of UA (50 μ M) from P0 L4 to 3-day old adult F1s reduces PLM axon breaks in *mec-17(ok2109); lon-2(e678)* F1 animals.’

5. Related to the point above, in Figure 1E it seems important to determine whether the incubation with UA of the P0 at early embryo (before L4) is sufficient to induce the effect in the next generation (i.e. the UA incubation described in “c” should be tested in the P0). This is also important for the statement in lines 131-133 in which only incubation of L4-adults gives a phenotype.

We have now performed these experiments to confirm that UA incubation in P0s and not F1s is neuroprotective. We incubated freshly laid embryos with UA continuously or until the L1 stage and scored PLM axon breaks (day 3 adults) in P0s or F1s. Only animals exposed to UA during the L4/young adult period reduced PLM axon breaks

in F1 progeny. These data also suggest that UA is unable to cross the eggshell. These new data are provided in Figure S2F and described in the results section.

6. In relation to *sptl-1*, did they test the function in different tissues? Did they test whether *sptl-1* and *sphk-1* RNAi also block the UA protection? These experiments should be performed.

We have now overexpressed *sptl-1* in the nervous system which, as one may predict from our data in the initial submission, does not rescue PLM breaks in *mec-17(-); lon-2(-)* animals (see Review Figure below - left). In addition, we overexpressed *sphk-1* in the nervous system which actually increases PLM axon breaks in *mec-17(-); lon-2(-)* animals (see Review Figure below - right). These data corroborate our hypothesis that PLM neuroprotection is governed by S1P generation outside the nervous system, and we don't think they add any further insight into the manuscript.

Legend for this figure:

Two independent transgenic lines overexpressing *sptl-1* (left) or *sphk-1* (right) in the nervous system. Neither overexpression construct reduces PLM axon breaks in *mec-17(-); lon-2(-)* animals, with *sphk-1* overexpression increases PLM axon breaks. *P* values assessed by unpaired *t* test. **P* ≤ 0.05; ***P* ≤ 0.01. Error bars indicate SEM. *n* = 84-98 (*Prab-3::sptl-1*), *n* = 118-128 (*Prab-3::sphk-1*).

In Figure S6 we show that using either *sphk-1* RNAi or a *sphk-1* deletion mutant blocks UA neuroprotection in *mec-17(-); lon-2(-)* animals. As *sptl-1* RNAi causes embryonic lethality and larval arrest we could not perform this experiment and hence our use of *sptl-1* overexpression in the intestine which reduces PLM axon breaks in *mec-17(-); lon-2(-)* animals (Figure 4).

7. While the authors show that gametes exposed to UA drive the main effect, it would be interesting to determine which gametes are sufficient to cause it or if both are necessary. This could be tested by exposing male nematodes to UA in the sperm producing stage and then crossing these animals to untreated hermaphrodites to then analyse neuroprotection in their cross-progeny.

We have clearly shown that the UA-effect acts through oocytes and requires the RME-2 yolk receptor. Testing the relevance of sperm would be potentially interesting but believe is beyond the scope of this work.

8. The authors hypothesize that the intergenerational effect is not due to maternal small RNA transfer by investigating HRDE-1-deficient animals; however, it is not clear whether this molecule is the sole nematode controller of sRNA transport. If that is not the case, then sRNA transfer cannot be ruled out as a mechanism as other molecules with redundant effect might compensate for the lack of HRDE-1.

Multiple studies of transgenerational inheritance by small RNAs have shown a major dependence on HRDE-1 (e.g. PMID: 31178120, PMID: 25018105, PMID: 22738726). Another RNA binding protein ZNFX-1 has also more recently been shown to control small RNA inheritance (PMID: 29769721, PMID: 35739318). We therefore examined the potential role for ZNFX-1 in intergenerational neuroprotection by UA and found that neither HRDE-1 nor ZNFX-1 are required. These new data are added to Figure 2C.

9. In relation to the *ASAH-1* expression, the authors did not test whether expression in PLM neurons can rescue the defect of the *asah-1* RNAi. Do *asah-1* mutants phenocopy the *asah-1* RNAi-treated animals? This would be

an important confirmation and, more importantly, will allow them to conclusively test cell-autonomy in a precise way, without over-expressing it in a wild-type background.

As suggested, we have now tested the *asah-1(tm495)* deletion mutant and show that *asah-1* loss enhances the percentage of PLM breaks observed in *mec-17(-); lon-2(-)* animals and prevents UA neuroprotection (phenocopying the *asah-1* RNAi) - New Figure 2F. As RNAi is very efficient in the intestine and barely works in the nervous system, and that overexpression of *asah-1* in the intestine protects the PLM neurons, our collective data support a function for *asah-1* in the intestine.

At this regard, in the experiments in which they observed an over-expression defect in mechanosensory neurons when ASAH-1 is driven by the *mec-4* promoter, what concentration and how many transgenic lines were studied? Have they tried the single copy expression to ensure the minimal amount is present?

We observed defects in the mechanosensory neurons when *asah-1* was overexpressed in two independent transgenic lines using either the *mec-4* or *rab-3* promoter. Crucially, we found that overexpression of the *mec-4* promoter alone (at the same concentration of 10ng/ul) does not cause axon outgrowth defects. To allay any residual concerns, we subsequently diluted the *mec-4p::asah-1* plasmid concentration to 1ng/ul and 0.1ng/ul and continued to observe *asah-1* outgrowth defects (see Reviewer lines for each and fully support Figure below). Two conditions were used our original findings.

Legend for this figure:

Two independent transgenic lines expressing a Pmec-4::asah-1 transgene at 1ng/ul or 0.1ng/ul in *zdl5* animals causes PLM outgrowth defects. 50% and 75% outgrowth defects refers to the approximate length of PLM axons compared to control. *P* values from one-way analysis of variance (ANOVA). **P* ≤ 0.05; ***P* ≤ 0.01; *****P* ≤ 0.0001. n = 72-78.

Single copy expression of *asah-1* using a mechanosensory neuron promoter is in theory a good idea. However, one would still need to use a mechanosensory neuron-specific promoter to drive expression of a single copy and this could very well be overexpressed and potentially cause PLM developmental defects and therefore would be challenging to interpret.

10. It is very curious that ASAH-1 overexpression specifically, affects PLM/ALM morphology and development but not movement controlling neurons. Moreover, the main neuroprotective effect observed in this study is also in PLM/ALM. It would be interesting to discuss why this is the case, especially considering that PLM/ALM are microtubule-rich neurons. Maybe ASAH-1 interferes with microtubule stabilization? ASAH-1 effect could be tested in conjunction with microtubules (de)stabilizing drugs, as also indicated in the two major points above.

Agreed - please refer to our new data described in Major Points 1 and 2 above.

11. To complement the SPTL-1 analysis, it would be interesting to try the conditional degradation of this molecule within the intestines through the AID system to overcome lethality.

This is potentially an interesting question, but we believe is beyond the scope of this work. The focus of this study is ASAH-1 and its downstream effect. SPTL-1 is very much an upstream component of the pathway that can affect the abundance of multiple types of ceramides and thus this experiment would be challenging to interpret.

Text suggestions

1. Missing reference on line 218 after "...ASAH-1 substrate..."

Reference added.

2. The statement starting on line 360 about outgrowth/guidance contradicts the previous statement where it is said that "...caused developmental axon outgrowth defects in the ALM and PLM neurons.". Perhaps the authors meant to say that outgrowth/guidance of movement controlling neurons is not affected?

We have now modified this sentence as suggested to read 'However, we did not observe overt motility defects in animals overexpressing *asah-1* in neurons, suggesting that axon outgrowth/guidance of motor neurons is largely intact, and that disrupted sphingolipid homeostasis has context-specific effects.'

3. Figure 4 legend, please replace 'worm' with nematode in "... (worm and mammalian ortholog) ..."

Done.

4. Figure 6 legend, please replace 'worm' with animal in "...of intestinal cells per worm..."

Done.

5. Figure 1D and 2D, n number is not indicated in the legend.

Done.

Reviewer #4:

Remarks to the Author:

Here, Wang et al., identify ursolic acid as a neuroprotectant model that prevents axon damage in *mec-17*; *lon-2* *C. elegans* mutants which are particularly prone to axon damage due to mechanical stress that occurs during movement. They show that ursolic acid induces the expression of the acid ceramidase *asah-1*, which in turn promotes sphingosine-1-phosphate production. Finally, the authors show that S1P can be provisioned into oocytes in a way that promotes neuroprotection in F1 and F2 descendants of animals exposed to ursolic acid. This study would be among the first to demonstrate the provisioning of a metabolite into oocytes that can promote heritable changes in animal physiology.

The major strength of this manuscript is the demonstration that maternal metabolites present in oocytes can influence a future generations long-term health, a likely understudied area of biology that potentially plays a more impactful role in biology than currently appreciated. The biggest drawback of this study is that the model system is couched in the framework of neurodegeneration (based on Neumann and Hillard 2014), but it remains unclear if this model (*mec-17*; *lon-2* mutants) really represents neurodegeneration or instead just axon fragility which indirectly leads to degeneration when neurons are mechanically damaged. (i.e. it seems unlikely that the mechanism of protection in this model system would apply to other protein aggregation models of neurodegeneration). There are also some drawbacks related to a, relatively, modest change in the average number of axon breaks on the population scale and unclear specificity of the mechanism. Overall, the claims and proposed mechanism are very exciting, but for suitability for publication in Nature Cell Biology I think the following concerns, including a few major concerns, need to be addressed:

Major concerns:

(1) The data for PQM-1 seem to conflict with the proposed model. In Fig. 6D the authors show that *pqm-1* mutants express more *asah-1*. Based on their model, this should result in fewer PLM axon breaks. However, in Fig. 6E RNAi knockdown of *pqm-1* has no effect on axon breaks and actually prevents UA from protecting animals from axon breaks. While RNAi and mutants can behave differently, at first pass these results seem fundamentally contradictory based on the model. The authors should perform the PLM axon assay +/- UA in the *pqm-1(ok485)* mutant model. In addition, an explanation for why RNAi of *pqm-1* does not lead to fewer breaks should be included.

We understand the perceived conflict here which we have now addressed in the manuscript results section

“As PQM-1 and CEH-60 are also required for faithful yolk delivery to oocytes due to their control of intestinal vitellogenin expression, we examined their role UA-induced neuroprotection. We found that *ceh-60* or *pqm-1* RNAi inhibited UA neuroprotection in *mec-17(ok2109)*; *lon-2(e678)* animals (Figure 7E-F). Therefore, PQM-1 and CEH-60 enable *asah-1* induction and neuroprotection in response to UA exposure”.

Answers to the specific reviewer requests are below:

1) We have now performed the PLM assay +/- UA in the *pqm-1(ok485)* mutant and this phenocopies the *pqm-1* RNAi data showing that PQM-1 is required for UA to reduce PLM axon fragility. These new data are now in Figure 7F.

2) As the reviewer points out, *asah-1* expression is increased in the *pqm-1(ok485)* mutant (Fig 7D). This suggests that PQM-1 is a repressor of *asah-1* expression. As the reviewer also highlights one would expect such *asah-1* upregulation to suppress PLM axon breaks in *mec-17*; *lon-2* animals. However, PQM-1 is critical for the expression of vitellogenins (see Figure below from PMID: 27401555 showing reduced mRNA levels of *vit-1-6*) and many other intestinal lipid-related genes (PMID: 30956009). Lowered vitellogenin expression results in reduced yolk provisioning to oocytes and is therefore would reduce S1P transfer from the intestine to oocytes. Hence, the inhibition of the UA-induced PLM axon break suppression upon *pqm-1* loss. We discussed this further in the manuscript

These data show that multiple vitellogenin genes have reduced expression compared to wild-type (white bars). This would reduce yolk provisioning to oocytes. PMID: 27401555.

We tested these assertions further by performing *pqm-1* RNAi on animals in which the PQM-1 binding site is mutated in the *asah-1* promoter. Our original data showed that this PQM-1 binding site mutant has elevated ASA-1::F2A::GFP::H2B levels (Figure 7H) and protects the PLM axons (Figure 7I). We found that *pqm-1* RNAi (and thus reduced yolk components) inhibits the positive effect the PQM-1 binding site mutant has on PLM axon breaks. Thus, even though these animals have elevated ASA-1 the PLM axons are not protected in *pqm-1* RNAi animals due to yolk defects. These Reviewer data are shown below.

asah-1::f2a::gfp::h2b (PQM-1 motif mutant);
mec-17(-); lon-2(-)

pqm-1 RNAi increases PLM axon breaks in *mec-17(-); lon-2(-)* animals, overexpressing *asah-1* (induced by the PQM-1 binding site mutation). *P* value assessed by unpaired t test **P* ≤ 0.05. Error bars indicate SEM. n = 93-99.

(2) The authors report that they identified UA in a screen of natural products, but I can't find anywhere in the manuscript how many molecules were screened or what the results of that screen were. This information is critical to interpreting how specific these findings are for UA or if many bioactive molecules/lipids might affect axon fragility. At minimum, it would be important to show data for 20-30 (if not more) additional bioactive molecules to determine what the variation is in both this assay and any screen. In particular, it would be important to test other ceramides or precursors/derivatives besides S1P to see how specific this effect might be.

We have now included data relating to the entire screen of 21 natural products in Figure S1. UA was the strongest hit from this screen and hence we pursued its mechanistic function in this manuscript.

As requested, we have now also tested other lipids:

Three fatty acids (palmitate, stearate and oleate) that can be used by *C. elegans* to generate other lipids. - they do not affect PLM axon breaks in *mec-17; lon-2* mutant animals.

We also tested sphingomyelin, which is converted to ceramide (predominantly in the plasma membrane), and two ceramides with different fatty acid chain lengths (C20 Ceramide (d18:1/20:0) and C22 Ceramide (d18:1/22:0)) that can be converted into sphingosine and S1P. We found that incubating animals with ceramide but not sphingomyelin reduced PLM axon breaks in *mec-17; lon-2* mutant animals. These experiments are challenging to interpret as the subcellular compartment they enter, and dynamics of metabolism could affect their ability to control axon health. As such, we present these data to the Reviewer as we think that introducing them in the manuscript would unnecessarily complicate the take home message.

Incubation with ceramide derivatives but not sphingomyelin, palmitate, stearate nor oleate reduces PLM axon breaks in *mec-17(-); lon-2(-)* animals. *P* values from one-way analysis of variance (ANOVA). **P* ≤ 0.05; ***P* ≤ 0.01; n.s. not significant. Error bars indicate SEM. n = 81-92.

(3) To conclusively claim that ASAH-1 activity in intestines affects offspring, and rule out any background effects or misexpression it would be ideal to cross heterozygous animals expressing *ges-1P::asah-1* with wild-type males and then perform the same assay on offspring that have lost the transgene. If the model is correct, the wild-type offspring from this cross should be protected from axon breaks in a way that cannot be explained by transgene misexpression or the direct exposure of germ cells to UA or S1P as in the supplementation experiments. This would substantially solidify the intergenerational claims.

We have now performed the following experiment:

Using the strain in which we express *asah-1* in the intestine (*ges-1p::asah-1*) in *mec-17(-); lon-2(-)* animals, we compared PLM break percentage in transgenic and non-transgenic progeny. We found that non-transgenic progeny (not expressing the transgene) also have reduced PLM axon breaks. This is now in Figure 3B and discussed in the results section.

(4) There exist *sphk-1* mutants in *C. elegans* that can be readily obtained from the CGC. These mutants should be assayed here and, if the model is correct, UA exposure to *sphk-1* mutants should not protect offspring because sphingosine production via increased *asah-1* expression (induced by UA) cannot be converted to S1P. This experiment would substantially add to the specificity of the claims here.

This experiment was performed in the original manuscript and supports our conclusions and the reviewers assertion - see Figure S6.

Along these lines, if S1P affects *asah-1* expression via a feedback loop then *asah-1* should exhibit altered expression in *sphk-1* mutants? Is this true? Seems like an easy test.

As requested, we crossed the *sphk-1(ok1097)* deletion mutant into our endogenous *asah-1::GFP::H2B* reporter and intriguingly we found that *asah-1* expression increases when *sphk-1* is absent. This increase in *asah-1* transcription is potentially due to the intestinal cells detecting a lack of S1P and upregulating *asah-1* expression in an attempt to provide more sphingosine (precursor of S1P). These data are shown in a graph below and not in the revised manuscript as we don't think that it provides substantial additional insight.

(5) RME-2/yolk deposits many lipids into oocytes. Because the specificity of this effect to S1P alone and not any other lipid is unclear, it would be helpful if the authors could assay other lipid pathways, using mutants, and demonstrate that decreasing the deposition of other lipids into oocytes does not have the same effect of offspring axon breaks. This is along the same lines as comment number 2. It would be important to rule out that supplementation of any lipid couldn't increase resistance to the mechanical stress that causes axon breaks.

We have addressed this question partially in point 2 above by incubating animals with various lipids. Of the lipids we tested, only ceramide (and S1P from the original manuscript) reduced PLM axon breaks.

To address this question genetically as the reviewer suggests however is challenging. If one used a mutant that affects yolk composition this may indirectly affect S1P transfer and would thus be difficult to interpret. In addition, by reducing various lipid levels one may affect cellular membrane composition that could affect neuronal health.

(6) In Fig. 6 the authors demonstrate increased *asah-1* expression in F1 animals, which is proposed to drive increased S1P production and continued protection from neuron breaks in F2 animals. However it remains unclear to me if an ~10% increase in *asah-1* expression would be sufficient to drive substantial increases in S1P levels deposited into F2 oocytes. This is particularly concerning because the authors demonstrate that supplementing S1P at both 5 and 10 micromolar did not affect axon breaks and they needed to get to 20 micromolar supplementation to see an effect – a rather large amount of S1P, so one might expect a substantial change in F1 S1P is required for this effect to be mediated via the mechanism proposed. Ideally, it would be

useful for the authors to quantify S1P levels in F1 and F2 embryos to see if the relatively modest change in *asah-1* expression can really drive a meaningful and impactful increase in S1P levels. (I am aware that the amount of S1P supplemented into media does not necessarily correlate with the concentrations in live *C. elegans*, however the authors could compare their metabolomics results to the amount of S1P increase that might be observed in animals supplemented with 10 or 20 micromolar S1P).

These are all good questions in theory, however, in practice S1P detection is extremely challenging. We attempted to detect S1P from 50ug of worms and were unable to do so. Thus, attempting to quantify S1P changes in F1 and F2 eggs as suggested is really not feasible. This is the main reason why we used a genetic approach using knockdown and overexpression of key sphingolipid biosynthetic enzymes and by S1P supplementation.

The reviewer suggests that a ~10% increase in *asah-1* transcripts is modest and is not clear whether this would affect a substantial increase in S1P levels. However, determining the effect of a 10% transcriptional increase of a biosynthetic enzyme is a challenge to associate with output of lipid product.

(7) The authors should remove “neurodegeneration” from the abstract/intro/results and perhaps leave the speculation on how this might apply to neurodegeneration (as people think about it in humans) for the discussion. What is studied here is axon breaks/axon fragility. Some definitions of neurodegeneration or links to classic neurodegeneration might consider this a model of neurodegeneration, but I suspect that many readers will be confused by this framing and misinterpret how much this actually applies to other models of neurodegenerations (right now it appears that the *mec-17*; *lon-2* model is only published in a single paper to date, so not exactly a well-established and long standing model). Plus, I do not think it hurts the authors main claim about the intergenerational effect to say it's an intergenerational protection of neurons from axon breaking, which is still important.

We agree and have now modified the text throughout to axon fragility or axon breaks rather than neurodegeneration.

Minor concerns that should be addressed and/or would benefit the manuscript:

(8) The authors show the individual replicate values for each replicate in Fig. 2G but do not do so in any other of the many bar graphs in this manuscript. To better interpret the variance and significance of these findings these values should be added. Along these lines, I am aware that the raw values are part of a supplemental table, which includes the percentages of animals in several replicates of each experiment but does not list the number of animals assayed in each experiment – only the total number of animals assayed across all experiments. (i.e. they report an “n” of 249 animals for one experimental condition, but then provide only the percentages for ~7-10 replicates without stating the number of animals assayed in each replicate). In this case it seems like the actual “n” for each experiment is 7-10 experiments and the number of animals assayed in each replicate should be included in the raw data.

We have now added individual replicate values to each graph in the revised manuscript.

“n” for each replicate has also now been included in the Source data file.

(9) The authors report that 49 genes were differentially expressed in response to UA. *Asah-1* was among them and they followed up on this one and it worked. It would be nice to know if they tested any of the other 48 and if they didn't work or would many of these genes do the same thing?

Due to our initial finding that maternal yolk transfer is required for the UA protective effect we focussed on *asah-1* as it was the only gene that had strong relevance to lipid/yolk biology. The other 48 genes were not investigated and is obviously well outside the scope of this extensive study.

(10) Sphk-1 antagonizes other ceramide related genes like *hyl-1*/*lagr-1*. It would be nice to see those mutants tested here to put this effect in context with other ceramide stress response studies in *C. elegans*.

As requested, we have now performed these experiments and found that knocking down each ceramide synthase encoding gene (*hyl-1*, *hyl-2* and *lagr-1*) does not affect PLM axons breaks in control or UA-treated *mec-17(-)*; *lon-2(-)* animals. This is not surprising as these genes have been shown to act in a partially redundant manner previously (PMID 30208318). As these data do not add anything substantive to this study, we provide the data below.

(11) The model in Fig. 7 could be cleaned up substantially to make it more interpretable. Also, for the F1/F2 generations in the model, are the authors proposing that the effect of S1P and/or asah-1 is in oocytes or developing embryos or in adults? Based on their experiments in the early figures of the manuscript the timing of these changes matters and the figure just labels generations with no regard to different developmental stages and timing of the action of these molecules. The model should be updated to account for these timing effects so readers can interpret it.

We have generated a new Figure 8 to enable interpretation of our data and that takes the timing of UA/S1P function into account.

(12) The authors should speculate on the potential role of this biology in wild-type animals. It is very unlikely such a mechanism would have evolved to prevent axon breaks in *mec-17; lon-2* mutants, and some more discussion of what the authors think they are actually tapping into here or why it might be important for other animals would be of high interest.

Our new data (new Figure 6) now show that UA and S1P reduce PLM axon breaks in wild type animals exposed to the microtubule de-stabilizing agent colchicine. In addition, UA and S1P promote axon transport of two models (UNC-104::GFP and mCherry::RAB-3). These data suggest that S1P enhances microtubule function and thereby axonal health. In addition to these new data, we discuss this implication and cite work from others (PMID 33380429) that support our conclusions.

Decision Letter, first revision:

Our ref: NCB-A49460A

5th June 2023

Dear Dr. Pocock,

Thank you for submitting your revised manuscript "Intergenerational Neuroprotection by an Intestinal Sphingolipid in *Caenorhabditis elegans*" (NCB-A49460A). Thank you so much for your patience with the re-review process, and I apologize for the delay in communicating our decision to you. The revision has now been seen by the original referees and their comments are below. The reviewers find that the paper has improved in revision, and therefore we'll be happy in principle to publish it in *Nature Cell Biology*, pending minor revisions to satisfy the referees' final requests and to comply with our editorial and formatting guidelines. In particular, we think it will be important to address the minor final comments of the reviewers and requests for text edits and clarifications. One reviewer also recommended that you include the ceramide data shown in response to Rev#4 pt #2 in the manuscript (not only in the rebuttal), and we agree this is a valuable addition for readers.

We are now performing detailed checks on your paper and will send you a checklist detailing our editorial and formatting requirements in about ~1-2 weeks. Please do not upload the final materials and make any revisions until you receive this additional information from us.

Thank you again for your interest in *Nature Cell Biology*. Please do not hesitate to contact me if you have any questions.

Sincerely,

Melina

Melina Casadio, PhD
Senior Editor, *Nature Cell Biology*
ORCID ID: <https://orcid.org/0000-0003-2389-2243>

Reviewer #1 (Remarks to the Author):

The authors have addressed all of my comments and suggestions adequately.

Reviewer #2 (Remarks to the Author):

Overall, the authors have addressed my comments. Based on the new data that organelle transport is not improved, I would still suggest to dampen the claim on the improvement of neural health.

Reviewer #3 (Remarks to the Author):

In this revised version of the manuscript, the authors have addressed the majority of the concerns raised, which has strengthened their conclusions and expanded the scope of their discovery. I commend the authors for a paper that is highly improved, and that now fits within the scope of Nature Cell Biology. There are a few minor elements that will need to be revised and that are indicated below.

1. Lines 115-117. Does the following statement refer to P0 or F1? "To assess the potency of UA-induced neuroprotection, we fed mec-17(ok2109); lon-2(e678) animals with different UA concentrations and examined PLM axon fragility in day 3 adults (Fig. 1a-d and S2a)." This needs to be clarified.
2. The authors do not discuss what makes the phenotype revert in the F3 generation. This seems an important open question, and a short paragraph included in the Discussion will enrich the paper.
3. In lines 165-167 it is stated that the incubation has been for 12 hours, rather than 16 hours as done before. Is there a reason for this?
4. In Fig 2g (related to the text in lines 185-188) there is only one image of intestinal expression (it is not clear if this is with or without UA). Two images should be provided, one with UA and the other the control with DMSO, which will allow the reader to visualize the difference in expression.
5. Line 205. The word "ability" in this sentence is confusing. Revise as: "...supporting the notion that ASAH-1 activity in the intestine protects the PLM neurons in the next generation".
6. Fig 4c (referred to in line 243) seems redundant as this is already shown in Fig 3a.
7. In line 288-289, the statement should remove the mention of Fig 6c (leaving only the reference 9) as the tested strain is MEC-17; LON-2. The mention of Fig 6c is fine in the following sentence in line 290.
8. Line 345. Revise typo: "...examined their role in UA-induced neuroprotection".
9. Line 381. The word "acute" in the supplementation might be misleading. The 16 hrs are a significant time considering C. elegans lifespan. Removing the word "acute" will still keep the main message of the sentence.
10. Line 452. There is a comma after the word Examples that should be removed.
11. Fig 5f, the images of uptake of the oocyte does not seem convincing. Can the authors use another and better image? In Panel 5g, there is a 10% on the x axis near rme-2. Is this supposed to be there and what does it mean?
12. Fig 6b, please annotate mec-4::GFP expression, as this can be confused with the previous UNC-104.
13. Line 61, add a comma after the word "changes".
14. Line 75, replace the word "worm" with "animal".
15. Line 93, revise as follows: "...axon protection, which account for the upregulation of asah-1 expression."
16. Line 111. Include panel d (Fig. 1b, d).
17. Line 121. Add a comma after "motility".
18. Line 127. "To investigate this aspect,..."
19. Line 220. "...system architecture is intact."

Reviewer #4 (Remarks to the Author):

In the revised version of this manuscript Wang et al., have added substantial additional data that adequately address all my previous concerns. This new manuscript will substantially advance the field and thus I think is suitable for publication in Nature Cell Biology.

Nick Burton

Decision Letter, second revision:

Our ref: NCB-A49460A

13th June 2023

Dear Dr. Pocock,

Thank you for your patience as we've prepared the guidelines for final submission of your Nature Cell Biology manuscript, "Intergenerational Neuroprotection by an Intestinal Sphingolipid in *Caenorhabditis elegans*" (NCB-A49460A). Please carefully follow the step-by-step instructions provided in the attached file, and add a response in each row of the table to indicate the changes that you have made. Please also check and comment on any additional marked-up edits we have proposed within the text. Ensuring that each point is addressed will help to ensure that your revised manuscript can be swiftly handed over to our production team.

In recognition of the time and expertise our reviewers provide to Nature Cell Biology's editorial process, we would like to formally acknowledge their contribution to the external peer review of your manuscript entitled "Intergenerational Neuroprotection by an Intestinal Sphingolipid in *Caenorhabditis elegans*". For those reviewers who give their assent, we will be publishing their names alongside the published article.

Nature Cell Biology offers a Transparent Peer Review option for new original research manuscripts submitted after December 1st, 2019. As part of this initiative, we encourage our authors to support increased transparency into the peer review process by agreeing to have the reviewer comments, author rebuttal letters, and editorial decision letters published as a Supplementary item. When you submit your final files please clearly state in your cover letter whether or not you would like to participate in this initiative. Please note that failure to state your preference will result in delays in

accepting your manuscript for publication.

Cover suggestions

As you prepare your final files we encourage you to consider whether you have any images or illustrations that may be appropriate for use on the cover of Nature Cell Biology.

Nature Cell Biology has now transitioned to a unified Rights Collection system which will allow our Author Services team to quickly and easily collect the rights and permissions required to publish your work. Approximately 10 days after your paper is formally accepted, you will receive an email in providing you with a link to complete the grant of rights. If your paper is eligible for Open Access, our Author Services team will also be in touch regarding any additional information that may be required to arrange payment for your article.

Please note that *Nature Cell Biology* is a Transformative Journal (TJ). Authors may publish their research with us through the traditional subscription access route or make their paper immediately open access through payment of an article-processing charge (APC). Authors will not be required to make a final decision about access to their article until it has been accepted. Find out more about Transformative Journals

Authors may need to take specific actions to achieve compliance with funder and institutional open access mandates. If your research is supported by a funder that requires immediate open access (e.g. according to Plan S principles) then you should select the gold OA route, and we will direct you to the compliant route where possible. For authors selecting the subscription publication route, the journal's standard licensing terms will need to be accepted, including self-archiving policies. Those licensing terms will supersede any other terms that the author or any third party may assert apply to any version of the manuscript.

Please use the following link for uploading these materials:
[Redacted]

Best regards,

Kendra Donahue
Staff
Nature Cell Biology

On behalf of

Melina Casadio, PhD
Senior Editor, Nature Cell Biology
ORCID ID: <https://orcid.org/0000-0003-2389-2243>

Reviewer #1:

Remarks to the Author:

The authors have addressed all of my comments and suggestions adequately.

Reviewer #2:

Remarks to the Author:

Overall, the authors have addressed my comments. Based on the new data that organelle transport is not improved, I would still suggest to dampen the claim on the improvement of neural health.

Reviewer #3:

Remarks to the Author:

In this revised version of the manuscript, the authors have addressed the majority of the concerns raised, which has strengthened their conclusions and expanded the scope of their discovery. I commend the authors for a paper that is highly improved, and that now fits within the scope of Nature Cell Biology. There are a few minor elements that will need to be revised and that are indicated below.

1. Lines 115-117. Does the following statement refer to P0 or F1? "To assess the potency of UA-induced neuroprotection, we fed mec-17(ok2109); lon-2(e678) animals with different UA concentrations and examined PLM axon fragility in day 3 adults (Fig. 1a-d and S2a)." This needs to be clarified.
2. The authors do not discuss what makes the phenotype revert in the F3 generation. This seems an

- important open question, and a short paragraph included in the Discussion will enrich the paper.
3. In lines 165-167 it is stated that the incubation has been for 12 hours, rather than 16 hours as done before. Is there a reason for this?
 4. In Fig 2g (related to the text in lines 185-188) there is only one image of intestinal expression (it is not clear if this is with or without UA). Two images should be provided, one with UA and the other the control with DMSO, which will allow the reader to visualize the difference in expression.
 5. Line 205. The word "ability" in this sentence is confusing. Revise as: "...supporting the notion that ASAH-1 activity in the intestine protects the PLM neurons in the next generation".
 6. Fig 4c (referred to in line 243) seems redundant as this is already shown in Fig 3a.
 7. In line 288-289, the statement should remove the mention of Fig 6c (leaving only the reference 9) as the tested strain is MEC-17; LON-2. The mention of Fig 6c is fine in the following sentence in line 290.
 8. Line 345. Revise typo: "...examined their role in UA-induced neuroprotection".
 9. Line 381. The word "acute" in the supplementation might be misleading. The 16 hrs are a significant time considering *C. elegans* lifespan. Removing the word "acute" will still keep the main message of the sentence.
 10. Line 452. There is a comma after the word Examples that should be removed.
 11. Fig 5f, the images of uptake of the oocyte does not seem convincing. Can the authors use another and better image? In Panel 5g, there is a 10% on the x axis near *rme-2*. Is this supposed to be there and what does it mean?
 12. Fig 6b, please annotate *mec-4::GFP* expression, as this can be confused with the previous UNC-104.
 13. Line 61, add a comma after the word "changes".
 14. Line 75, replace the word "worm" with "animal".
 15. Line 93, revise as follows: "...axon protection, which account for the upregulation of *asah-1* expression."
 16. Line 111. Include panel d (Fig. 1b, d).
 17. Line 121. Add a comma after "motility".
 18. Line 127. "To investigate this aspect,..."
 19. Line 220. "...system architecture is intact."

Reviewer #4:

Remarks to the Author:

In the revised version of this manuscript Wang et al., have added substantial additional data that adequately address all my previous concerns. This new manuscript will substantially advance the field and thus I think is suitable for publication in Nature Cell Biology.

Nick Burton

Author Rebuttal, first revision:

Reviewer #1:

Remarks to the Author:

The authors have addressed all of my comments and suggestions adequately.

Reviewer #2:

Remarks to the Author:

Overall, the authors have addressed my comments. Based on the new data that organelle transport is not improved, I would still suggest to dampen the claim on the improvement of neural health.

We don't think this is necessary. Our study does show that UA/S1P improve neural health.

Reviewer #3:

Remarks to the Author:

In this revised version of the manuscript, the authors have addressed the majority of the concerns raised, which has strengthened their conclusions and expanded the scope of their discovery. I commend the authors for a paper that is highly improved, and that now fits within the scope of Nature Cell Biology. There are a few minor elements that will need to be revised and that are indicated below.

1. Lines 115-117. Does the following statement refer to P0 or F1? "To assess the potency of UA-induced neuroprotection, we fed *mec-17(ok2109); lon-2(e678)* animals with different UA concentrations and examined PLM axon fragility in day 3 adults (Fig. 1a-d and S2a)." This needs to be clarified.

Changed to this:

To assess the potency of UA-induced neuroprotection, we fed *mec-17(ok2109); lon-2(e678)* mothers with different UA concentrations and examined PLM axon fragility in day 3 adult progeny.

2. The authors do not discuss what makes the phenotype revert in the F3 generation. This seems an important open question, and a short paragraph included in the Discussion will enrich the paper.

We did mention this in the discussion" We suggest that a threshold of intestinal S1P level must be maintained to initiate *asah-1* upregulation and this threshold may be lost over time – hence the reason for the intergenerational rather than transgenerational effect."

3. In lines 165-167 it is stated that the incubation has been for 12 hours, rather than 16 hours as done before. Is there a reason for this?

No reason for this. These were independent experiments asking different questions. No need to clarify in our opinion.

4. In Fig 2g (related to the text in lines 185-188) there is only one image of intestinal expression (it is not clear if this is with or without UA). Two images should be provided, one with UA and the other the control with DMSO, which will allow the reader to visualize the difference in expression.

We don't think this is necessary. We believe that the data presented in the graph below the image is sufficient.

5. Line 205. The word “ability” in this sentence is confusing. Revise as: “...supporting the notion that ASAH-1 activity in the intestine protects the PLM neurons in the next generation”.

Agreed - changed to:

“We found that the presence of the *Pges-1::asah-1* transgene in progeny was not required for PLM neuroprotection (Fig. 3b), supporting the notion that ASAH-1 activity in the hermaphrodite intestine protects the PLM neurons in the next generation.”

6. Fig 4c (referred to in line 243) seems redundant as this is already shown in Fig 3a.

Disagree - we believe these data are useful to present in Figure 4 so that comparisons between each knockdown can be made.

7. In line 288-289, the statement should remove the mention of Fig 6c (leaving only the reference 9) as the tested strain is MEC-17; LON-2. The mention of Fig 6c is fine in the following sentence in line 290.

Done.

8. Line 345. Revise typo: “...examined their role in UA-induced neuroprotection”.

Done.

9. Line 381. The word “acute” in the supplementation might be misleading. The 16 hrs are a significant time considering *C. elegans* lifespan. Removing the word “acute” will still keep the main message of the sentence.

Done.

10. Line 452. There is a comma after the word Examples that should be removed.

Done.

11. Fig 5f, the images of uptake of the oocyte does not seem convincing. Can the authors use another and better image? In Panel 5g, there is a 10% on the x axis near *rme-2*. Is this supposed to be there and what does it mean?

These are the best images we have been able to generate and clearly show the fluorescence in oocytes. We have removed the 10% on the x axis.

12. Fig 6b, please annotate mec-4::GFP expression, as this can be confused with the previous UNC-104.

Done.

13. Line 61, add a comma after the word “changes”.

Done.

14. Line 75, replace the word “worm” with “animal”.

Done.

15. Line 93, revise as follows: “...axon protection, which account for the upregulation of asah-1 expression.”

Disagree - we left this sentence unchanged.

16. Line 111. Include panel d (Fig. 1b, d).

Done.

17. Line 121. Add a comma after “motility”.

Done.

18. Line 127. “To investigate this aspect,…”

Disagree - we left this sentence unchanged.

19. Line 220. “..system architecture is intact.”

Done.

Reviewer #4:

Remarks to the Author:

In the revised version of this manuscript Wang et al., have added substantial additional data that adequately address all my previous concerns. This new manuscript will substantially advance the field and thus I think is suitable for publication in Nature Cell Biology.

Nick Burton

Final Decision Letter:

Dear Dr Pocock,

I am pleased to inform you that your manuscript, "An intestinal sphingolipid confers intergenerational neuroprotection", has now been accepted for publication in Nature Cell Biology.

If you have any questions about our publishing options, costs, Open Access requirements, or our legal

forms, please contact ASJournals@springernature.com

Please note that *Nature Cell Biology* is a Transformative Journal (TJ). Authors may publish their research with us through the traditional subscription access route or make their paper immediately open access through payment of an article-processing charge (APC). Authors will not be required to make a final decision about access to their article until it has been accepted. Find out more about Transformative Journals

If you have not already done so, we strongly recommend that you upload the step-by-step protocols used in this manuscript to the Protocol Exchange (www.nature.com/protocolexchange), an open online resource established by Nature Protocols that allows researchers to share their detailed experimental know-how. All uploaded protocols are made freely available, assigned DOIs for ease of citation and are fully searchable through nature.com. Protocols and Nature Portfolio journal papers in which they are used can be linked to one another, and this link is clearly and prominently visible in the online versions of both papers. Authors who performed the specific experiments can act as primary authors for the Protocol as they will be best placed to share the methodology details, but the Corresponding Author of the present research paper should be included as one of the authors. By uploading your Protocols to Protocol Exchange, you are enabling researchers to more readily reproduce or adapt the methodology you use, as well as increasing the visibility of your protocols and papers. You can also establish a dedicated page to collect your lab Protocols. Further information can be found at www.nature.com/protocolexchange/about

With kind regards,

Melina Casadio, PhD
Senior Editor, Nature Cell Biology
ORCID ID: <https://orcid.org/0000-0003-2389-2243>
